# DELE1 maintains muscle proteostasis to promote growth and survival in mitochondrial myopathy

Hsin-Pin Lin[1], Jennifer D Petersen[1], Alexandra J Gilsrud[1], Angelo Madruga [1], Theresa M D'Silva[1], Xiaoping Huang[1], Mario K Shammas[1], Nicholas P Randolph [1], Kory R Johnson[2], Yan Li[3], Drew R Jones[4], Michael E Pacold[5,6] & Derek P Narendra [1]✉

## Abstract

**Mitochondrial dysfunction causes devastating disorders, including mitochondrial myopathy, but how muscle senses and adapts to mitochondrial dysfunction is not well understood. Here, we used diverse mouse models of mitochondrial myopathy to show that the signal for mitochondrial dysfunction originates within mitochondria. The mitochondrial proteins OMA1 and DELE1 sensed disruption of the inner mitochondrial membrane and, in response, activated the mitochondrial integrated stress response (mt-ISR) to increase the building blocks for protein synthesis. In the absence of the mt-ISR, protein synthesis in muscle was dysregulated causing protein misfolding, and mice with early-onset mitochondrial myopathy failed to grow and survive. The mt-ISR was similar following disruptions in mtDNA maintenance (Tfam knockout) and mitochondrial protein misfolding (CHCHD10 G58R and S59L knockin) but heterogenous among mitochondria-rich tissues, with broad gene expression changes observed in heart and skeletal muscle and limited changes observed in liver and brown adipose tissue. Taken together, our findings identify that the DELE1 mt-ISR mediates a similar response to diverse forms of mitochondrial stress and is critical for maintaining growth and survival in early-onset mitochondrial myopathy.**

**Keywords** Mitochondria Unfolded Protein Response (mt-UPR); Mitophagy; Mitochondrial Disorders; Mitonuclear Communication; Mitohormesis
**Subject Categories** Organelles; Translation & Protein Quality

## Introduction

The mitochondrial network generates cellular energy through oxidative phosphorylation (OXPHOS), which is critical for tissue homeostasis. Mutations in over 250 genes can cause primary mitochondrial disorders by disrupting OXPHOS and its biosynthetic pathways (Mayr et al, 2015), often resulting in mitochondrial myopathy (MM) and abnormal growth (Boal et al, 2019). Patient survival may depend on how well striated muscle adapts to these perturbations in mitochondrial metabolism (Hathazi et al, 2020). A key adaptation is mediated by retrograde mitochondria-to-nucleus (mitonuclear) signaling and is aimed at changing the overall cellular state to accommodate decreased mitochondrial function.

The first mitonuclear signal in mammalian cells was identified in the setting of mitochondrial protein unfolding stress and was named the mitochondrial unfolded protein response (mt-UPR) (Zhao et al, 2002). Subsequently, an amino acid starvation-like transcriptional response was identified in skeletal muscle of the "deletor" mouse model, which has excess multiple mtDNA mutations in skeletal muscle, due to transgenic expression of a *Twinkle* variant (Forsström et al, 2019; Tyynismaa et al, 2010; Nikkanen et al, 2016; Tyynismaa et al, 2005). Similar transcriptional responses were subsequently described in the hearts of mouse models with mitochondrial cardiomyopathy due to defects in mtDNA maintenance (e.g., *Twinkle* or *Tfam* KO), mtRNA expression or stability (*Polrmt* or *Lrprrc* KO), and mtDNA translation (e.g., *Dars2* or *Mterf4* KO) (Dogan et al, 2014; Kühl et al, 2017), as well as models with cardiomyopathy due to mitochondrial protein misfolding (e.g., *Chchd10* S59L knock-in (KI) and *Lonp1* KO) (Anderson et al, 2019; Lu et al, 2023). These responses share activation of Amino Acid Response Element (AARE)-related genes, with upregulation of central anabolic pathways, such as those involved in serine, glutathione, and purine biosynthesis, in addition to mitokines like FGF21, which exert both paracrine and endocrine effects (Kühl et al, 2017; Nikkanen et al, 2016; Bao et al, 2016; Mick et al, 2020; Forsström et al, 2019). These AARE genes (also known as CARE genes) are defined by a 9 bp core element (TGATGXAAX) that is bound by the transcription factor activating transcription factor 4 (ATF4) and other bZIP transcription factors, including CCAAT/enhancer binding protein β (C/EBP), ATF2, ATF3, ATF5, and CHOP (Kilberg et al, 2009; Nakano et al, 2021). Responses upregulating these AARE genes have also been observed in cultured cells challenged with diverse inhibitors of OXPHOS, disruption of mitochondrial translation,

[1]Mitochondrial Biology and Neurodegeneration Unit, Neurogenetics Branch, National Institute of Neurological Disorders and Stroke, National Institutes of Health, Bethesda, MD 20892, USA. [2]Bioinformatics Core, National Institute of Neurological Disorders and Stroke, National Institutes of Health, Bethesda, MD 20892, USA. [3]Proteomics Core Facility, National Institute of Neurological Disorders and Stroke, National Institutes of Health, Bethesda, MD 20892, USA. [4]Department of Biochemistry and Molecular Pharmacology, NYU Langone Health, New York, USA. [5]Department of Radiation Oncology, NYU Langone Health, New York, USA. [6]Perlmutter Cancer Center, NYU Langone Health, New York, USA. ✉E-mail: derek.narendra@nih.gov

and inhibition of CHCHD4-dependent mitochondrial import (Mick et al, 2020; Forsström et al, 2019; Quirós et al, 2017), and were shown to depend on ATF4 in many of these cell models (Mick et al, 2020; Forsström et al, 2019; Quirós et al, 2017).

At least one mechanism activating ATF4 in cultured dividing cells was recently found to originate within mitochondria, through the OMA1-DELE1-HRI-eIF2α-ATF4 cascade (hereafter, the DELE1 mt-ISR) (Fessler et al, 2020; Guo et al, 2020). In this pathway, stress is sensed at the IMM by OMA1, which cleaves DELE1 as it is being imported into mitochondria, releasing short DELE1 (S-DELE1) into the cytosol, where it oligomerizes and binds the eIF2α kinase, HRI, activating it (Fessler et al, 2020; Guo et al, 2020; Yang et al, 2023). Active HRI phosphorylates eIF2α on serine 51, the defining molecular event in the integrated stress response (ISR). pS51-eIF2α, at least acutely, inhibits translation initiation of most transcripts and, at the same time, increases translation of ATF4 and other select transcripts with upstream open reading frames. Newly translated ATF4 (and perhaps other transcription factors) then upregulates AARE-related genes (Pakos-Zebrucka et al, 2016; Wek, 2018).

Recently, we demonstrated that the OMA1-DELE1 pathway can also signal the mt-ISR in vivo, in response to protein misfolding of CHCHD10 with the IMMD-causing mutation (Shammas et al, 2022). We additionally found that the OMA1-dependent stress response is strongly protective in the IMMD model. Similar findings were independently reported for two other cardiomyopathy models (one from conditional knockout (KO) of a Complex IV subunit, COX10 (Ahola et al, 2022), and the other from KO of the cardiolipin remodeling protein Tafazzin (Huynh et al, 2022; Zhu et al, 2022)), together supporting the idea that the DELE1 mt-ISR mediates at least some mitochondria-to-nucleus signaling in response to mitochondrial stress in vivo.

However, it remains unknown whether the OMA1-DELE1 pathway represents the dominant pathway for transcriptional responses to mitochondrial stress in striated muscle. For instance, it has been suggested that stress from defects of mtDNA maintenance or expression may not robustly activate OMA1 (Forsström et al, 2019; Shammas et al, 2022; Guo et al, 2020; Kühl et al, 2017), in contrast to stressors resulting in greater IMM disruption. Similarly, a study of cultured C2C12 myoblasts and myotubes, a common cell culture model of striated muscle, concluded that there may be multiple upstream stress signaling pathways converging on ATF4-targeted genes (Mick et al, 2020). This raises the possibility that AARE genes are activated by different mechanisms in response to defects of mtDNA maintenance and expression in striated muscle, compared to those resulting from damage to the IMM (e.g., from protein unfolding, cardiolipin remodeling defects, or loss of a structural OXPHOS proteins). In addition, while mitochondrial stress responses in skeletal muscle are well-described (Mick et al, 2020; Forsström et al, 2019), it is not known whether they are also mediated through activation of the ISR via the OMA1-DELE1 pathway, similar to what was recently reported in the heart. Finally, it is not known whether the OMA1-DELE1 mt-ISR is equally critical in models with early-onset vs. late-onset mitochondrial stress.

To address these outstanding questions, we developed a novel *Dele1* KO mouse and crossed it four models of MM/mitochondrial cardiomyopathy (MC), each differing in the mechanism and temporal onset of mitochondrial stress. These included models with either early or late mitochondrial protein unfolding stress, due to two different KI mutations in the mitochondrial protein CHCHD10 (G58R and S59L); a model with severe mitonuclear imbalance from a defect in mtDNA maintenance (*Tfam* KO) (Hansson et al, 2004); and a model with relatively preserved OXPHOS but structural damage to the mitochondrial cristae (due to double KO (DKO) of the paralogs *Chchd2* and *Chchd10*). Comparing the DELE1 mt-ISR among these models, our data establish several key findings: (1) DELE1 is responsible for a protective ISR mitochondrial stress in response to both defects of mtDNA maintenance and protein unfolding in striated muscle; (2) the DELE1 mt-ISR is especially critical for growth and survival in the setting of early-onset mitochondrial stress; (3) DELE1 and OMA1 have overlapping but separable protective effects in MM, (4) the DELE1 mt-ISR increases metabolic flexibility to maintain pro-anabolic pathways and growth but fails to resolve underlying structural defects and OXPHOS deficiency, (5) DELE1 mediates overlapping transcriptionally responses in mitochondrial-rich tissues, including heart, skeletal muscle, brown fat, and liver, with the highest overlap between heart and skeletal muscle, and (6) DELE1 is required to maintain translation-related protein folding in skeletal muscle under stress.

## Results

### Basal mitochondrial homeostasis is unaltered in Dele1 KO mice

We previously found that OMA1 promotes survival in a model of early-onset MM (Shammas et al, 2022), via the OMA1-DELE1 pathway (Fessler et al, 2020; Guo et al, 2020) (Appendix Fig. S1A, top).

To isolate the effects of OMA1 on the mt-ISR, we generated a constitutive *Dele1* KO mouse by CRISPR-Cas9 genome editing (Appendix Fig. S1A, bottom). No antibodies are currently available to detect endogenous DELE1, however, we verified that the mutation destabilizes *Dele1* mRNA in several tissues, including brown adipose tissue (BAT), heart, gastrocnemius skeletal muscle, and liver, likely by nonsense-mediated decay (Appendix Fig. S1B). In addition, the predicted truncated protein product (resulting from the predicted frameshift mutation p.Arg21GlnfsTer19), if expressed, would lack part of the targeting sequence needed for mitochondrial localization and the tetratricopeptide repeat (TPR) domains required for HRI binding and downstream signaling (Yang et al, 2023).

*Dele1* KO mice appeared grossly normal with similar body weight to wild-type (WT) mice up to at least 4–6 months of life (Appendix Fig. S1C), and the heart mitochondrial proteome was not significantly altered by *Dele1* KO (Appendix Fig. S1D). Consistently, mitochondria from WT and *Dele1* KO hearts were indistinguishable by thin section transmission electron microscopy (TEM) at 28 days (Appendix Fig. S1E), with similar median areas (0.57 vs. 0.61 μm²) and aspect ratios (1.47 vs. 1.50) (Appendix Fig. S1F and S1G). Together, these findings suggest that *Dele1* KO does not disrupt mitochondria in the absence of mitochondrial stress.

## DELE1 activates the mt-ISR in response to diverse mitochondrial stressors in striated muscle to promote growth and survival

We next asked whether DELE1 is responsible for the mt-ISR in response to diverse mitochondrial stressors. To address this question, we crossed the *Dele1* KO mouse with four different mouse models of MM/Cardiomyopathy (CM) (Fig. 1A–L).

Importantly, the primary mitochondrial stress was different in each model. *Tfam* mKO is the prototypical model of OXPHOS deficiency due to a defect mtDNA maintenance (Hansson et al, 2004). C10 G58R and C10 S59L are models of intramitochondrial protein misfolding (Shammas et al, 2022; Genin et al, 2019; Anderson et al, 2019; Liu et al, 2020). The C10 G58R and S59L mutations cause CHCHD10 to misfold into two distinct toxic conformations within the mitochondria IMS with differential impacts on the mitochondria and the overall phenotype (Shammas et al, 2022). C10 G58R has an earlier and more severe impact on skeletal muscle than C10 S59L, leading to early-onset mitochondrial myopathy in both mice and humans (Shammas et al, 2022; Ajroud-Driss et al, 2015; Heiman-Patterson et al, 1997; Bannwarth et al, 2014; Anderson et al, 2019; Genin et al, 2019). C2/C10 DKO likely also disrupts mitochondrial cristae proteostasis but has a milder phenotype than protein misfolding from C10 G58R and C10 S59L (Liu et al, 2020; Huang et al, 2018). Indeed, the C2/C10 DKO are remarkable for having a near normal lifespan and healthspan, despite having early and pervasive activation of the mt-ISR (Liu et al, 2020; Nguyen et al, 2022).

Despite the diversity of the underlying stress, DELE1 was protective in each of the four models (Fig. 1C,F,I,L). The survival benefit was especially pronounced for the C2/10 DKO and C10 G58R models (Fig. 1C,F), which also had the earliest activation of the mt-ISR, indicated by elevation of the maker protein MTHFD2 by postnatal day 14 (P14) (Fig. 1M,N). They also exhibited the greatest OMA1 activation, measured by cleavage of L-OPA to the OMA1-specific S-OPA1 products (Fig. 1M; Appendix Fig. S2). Strikingly, the C2/C10 DKO, which have a near normal lifespan and healthspan in the presence of DELE1 (Liu et al, 2020; Nguyen et al, 2022), lived a median of 10 days without DELE1 (Fig. 1C). Similarly, C10 G58R mice, which have a median life expectancy of more than 18 months with DELE1 (Shammas et al, 2022), survived a median of 1 month in the absence of DELE1 (Fig. 1F).

The early death in these models correlated with decreased growth in the first weeks of life. C2/C10/Dele1 triple KO mice had a similar weight as their siblings at birth, but their weight gain slowed from P4 (Fig. 2B). Similarly, weights of C10 G58R; *Dele1* KO mice were reduced relative to their littermates at 21 days and did not further increase from P21 to P28 (Fig. 1E). The failure to gain body mass also correlated with decreased motor function: C10 G58R; *Dele1* KO mice had decreased grip strength and an increased composite phenotype score compared to their C10 G58R; *Dele1* + (either +/+ or +/−) siblings at 28 days (Appendix Fig. S3A). Hand feeding only modestly increased survival of C10 G58R; *Dele1* KO mice (by 10 days), suggesting that food access was a minor contributor to their failure to thrive (Appendix Fig. S3B). Hypertrophic growth of skeletal muscle accounts for about half of the 8-fold body mass increase in the first three weeks of life (White et al, 2010; Gokhin et al, 2008), and so we examined muscle fiber hypertrophy by measuring the cross-sectional area (CSA) of muscle fibers in the gastrocnemius muscle. Consistent with a decrease in skeletal muscle hypertrophy, body weights directly correlated with muscle fiber CSA among littermates in the C10 G58R; *Dele1* KO litters ($r^2 = 0.7272$; Appendix Fig. S3C). Thus, in the two MM/CM models with early DELE1 mt-ISR activation, early death correlated with decreased growth in the early weeks of life and worsening motor function.

In the other two models, *Tfam* mKO and C10 S59L, the mt-ISR was activated later: after P14 for *Tfam* mKO and after P28 for C10 S59L (Fig. 1M,N). OMA1 cleavage of OPA1 was less robust for Tfam mKO compared to the other models, but could be detected preceding MTHFD2 elevation, when the five isoforms of OPA1 were separated on a 7.5% Tris-Glycine gel (Fig. 1N; Appendix Fig. S2). The late activation of the mt-ISR in *Tfam* mKO mice likely reflects the postnatal expression of Cre from the *Ckmm* promoter, which has been estimated to recombine floxed alleles in heart and skeletal muscle between P7 and P21 (He et al, 2010). Late activation of the mt-ISR in C10 S59L mice is likewise consistent with our prior observations that C10 S59L reaches a higher protein abundance and takes longer to trigger an OMA1 stress response in both cultured cells and heart tissue compared to C10 G58R (Shammas et al, 2022). Body weights of *Tfam* mKO and C10 S59L were similar in the presence or absence of DELE1 (Fig. 1H,K). However, their heart to body weight ratio were significantly higher in the absence of DELE1 (Appendix Fig. S3D,E), suggesting exacerbation of cardiomyopathy as the likely cause of early mortality in the absence of DELE1 (Fig. 1I,L).

Considered together, these findings suggest that the DELE1 mt-ISR is generally protective against diverse sources of mitochondrial stress in striated muscle and may be particularly critical when mitochondrial stress is present in striated muscle during early postnatal growth (Appendix Fig. S3F).

## DELE1 functionally overlaps with OMA1 to protect against CHCHD10 myopathy

Next, we utilized the strong phenotype observed for C10 G58R; *Dele1* KO mice to genetically dissect the OMA1-DELE1 pathway. OMA1 has multiple substrates in addition to DELE1 in the mitochondria, including most notably the mitochondrial fusion protein OPA1 (Ehses et al, 2009; Head et al, 2009) (Appendix Fig. S1A). It is not known if the DELE1 mt-ISR confers all the survival benefit of the OMA1 stress response or whether cleavage of other substrates such as OPA1 also promote survival. In addition, recent cellular studies suggest that while OMA1 facilitates DELE1 signaling, in some settings DELE1 can signal the mt-ISR without cleavage by OMA1 (Sekine et al, 2023; Fessler et al, 2022). This would predict that *Dele1* KO may have a stronger effect on the mt-ISR than *Oma1* KO, but this has not yet been tested in vivo.

To address these questions, we directly compared the phenotypes of *Dele1* KO and *Oma1* KO mice under mitochondrial stress from C10 G58R protein misfolding, in litters triple mutant for C10 G58R, *Dele1*, and *Oma1* (Fig. 2A). In these litters, *Oma1* KO had a substantially stronger effect on survival than *Dele1* KO (median survival 6.5 vs. 35 days), demonstrating that the *Oma1* stress response protects through multiple mechanisms, in addition to its activation of the DELE1 mt-ISR (Fig. 2A).

To identify additional mechanisms of OMA1 protection, we genetically blocked OMA1 from cleaving its other major substrate,

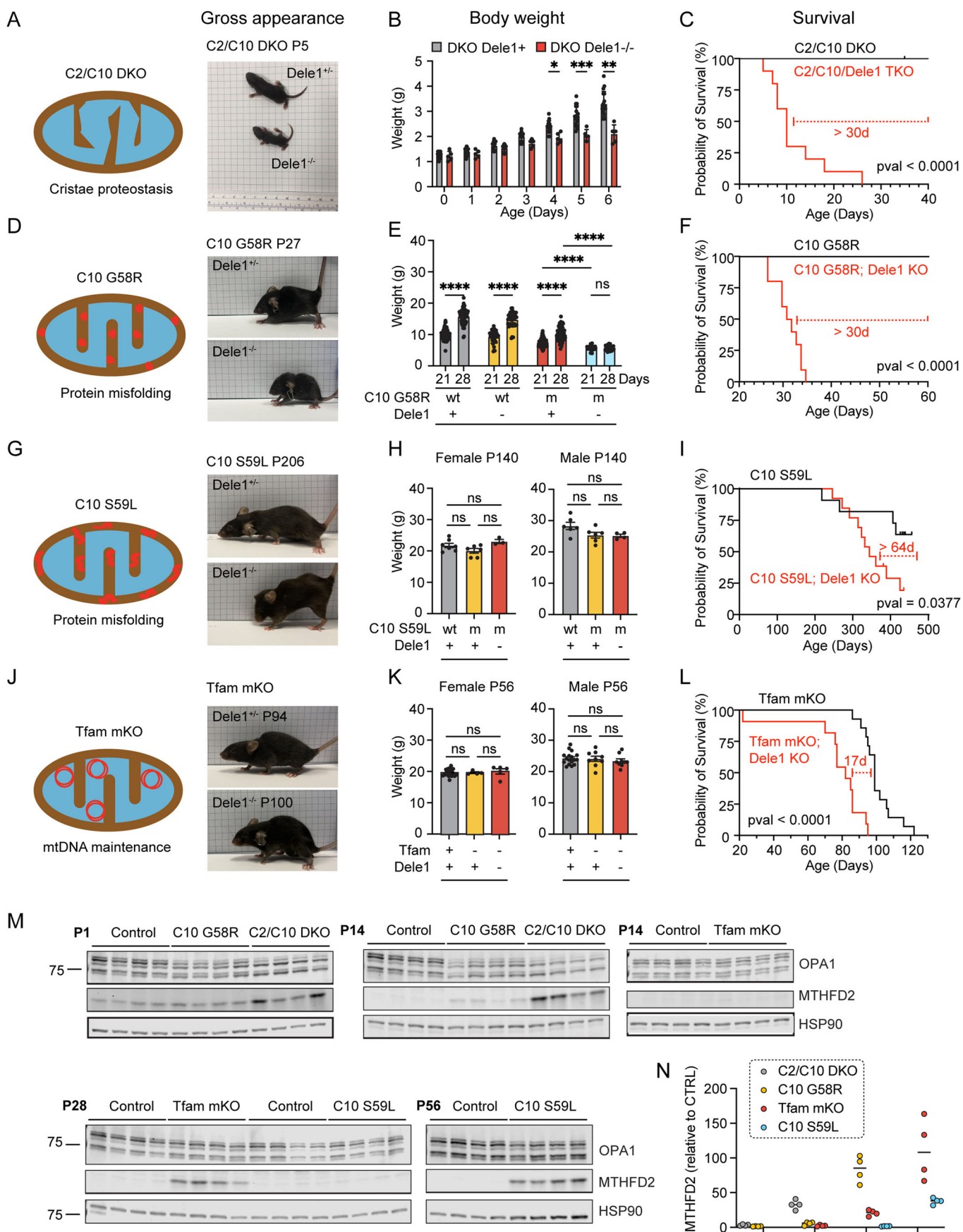

**Figure 1. Dependence on DELE1 for growth and survival correlates with onset of mitochondrial stress in diverse models of myopathy and cardiomyopathy.**

(A) Schematic depicting the cristae proteostasis stress induced by C2/C10 DKO and gross appearance of C2/C10 DKO mice with and without *Dele1* at P5. (B) Body weights of C2/C10 DKO mice with and without *Dele1* (*Dele1+* indicates either *Dele1*$^{+/-}$ or *Dele1*$^{+/+}$). Statistical analysis performed using mixed-effects model with Geisser-Greenhouse correction and Šidák's multiple comparisons test with individual variances computed for each comparison. N ≥ 3 mice per group (genotype/age). *, **, *** indicate p ≤ 0.05, 0.01, 0.001, respectively. From left to right, adjusted p-values are 0.0359, 0.0007, and 0.0017. Bar and error bars indicate mean and standard deviation (SD), respectively. (C) Survival analysis of C2/C10 DKO mice. Statistical analysis performed with Log-rank (Mantel-Cox) test. (D) Cartoon depicting protein misfolding stress of C10 G58R (aggregates in IMS) and gross appearance of C10 G58R mice with and without *Dele1* at P27. (E) Body weights of C10 G58R mice with and without *Dele1* (*Dele1+* indicates either *Dele1*$^{+/-}$ or *Dele1*$^{+/+}$). wt = wild type; m = mutant. Statistical analysis performed with Welch ANOVA and Dunnett's T3 multiple comparisons test. N ≥ 16 mice per group (genotype/age). **** indicates p ≤ 0.0001, "ns" not significant. Adjusted p-values are <0.0001, <0.0001, <0.0001, 0.9986 (bottom row, left to right), <0.0001 (middle row), <0.0001 (top row). Bar and error bars indicate mean and SD, respectively. (F) Survival analysis of C10 G58R mice. Statistical analysis performed with Log-rank (Mantel-Cox) test. (G) Cartoon depicting protein misfolding stress of C10 S59L (aggregates in IMS) and gross appearance of C10 S59L mice with and without Dele1 at P206. (H) Body weights of C10 S59L mice with and without *Dele1* (*Dele1+* indicates either *Dele1*$^{+/-}$ or *Dele1*$^{+/+}$). wt = wild type; m = mutant. Statistical analysis performed with Welch ANOVA and Dunnett's T3 multiple comparisons test. Bar and error bars indicate mean and standard error of the mean (SEM), respectively. N ≥ 3 mice per group (genotype/age/sex). (I) Survival analysis of C10 S59L mice. Statistical analysis performed with Log-rank (Mantel-Cox) test. (J) Cartoon depicting disruption of mtDNA maintenance stress induced by *Tfam* mKO, and gross appearance of *Tfam* mKO mice with Dele1 at P94 or without Dele1 at P100. (K) Body weights of *Tfam* mKO mice with and without *Dele1* (*Dele1+* indicates either *Dele1*$^{+/-}$ or *Dele1*$^{+/+}$). wt = wild type; m = mutant. Statistical analysis performed with Welch ANOVA and Dunnett's T3 multiple comparisons test. Bar and error bars indicate mean and SEM, respectively. N ≥ 4 mice per group (genotype/age/sex). (L) Survival analysis of *Tfam* mKO mice. Statistical analysis performed with Log-rank (Mantel-Cox) test. (M) Immunoblots of OPA1 cleavage and mt-ISR marker protein MTHFD2 expression, showing differential time course of activation of the mt-ISR postnatally, in the sequence C2/C10 DKO, C10 G58R, *Tfam* mKO, and C10 S59L. N = 4 mice per timepoint. Note: in wild-type mouse hearts MTHFD2 is expressed embryonically and on P1 but declines during postnatal development (Nilsson et al, 2014). (N) Quantification of MTHFD2 protein levels from immunoblots in (M). Levels for each group and timepoint are normalized to littermate controls except for C2/C10 DKO, which were normalized to controls for C10 G58R. Median values indicated by bars. N = 4 mice per group (genotype/age). Source data are available online for this figure.

OPA1, by generating a novel transgenic mouse line in which the preferred OMA1 cleavage site within OPA1, s1, is deleted (Ishihara et al, 2006) (Fig. 2B; Appendix Fig. S3G). In primary fibroblasts from *Opa1*$^{\Delta s1/\Delta s1}$ mice, the s1 deletion blocked basal cleavage and mitigated (but did not completely block) stress-induced cleavage by OMA1 (Appendix Fig. S3G). The residual cleavage under stress is consistent with prior reports and suggests that OMA1 can cleave OPA1 at other sites in the absence of s1, albeit less efficiently (Ishihara et al, 2006). We next crossed *Opa1*$^{\Delta s1/\Delta s1}$ and C10 G58R mice (Fig. 2B). Elimination of the s1 site reduced OMA1 cleavage of L-OPA1 both basally and in response to C10 G58R protein misfolding stress; cleavage of the *b* band to the *e* band was reduced by approximately 56% under basal conditions and 36% with C10 G58R stress, based on the e/b ratio (Fig. 2B). As expected, the Opa1Δs1 allele did not block the mt-ISR, as indicated by elevation of the marker protein MTHFD2. Partially blocking OPA1 cleavage in C10 G58R mice reduced body weight (significantly in male but not female mice) and grip strength at 13 weeks, suggesting that OPA1 cleavage by OMA1 is important for maintaining muscle function (Fig. 2C). However, the overall effect on C10 G58R mice was less pronounced than the effect of *Oma1* KO or *Dele1* KO, and the composite phenotype score and median survival were not significantly reduced (Fig. 2D; Appendix Fig. S3H). The weak effect of the *Opa1* Δs1 allele is likely due to its incomplete block of OMA1 cleavage, although it is also possible OMA1 protects by cleaving other substrates in addition to OPA1 and DELE1. Nonetheless, these results suggest that cleavage of OPA1 by OMA1 is protective in vivo, independent of its activation of the mt-ISR.

We next considered whether DELE1 also retains some of its function in the absence of OMA1. We previously observed that approximately a quarter of C10 G58R; *Oma1* KO mice escape neonatal lethality and, surprisingly, have upregulation of mt-ISR-associated genes in the heart at 14 weeks (Shammas et al, 2022). This contrasted with knockdown of *Oma1* in adult C10 G58R animals, which reliably suppressed the mt-ISR-associated genes (Shammas et al, 2022). Using gene expression analysis, we assessed

whether the same genes could also be activated in the absence of DELE1. Notably, all C10 G58R; *Dele1* KO animals failed to activate this prespecified mt-ISR gene signature (Fig. 2E), including one outlier that survived to 16 weeks, close to the age of C10 G58R; *Oma1* KO mice with an activated ISR (Appendix Fig. S3I,J). Consistent with the idea that DELE1 retains some limited function independent of OMA1, the C10 G58R; *Dele1* KO; *Oma1* KO triple mutants died earlier than the double mutant animals (Fig. 2A, yellow vs. blue and red lines). Together these findings demonstrate that DELE1 is strictly required for activation of the mt-ISR in the setting of MM, in sharp contrast to OMA1. Thus, OMA1 and DELE1 have overlapping but separable protective effects on striated muscle in vivo.

## DELE1 mt-ISR has minor effects on OXPHOS complex subunit expression and function

Next, to better understand how the DELE1 mt-ISR mediates protection against mitochondrial dysfunction, we considered whether the DELE1 mt-ISR corrects the underlying mitochondrial defect caused by stress in each MM/CM model.

We first examined the effect of the *Dele1* KO on OXPHOS subunit expression and function (Fig. 2F–K; Appendix Fig. S4A,B). The models differed in the degree of OXPHOS subunit disruption and activity in the heart, with *Tfam* mKO most severely affected, followed by C10 G58R, and then by C2/C10 DKO (Fig. 2F–H,J,K; Appendix Fig. S4A,B). In Tfam mKO, mtDNA encoded subunits were more affected than nuclear encoded subunits, consistent with a primary defect in mtDNA maintenance and expression; by contrast, in C10 G58R mtDNA and nuclear encoded subunits were similarly affected, suggesting a mechanism independent of mtDNA maintenance or expression (Fig. 2I). Notably, *Dele1* KO did not significantly change OXPHOS subunit levels or CI or CIV activities in either C10 G58R or *Tfam* mKO models (Fig. 2F,G, bottom, Fig. 2J; Appendix Fig. S4A). Thus, the DELE1 mt-ISR does not resolve the underlying OXPHOS deficiency in heart muscle.

 

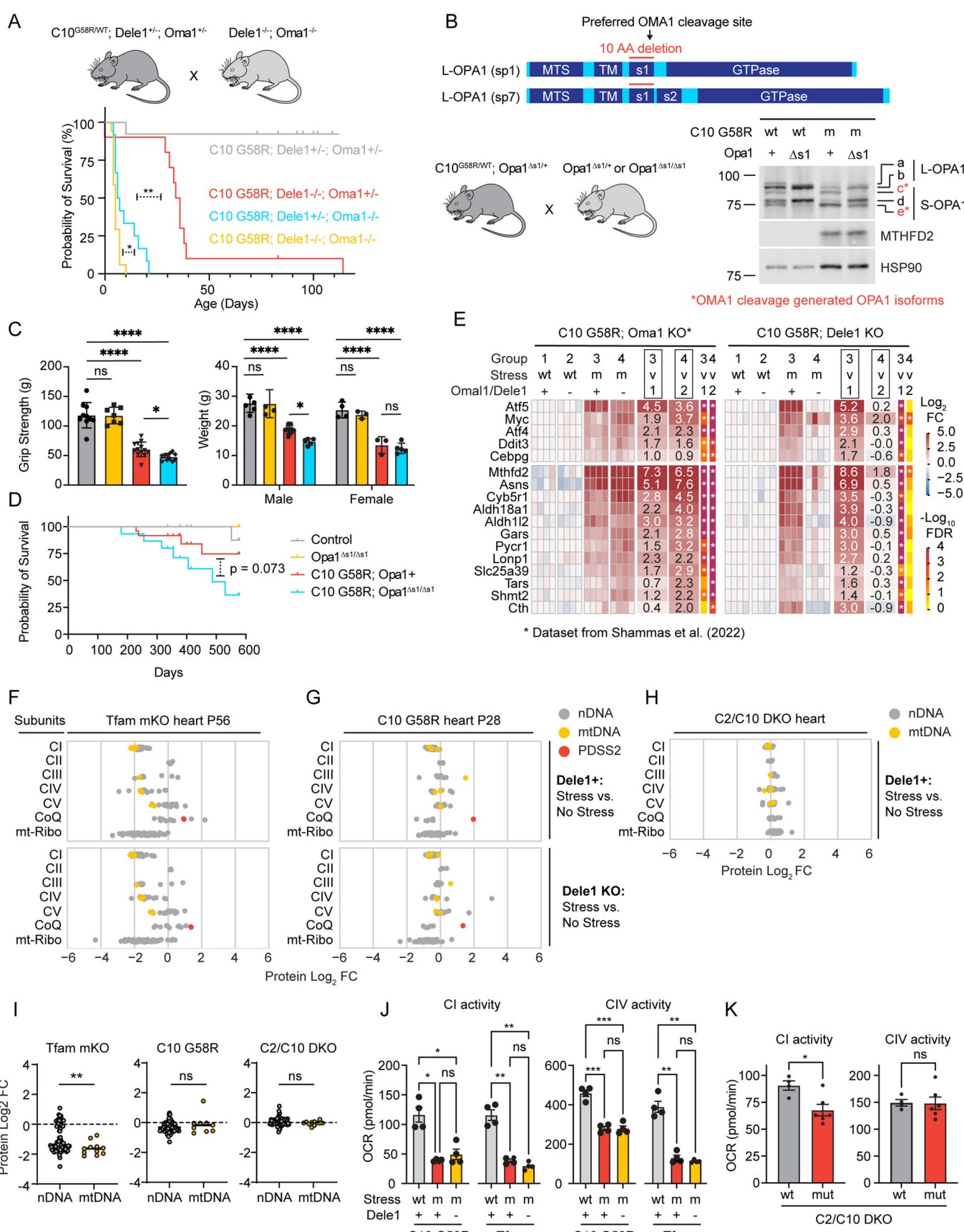

**Figure 2.   The DELE1 mt-ISR mediates only some of the protective OMA1 stress response and does not reverse underlying OXPHOS defect.**

(A) Illustration of breeding strategy to create the cohort. Survival analysis for C10 G58R; *Oma1* KO; *Dele1* KO triple mutant mice and their littermates. Statistical analysis performed with Log-rank (Mantel-Cox) test. *, **** indicate $p \leq 0.05$ and $p \leq 0.0001$, respectively. From left to right, *p*-values are 0.0127 and <0.0001. (B) Schematic demonstrating generation of *Opa1^{Δs1/Δs1}* mice through a 30 bp deletion removing the OMA1 preferred cleavage site, s1, from OPA1. Wild-type splice variants 1 and 7 (sp1 and sp7) contain the s1 site. (top); and OPA1 cleavage patterns of C10 G58R; *Opa1^{Δs1/Δs1}* mice and control mouse heart lysates (bottom, right). (C) Grip strength (left) and body weights (right) of C10 G58R; *Opa1^{Δs1/Δs1}* and their littermates at 13 weeks. Gray bar = control; yellow bar = *Opa1^{Δs1/Δs1}*; red bar = C10 G58R; blue bar = C10 G58R; *Opa1^{Δs1/Δs1}*. Each dot represents one mouse. Statistical analysis for grip strength (left graph) performed with Welch ANOVA and Dunnett's T3 multiple comparisons test. $N \geq 7$ mice per group (genotype/age/sex). Adjusted *p*-values are 0.0243 (bottom row), >0.9999 (second from bottom), <0.0001 (third from bottom), <0.0001 (top row). (right) (H) Statistical analysis for body weight performed with an ordinary two-way ANOVA with Tukey's multiple comparisons test, with a single pooled variance. (for males) adjusted *p*-values are 0.0298 (bottom), 0.9997(second from bottom), <0.0001 (third from bottom), <0.0001 (top). (for females) adjusted *p*-values are 0.9241 (bottom), 0.8223 (second from bottom), <0.0001 (third from bottom), <0.0001 (top). In all graphs, *, **** indicates $p \leq 0.05$ and $p \leq 0.0001$, respectively, "ns" not significant. Bar and error bars indicate mean and SD, respectively. (D) Survival analysis of C10 G58R; *Opa1^{Δs1/Δs1}* mice and their littermates. Statistical analysis performed with Log-rank (Mantel-Cox) test. (E) Gene expression analysis of prespecified ISR genes from C10 G58R; *Dele1* KO mice and littermates at P28 compared to a previously published dataset from 14-week-old C10 G58R; *Oma1* KO mice (Shammas et al, 2022). Data from same C10 G58R; *Dele1* KO dataset also appears in (Appendix Figs. S1B; Figs. 4A–D,G and 6C). wt = wild type; m = mutant. (F) Scatterplot depicting relative abundance of OXPHOS complexes I–V subunits, Coenzyme Q biosynthesis pathway, and mito-ribosome in *Dele1* KO mice and littermates detected from crude mitochondrial preparations of mouse hearts. See also Appendix Fig. S4A for statistical analysis. $N = 4$ mice per group. Data from this proteomics dataset also appears in Fig. 4F and Appendix Fig. S7B,C,E. (G) Scatterplot depicting relative abundance of OXPHOS complexes I–V subunits, Coenzyme Q biosynthesis pathway, and mito-ribosome in C10 G58R; *Dele1* KO mice and littermates detected from crude mitochondrial preparations of mouse hearts. See also Appendix Fig. S4A for statistical analysis comparing differences in subunit expression between genotypes. $N = 4$ mice per group. Data from this proteomics dataset also appears in Appendix Fig. S1D, Fig. 4E,G, and S7C,D. (H) Scatterplot depicting relative abundance of OXPHOS complexes I–V subunits, Coenzyme Q biosynthesis pathway, and mito-ribosome in C2/C10 DKO and unrelated age-matched WT animals detected from crude mitochondrial preparations of mouse hearts. See also Appendix Fig. S4A for statistical analysis. $N = 4$ mice per group. (I) Graphs compare protein abundance of mtDNA- vs. nDNA-encoded OXPHOS subunits from data in (F–H). Statistical analysis for Tfam (left graph) was performed using the Mann–Whitney test, as the data were not normally distributed, and statistical analysis for C10 G58R (middle graph) and C2/C10 DKO (right graph) were performed using the unpaired Welch's t-test. *P*-values = 0.0087 (left graph), 0.4005 (middle graph), and 0.1508 (right graph). In all graphs, ** indicates $p \leq 0.01$ and "ns" not significant. Bars indicate median values. (J, K) Seahorse oxygen consumption-based measurements of CI and CIV activities from frozen mitochondria isolated from hearts of the indicated genotypes. wt = wild type; m or mut = mutant. Statistical analysis performed with Welch ANOVA and Dunnett's T3 multiple comparisons test (J) and unpaired Welch's t-test (K). In (J, left most graph) *p*-values = 0.02304 and 0.6378 (bottom row, left to right) and 0.0188 (top row); in (J, second graph from left) *p*-values = 0.0032 and 0.1719 (bottom row, left to right) and 0.0022 (top row); in (J, third graph from left) *p*-values = 0.0006 (bottom), >0.9999 (middle) and 0.0006 (top)); in (J, right most graph) *p*-values = 0.0029 (bottom), 0.5977 (middle) and 0.0054 (top). In (K), *p*-value = 0.0102 (left graph) and 0.9437 (right graph). In all graphs, *, **, and *** indicate $p \leq 0.05$, $p \leq 0.01$, $p \leq 0.001$, respectively, and "ns" not significant. Bars indicate mean values and error bars SEM. Source data are available online for this figure.

## DELE1 mt-ISR has minor effects on mitochondrial ultrastructure

We next examined the ultrastructure of heart mitochondria by thin section TEM. Mitochondria were analyzed using a combination of deep-learning segmentation and manual scoring of key morphological features in >600 mitochondria per genotype (8923 mitochondria evaluated in total) (scheme depicted in Fig. 3A). As expected, the morphological defects were highly diverse among the four models, reflecting the diversity of the underlying mitochondrial stress. However, each model caused disruption of the cristae and the contiguous IMM, the site of both OXPHOS and OMA1 sensing of mitochondrial stress (Fig. 3B–F).

The C10 G58R model showed three prominent classes of abnormal mitochondria: mitochondria with inclusions composed of cristae membranes (18% of mitochondria); nano-mitochondria, defined as measuring <250 nm in their minor axis with an aspect ratio less than 1.5 (6% of mitochondria); and electrolucent mitochondria characterized by an enlarged matrix area absent of electron-dense substance and fewer cristae (5% of mitochondria) (Figs. 3C,G and EV1C–J), similar to the phenotype we observed previously (Shammas et al, 2022). Mitochondrial ultrastructure in adult C2/C10 DKO hearts resembled that of C10 G58R, but with decreased frequency of cristal inclusions (4% of mitochondria) and electrolucent mitochondria (2% of mitochondria) (Figs. 3D,H and EV2). This suggests that loss of C2/C10 function and C10 G58R misfolding may exert a similar stress on the IMM, which is more severe in the adult heart with C10 G58R misfolding.

Notably, similar proportions of nano-mitochondria and cristal inclusions were observed in C10 G58R mice in the presence or the

absence of one copy of *Dele1* (Fig. 3G). Likewise, the average area of mitochondria was the same in the presence and absence of *Dele1* (Fig. EV1A). In contrast, electrolucent mitochondria were more frequent in the absence of *Dele1* (Fig. 3G). This trend was also found in C10 G58R; *Oma1* KO hearts, upon reanalyzing samples from a previously published study (Shammas et al, 2022) (Appendix Fig. S5A–E). In both C10 G58R; *Dele1* KO and C10 G58R; *Oma1* KO cells, there was a high degree of cellular heterogeneity; in some cardiomyocytes most of the mitochondrial area was occupied by large electrolucent mitochondria (Appendix Fig. S5C). These electrolucent mitochondria were also more likely to undergo outer mitochondrial membrane (OMM) rupture and mitophagy, events that were observed almost exclusively in the absence of *Dele1* (Figs. 3G and EV1I,J). Taken together these findings suggest that loss of the OMA1-DELE1 mt-ISR is responsible for the increase in electrolucent mitochondria.

Notably, in contrast to the C10 G58R; *Dele1* KO samples, C10 G58R; *Oma1* KO samples had a prominent population of elongated mitochondria (black arrows in Appendix Fig. S5B and quantified in Appendix Fig. S5F), indicating that OMA1 mediates morphological changes independently of the DELE1 mt-ISR. This is most likely through L-OPA1 cleavage by OMA1, although this was not demonstrated conclusively.

In contrast to C10 G58R mice, the C10 S59L mice had few intracristal inclusions and electrolucent mitochondria, suggesting that C10 S59L exerts a different stress on mitochondria than C10 G58R, despite the proximity of the mutations (Figs. 3E,I and EV3). Instead, the OMM of C10 S59L mitochondria were more frequently ruptured and partially or fully enclosed by double-membraned autophagophores or autophagosomes, respectively, as has also been

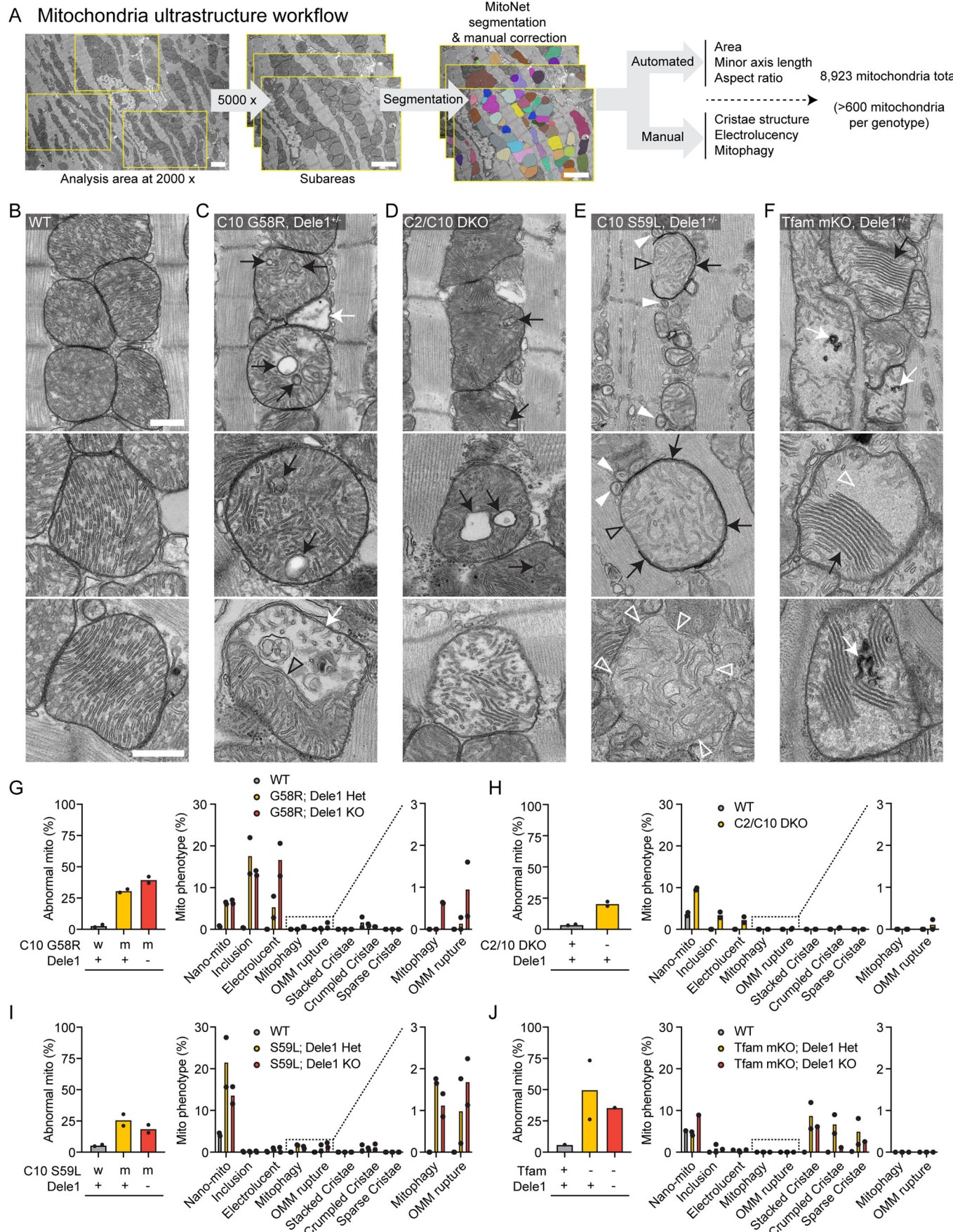

**Figure 3. Mitochondrial stressors alter the ultrastructure of heart mitochondria in characteristic ways, most of which are not suppressed by the DELE1 mt-ISR.**

(A) Workflow for analysis of mitochondria from hearts of diverse myopathy/cardiomyopathy models, combining deep learning aided segmentation of TEM images with manual scoring of characteristic ultrastructural features. Scale bars = 2 μm. (B) Representative TEM images of mitochondria from wild-type hearts. A field of view containing several mitochondria (top) and higher magnification views of individual mitochondria (below). Scale bars = 500 nm. (C) Representative TEM images of mitochondria in C10 G58R hearts which exhibited cristal inclusions (black arrows) and electrolucent mitochondria (white arrow in top and bottom images). The bottom image shows a segmented type of electrolucent mitochondria with a cut-through cristae (open black arrowhead) forming the boundary between the electrolucent portion of the mitochondria, and that with a more typical appearance. Scale bars = 500 nm. (D) Representative TEM images of mitochondria in the C2/C10 DKO hearts which were characterized by cristal inclusions (black arrows) and the presence of electrolucent mitochondria (bottom). Scale bars = 500 nm. (E) Representative TEM images of mitochondria in C10 S59L hearts showed nano-sized mitochondria defined as measuring <250 nm diameter in their minor axis with an aspect ratio <1.5 (white arrowheads), mitochondria partially enclosed in electron-dense phagophore membranes (black arrows) with a portion of their periphery not enclosed (open black arrowheads), and mitochondria with a ruptured OMM (bottom image). Open white arrowheads indicate sites where the single layer of IMM is visible and OMM is absent. Scale bars = 500 nm. (F) Representative images of mitochondria in *Tfam* mKO hearts that displayed closely aligned stacked cristae (black arrows) and crumpled cristae (white arrows), as well as areas of where cristae were sparse, and a homogeneous light gray matrix material was present (open white arrowheads). Scale bars = 500 nm. The mitochondrion in middle image is cropped from same cell as the mitochondrion shown in (Fig. EV4E, bottom left). (G) Quantification of ultrastructural features for C10 G58R; *Dele1* KO at P28 and indicated littermates. (H) Quantification of ultrastructural features for C2/C10 DKO animals and unrelated age-matched controls at 3–4 months old. (I) Quantification of ultrastructural features for C10 S59L; *Dele1* KO at P140 and indicated littermates. (J) Quantification of ultrastructural features for *Tfam* mKO; *Dele1* KO at P56 and indicated littermates. Source data are available online for this figure.

reported previously by others (Genin et al, 2019) (Figs. 3E, I and EV3E,F). C10 S59L also had the highest frequency of nano-mitochondria (22%) in thin sections among the models (Fig. 3I). Tracking nano-mitochondria through several 60-nm-thick serial sections showed that some were indeed spherical, while others were short tubules (Fig. EV3G,H), but most were nano-tubes emerging from larger mitochondria and ending in a dead-end or connecting to other mitochondria. Similar numbers of OMM rupture and mitophagy events were observed in the presence or absence of *Dele1*, whereas fewer nano-mitochondria were observed in the absence of *Dele1* (Fig. 3I). The average area of mitochondria was similar in the presence and absence of *Dele1* (Fig. EV1A). Thus, the DELE1 mt-ISR did not prevent the mitochondrial morphological defects caused by C10 S59L protein misfolding stress.

Mitochondria in adult *Tfam* mKO hearts also had a distinct ultrastructure, compared to the other three models, suggesting loss of mtDNA exerts a different stress on mitochondria than C10 protein misfolding or C2/C10 DKO (Figs. 3F,J and EV4). In *Tfam* mKO, mitochondrial cristae often appeared to adhere together forming patches of typically five or more "stacked" cristae, with little intralumenal space. Often, in the same mitochondrion, large regions of homogenous light gray matrix material were devoid of cristae, in a "sparse cristae" phenotype. Cristae that were crumpled into electron dense swirls were also observed, as well as cristae that appeared tubular, with increased diameter (Figs. 3F and EV4E). The extent of these phenotypes was highly variable among cardiomyocytes in the same sample, with mildly affected cardiomyocytes often bordering severely affected cardiomyocytes in a mosaic pattern. This mirrored the cellular heterogeneity observed by COX and SDH histochemistries and TEM in prior reports (Wang et al, 1999) (Fig. EV4B). Notably, these mitochondrial phenotypes were milder in a *Tfam* mKO; *Dele1* KO heart sample compared to two *Tfam* mKO; *Dele1*+ hearts, suggesting that the DELE1 mt-ISR likely does not protect against these mitochondrial defects (Fig. 3J).

Considered together, mitochondrial ultrastructure was distinct among the four models, although each resulted in disruption of the IMM. The DELE1 mt-ISR did not reverse the underlying mitochondrial phenotypes, excepting the electrolucent mitochondria in the C10 G58R model. Thus, the DELE1 mt-ISR does not mitigate most structural abnormalities caused by mitochondrial stress.

## The mt-ISR mediates a core transcriptional response to diverse mitochondrial stressors in striated muscle

To obtain a global view of the effect of the DELE1 mt-ISR on striated muscle, we performed transcriptomics in the hearts of three models that survived past weaning: C10 G58R, C10 S59L, and *Tfam* mKO (Fig. 4A–D and Dataset EV1). To identify those differentially expressed genes that are DELE1-dependent, we examined the genes that (1) significantly changed with stress on a Dele1+ background (group 3: Stress+Dele1+ vs. group 1: Stress-Dele1-), and (2) significantly reverted toward control levels (group 1: Stress-Dele1+) in the group 4 (Stress+; Dele1 KO) vs. group 3 (Stress+; Dele1+) comparison. Despite the diversity of mitochondrial stress in these models, they shared 111 mito-stress DEGs in the heart (Appendix Fig. S6B), of which 51 (46%) were DELE1-dependent (hereafter, referred to as the DELE1 mt-ISR heart signature) (Fig. 4A,B). An additional 39 DELE1-dependent DEGs were shared in 2 of 3 models (Fig. 4C; Appendix Fig. S6A). Considering each model individually, the percent DELE1 dependence did not differ significantly among the models and a similar proportion of DEGs were DELE1-dependent (18–33%), indicating that the DELE1 mt-ISR accounts for a substantial portion of the overall transcriptional response to mitochondrial stress in each of the three models (Appendix Fig. S6C,D). Considered together, these results indicate that a common DELE1-dependent mechanism mediates signaling in response to diverse stressors such as mitochondrial protein unfolding and decreased mtDNA maintenance.

Overall, the core DELE1 mt-ISR had several recognizable components (Fig. 4B), including: (1) transcriptional regulation; (2) amino acid transport and biosynthesis (particularly, for serine, glycine, asparagine, and proline) (13/51, 25% of genes); (3) protein translation, including amino acid tRNA synthetases, tRNA export (by *Xpot*), and protein synthesis re-initiation (by *Eif3c*) (12/51, 24% of genes) (Guan et al, 2017); and (4) mitochondrial 1C metabolism (which also intersects with serine and glycine metabolism and, through NADPH production, proline metabolism) (Ducker and Rabinowitz, 2017; Ducker et al, 2016) (Fig. 5C). Strikingly, nearly half of the genes in the core DELE1 mt-ISR signature were directed at promoting protein synthesis, as discussed further below. These genes were significantly enriched for those with the ATF4 enhancer element (consensus sequence, RNMTGATGCAAY), and 40/51

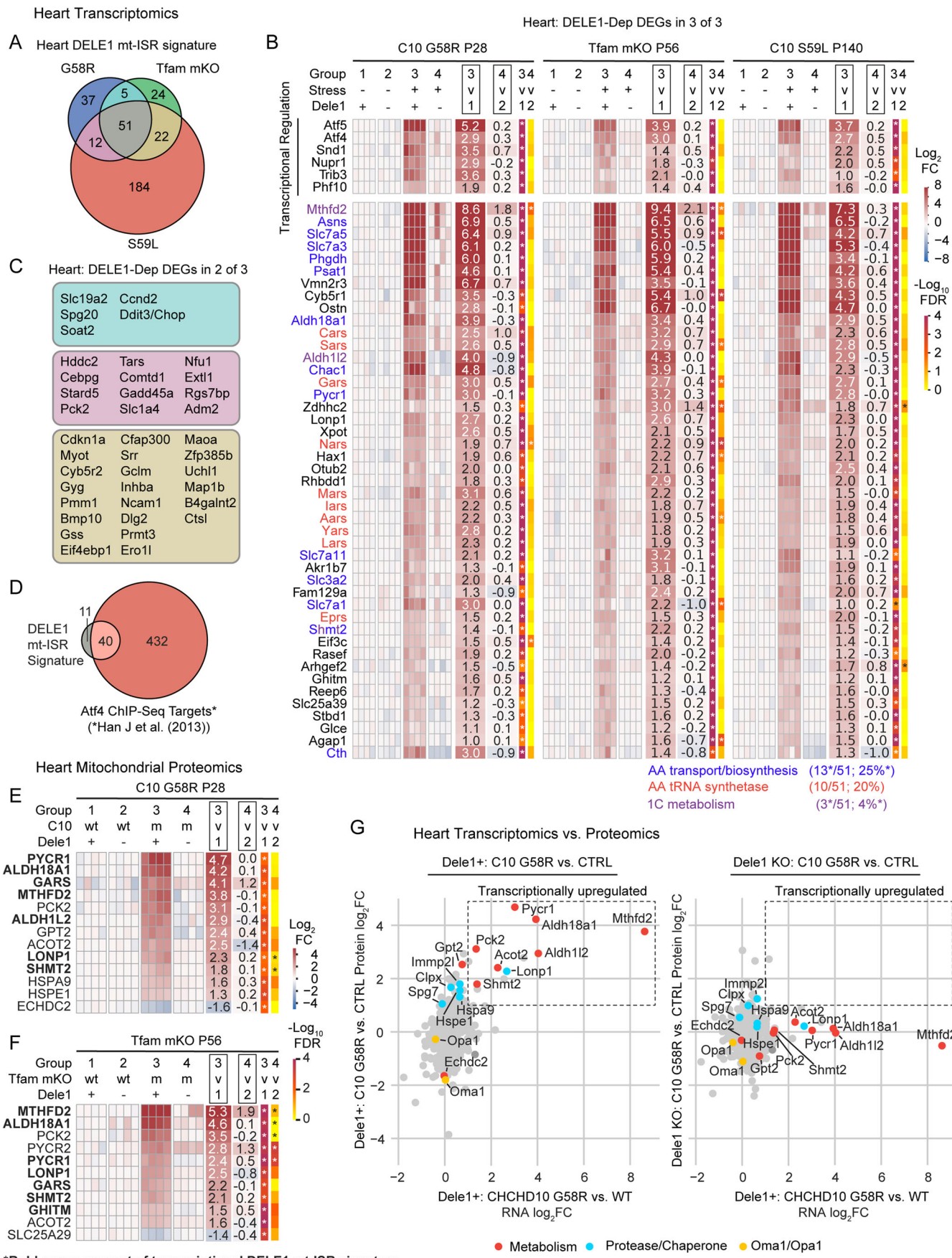

Heart Transcriptomics

**A** Heart DELE1 mt-ISR signature

**B** Heart: DELE1-Dep DEGs in 3 of 3

**C** Heart: DELE1-Dep DEGs in 2 of 3

**D** DELE1 mt-ISR Signature — Atf4 ChIP-Seq Targets* (*Han J et al. (2013))

AA transport/biosynthesis (13*/51; 25%*)
AA tRNA synthetase (10/51; 20%)
1C metabolism (3*/51; 4%*)

Heart Mitochondrial Proteomics

**E** C10 G58R P28

**F** Tfam mKO P56

*Bold genes are part of transcriptional DELE1 mt-ISR signature

**G** Heart Transcriptomics vs. Proteomics

Dele1+: C10 G58R vs. CTRL

Dele1 KO: C10 G58R vs. CTRL

● Metabolism  ● Protease/Chaperone  ● Oma1/Opa1

**Figure 4. DELE1 drives a stereotyped transcriptional response to diverse mitochondrial stressors and reshapes the mitochondrial proteome in the heart.**

(A) Venn diagram depicting intersection of 51 DELE1-dependent DEGs common to all three MM/CM models (the DELE1 mt-ISR heart signature), identified in microarray data. Data from the C10 G58R; *Dele1* KO dataset also appears in Appendix Fig. S1B, Figs. 2E and 6C. (B) Heat map depicting fold change for DELE1-dependent DEGs identified in all three models (left). Those annotated as regulating transcription in Lambert et al, (2018) are separated from other genes. *Shmt2* is in both the "AA transport/biosynthesis" and the "1C metabolism" categories. Reported FDR was calculated in TAC v4.0.3 software across all genes in dataset, using the default settings, which uses a one-way ANOVA corrected with the Benjamini-Hochberg procedure for multiple procedures. (C) DELE1-dependent DEGs common to two of three myopathy/cardiomyopathy models (color coded as in (A)). (D) Venn diagram shows overlap between DELE1 mt-ISR heart signature and previously identified ATF4 targets in mouse cells, using ATF4 ChIP-Seq from (Han et al, 2013). (E) Heat map showing significant DELE1-dependent changes in mitochondrial protein abundance in C10 G58R hearts. $N = 4$ mice per condition. Statistics for proteomics data (which also appear in Dataset EV3) were performed as described in the Methods with correction for multiple comparisons across all protein groups in the dataset. (F) Heat map showing significant DELE1-dependent changes in mitochondrial protein abundance in *Tfam* mKO hearts. $N = 4$ mice per condition, except for C10 S59L group2 which was $N = 3$ mice. Statistics for proteomics data (which also appear in Dataset EV3) were performed as described in the Methods with correction for multiple comparisons across all protein groups in the dataset. (G) Scatterplots compares RNA $log_2$FC for C10 G58 vs. control (in the presence of *Dele1*) and mitochondrial protein $log_2$FC for C10 G58R vs. control animals in the presence of *Dele1* (left) or the absence of *Dele1* (right). Source data are available online for this figure.

(78%) overlapped with ATF4 targets (Fig. 4D; Appendix Fig. S6E and Dataset EV2), which were identified in a previously published ATF4 ChIP-Seq dataset from mouse embryonic fibroblasts (Han et al, 2013).

These genes also overlapped extensively with those previously described as part of mitochondrial stress responses in MM, including *Atf4*, *Atf5*, *Mthfd2*, *Asns*, *Psat1*, *Phgdh*, and *Lonp1* (Fig. 4B) (Anderson et al, 2019; Tyynismaa et al, 2010; Dogan et al, 2014; Kühl et al, 2017; Nikkanen et al, 2016). Others previously associated with mitochondrial stress responses were either inconsistently elevated by mitochondrial stress or inconsistently DELE1-dependent. *Ddit3* (also known as, *Chop*), *Gdf15*, and *myc*, for instance, were strongly DELE1-dependent in some but not other models. The mitokine *Fgf21* appeared to be DELE1-dependent in all models but only increased > 2-fold in C10 G58R hearts and so was not included among common DELE1-dependent DEGs in the heart. *Gpx4*, which was found to be regulated by the OMA1-DELE1 pathway in Cox10 cKO hearts (Ahola et al, 2022), was elevated by about 50% at the mRNA and protein levels across models tested but was not consistently DELE1-dependent (reaching nominal significance only in mitochondrial proteomics data from C10 G58R) (Appendix Fig. S6F–H). Thus, DELE1 accounts for upregulation of many but not all classic markers of mitochondrial stress in mouse striated muscle, in response to diverse mitochondrial stressors.

Notably, a substantial proportion of the core DELE1 mt-ISR genes encode mitochondrial proteins (11/51, 21.6%). These were enriched for genes involved in metabolism (proline synthesis, serine/glycine biosynthesis, and 1C metabolism), in addition to three involved in quality control: *Lonp1*, encoding a matrix quality control AAA+ protease; *Ghitm* (also known as *Tmbib5*), recently identified to encode a $H^+/Ca^{2+}$ antiporter (Austin et al, 2022; Zhang et al, 2022; Patron et al, 2022); and *Slc25a39*, which was recently identified to encode a glutathione transporter (Wang et al, 2021; Shi et al, 2022). We compared this set to DELE1-dependent proteins in mitochondrial proteomes from C10 G58R and *Tfam* mKO mice (Appendix Fig. S7A and Dataset EV3). Notably, 6 of 7 detected mitochondrial proteins in this group were found to significantly increase with stress and were DELE1-dependent at the protein level (Fig. 4E,F, proteins overlapping with Common DELE1 DEGs in bold). The exception was AKR1B7, which has multiple cellular locations in addition to mitochondria. Thus, DELE1-dependent changes in gene expression at the transcript level are largely

reflected in the mitochondrial proteome in diverse models of mitochondrial stress.

In contrast to these DELE1 mt-ISR-regulated genes, total mitochondrial protein levels of OPA1 and OMA1 were differentially regulated in the C10 G58R hearts and *Tfam* mKO hearts. Whereas OMA1 levels were decreased in C10 G58R hearts, consistent with its self-cleavage on activation (Fig. 4G), OMA1 levels were higher in *Tfam* mKO hearts than controls (Appendix Fig. S7B,C). These results suggest that in the context of chronic stress, steady-state levels of OMA1 and OPA1 are not reliable biomarkers for the OMA1-DELE1 mt-ISR activation, in contrast to ratio of OPA1 cleavage products (Fig. 1M; Appendix Fig. S2), and the upregulation of DELE1-dependent genes (Fig. 4B).

The correspondence between transcriptional and protein level regulation for these core DELE1 mt-ISR genes prompted us to consider the inverse scenario: is the mitochondrial proteome transcriptionally regulated by stress independently of the DELE1 mt-ISR? Strikingly, we found that DELE1 accounted for all transcriptionally driven increases in the mitochondrial proteome of greater than 2-fold in C10 G58R hearts compared to control (Fig. 4G)). Similar results were obtained in *Tfam* mKO hearts, in which DELE1 accounted for all the transcriptionally driven increases in the mitochondrial proteome, with the exceptions of TIMM10, PRDX6, COMTD1, and MTHFD2 (Appendix Fig. S7B,C). Notably, except for TIMM10 (in *Tfam* mKO) and IMMP2L (in C10 G58R), all increases in proteases or chaperones were either DELE1-dependent (as in the case of LONP1, HSPA9, and HSPE1) or were likely post-transcriptional (Appendix Fig. S7D,E). This suggests that the DELE1 mt-ISR is the principal mito-nuclear signal mediating the mito-UPR in striated muscle.

## The DELE1 mt-ISR upregulates anabolic pathways that run through the mitochondrial matrix

Having defined that the DELE1 mt-ISR is activated in several preclinical models of mitochondrial disease that exhibit progressive and chronic pathology, we next investigated its tissue-specific modulation during acute metabolic stress. Many of the amino acid biosynthesis pathways upregulated as part of the core DELE1 mt-ISR pass through the mitochondria, including those involved in asparagine synthesis (from the TCA intermediate oxaloacetate via aspartate), proline synthesis, and glycine synthesis (generated in the mitochondrial matrix from serine through the mitochondrial 1C

 

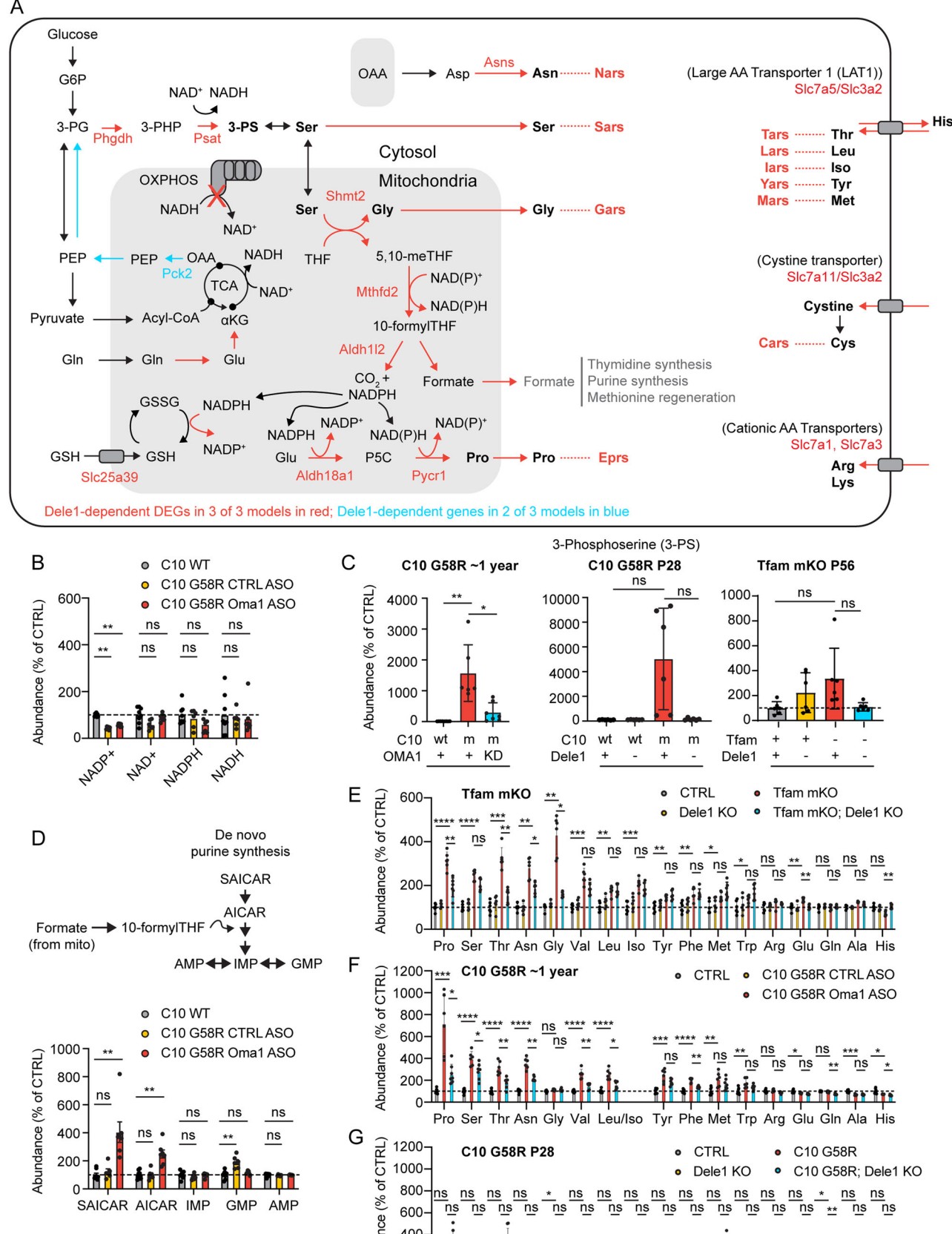

**Figure 5.  DELE1 mt-ISR maintains anabolic mitochondrial pathways, including for protein synthesis intermediates.**

(A) Diagram depicts major intersections among 22 (out of 51) pro-anabolic genes that are upregulated as part of the DELE1 mt-ISR heart signature (gene names in red). Additional genes (in blue) were identified in 2/3 models. Key pathways intersecting mitochondria include those for the biosynthesis of glycine, serine, proline, and asparagine. Upregulation of these pathways is coordinated with upregulation of genes for the corresponding aminoacyl tRNA synthases. (B) Individual data for levels of $NAD^+$, NADH, $NADP^+$, and NADPH detected by untargeted metabolomics of heart tissue from ~1-year-old (319–411 days) C10 G58R mice injected with CTRL or *Oma1* ASOs or wild-type littermates injected with PBS for 6 weeks prior to sacrifice. Data is from the named dataset that also appears in Dataset EV4. Statistical analysis for the metabolomics dataset is described in Methods. Significance indicated on graph, tested using two-sided Student's t-tests, was corrected for multiple comparisons across all metabolites in the dataset, using the Benjamini–Hockberg procedure. Error bars indicate the SD. From left to right, adjusted $p$-values = 2.58E−07, 0.04948, 0.6861, 0.8274 (bottom row) and 1.08E−06, 0.5264, 0.1282, 0.7702 (top row). ** indicates $p \leq 0.01$ and "ns" not significant. (C) Levels of 3-phosphoserine measured by untargeted proteomics in adult C10 G58R mice as in (B) or by targeted metabolomics in P28 C10 G58R mice and P56 *Tfam* mKO mice. W denotes wild type; m denotes mutant. Data is from the named dataset that also appears in Dataset EV4 (for C10 G58R ~ 1 year) and Dataset EV5 (for P28 C10 G58R and Tfam mKO). Statistical analysis for the metabolomics dataset is described in Methods. Significance indicated on graph, tested using two-sided Student's t-tests, was corrected for multiple comparisons across all metabolites in the dataset, using the Benjamini–Hockberg procedure. Error bars indicate the SD. (In left graph) adjusted $p$-values = 0.001630 and 0.05766 (left to right); (in middle graph) adjusted $p$-values = 0.3344 and 0.2107 (left to right); and (in right graph) adjusted $p$-values = 0.1039 and 0.2845 (left to right). *, ** indicates $p \leq 0.05$ and $p \leq 0.01$, respectively, and "ns" not significant. $N \geq 6$ mice per group (genotype). (D) Levels of intermediates of de novo purine synthesis that are sensitive to blocks in the mitochondrial 1C metabolism pathway measured from hearts of ~1-year-old C10 G58R by untargeted metabolomics as in (B). Data is from the unnamed dataset that also appears in Dataset EV4 (for C10 G58R ~ 1 year). Statistical analysis for the metabolomics dataset is described in Methods. Significance indicated on graph, tested using two-sided Student's t-tests, was corrected for multiple comparisons across all metabolites in the dataset, using the Benjamini–Hockberg procedure. Adjusted $p$-values (for metabolites left to right) for C10 WT vs. C10 G58R; CTRL ASO are 0.9709, 0.5572, 0.4919, 0.004337, and 0.6930. Adjusted $p$-values (for metabolites left to right) for C10 WT vs. C10 G58R; OMA1 ASO are 0.003006, 0.007981, 0.8724, 0.3951, and 0.8988. Error bars indicate the SD. ** indicates $p \leq 0.01$ and "ns" not significant. $N \geq 6$ mice per group (genotype). (E–G) Levels of amino acids measured from hearts of P56 *Tfam* mKO, ~1-year-old C10 G58R, or P28 C10 G58R mice as in (C). Data is from the unnamed dataset that also appears in Dataset EV4 (for C10 G58R ~ 1 year) and the named datasets in Dataset EV5 (for P28 C10 G58R and *Tfam* mKO). Statistical analysis for the metabolomics dataset is described in Methods. Significance indicated on graph, tested using two-sided Student's t-tests, was corrected for multiple comparisons across all metabolites in the dataset, using the Benjamini–Hockberg procedure. In (E), adjusted $p$-values for CTRL vs. *Tfam* mKO; *Dele1* KO comparison (for metabolites listed left to right) were 2.67E−05, 9.30E−06, 0.0001437, 0.001376, 0.0002088, 0.001959, 0.0002551, 0.007751, 0.005143, 0.01233, 0.03223, 0.08644, 0.001959, 0.5636, 0.0002551, 0.1087. In (E), adjusted $p$-values for *Tfam* mKO vs. *Tfam* mKO; *Dele1* KO comparison (for metabolites listed left to right) were 0.003740, 0.0006115, 0.01132, 0.001234, 0.003740, 0.001615, 0.001740, 0.001615, 0.004377, 0.003740, 0.004263, 0.004849, 0.06566, 0.2845, 0.1847, 0.1052, 0.1515. In (F) adjusted $p$-values for CTRL vs. C10 G58R; CTRL ASO comparison (for metabolites listed left to right) were 0.0006519, 3.58E−06, 4.57E−05, 2.71E−06, 0.04601, 3.55E−05, 6.62E−05, 0.0005232, 3.16E−05, 0.007724, 0.01307, 0.5611, 0.06687, 0.2196, 0.001016, 0.04463. In (F), adjusted $p$-values for C10 G58R; CTRL ASO vs. C10 G58R; *Oma1* ASO KO comparison (for metabolites listed left to right) were 0.004201, 0.0005487, 0.01111, 1.47E−05, 0.1337, 0.005503, 0.007491, 0.001017, 0.001807, 0.03986, 0.08713, 0.09570, 0.005235, 5.26E−05, 0.0009977, 0.009547. In (G), adjusted $p$-values for CTRL vs. C10 G58R *Dele1* KO comparison (for metabolites listed left to right) were 0.1762, 0.06858, 0.1762, 0.06858, 0.2655, 0.2837, 0.2655, 0.2655, 0.1630, 0.1242, 0.1866, 0.1418, 0.7218, 0.6800, 0.2655, 0.1762, 0.8507. In (G), adjusted $p$-values for C10 G58R vs. C10 G58R; *Dele1* KO comparison (for metabolites listed left to right) were 0.2684, 0.1241, 0.3593, 0.1500, 0.2196, 0.9517, 0.8701, 0.6962, 0.1512, 0.2196, 0.2196, 0.2196, 0.2196, 0.7206, 0.008545, 0.3180, 0.1500. In all graphs, *, **, ***, **** indicates $p \leq 0.05$, 0.01, 0.001, 0.0001, respectively, and "ns" not significant. $N \geq 6$ mice per group (genotype). Error bars represent SD. Source data are available online for this figure.

---

metabolism pathway) (Fig. 5A, core genes in red). Flux through these interconnected pathways may be altered by OXPHOS dysfunction due, in part, to reductive stress in the mitochondrial matrix, as CI (and thus OXPHOS as a whole) is responsible for most of the NADH oxidation in the mitochondrial matrix. Consistent with reductive stress at the tissue level in adult C10 G58R hearts, $NADP^+$ levels were significantly decreased and $NAD^+$ levels trended toward decreased, in an untargeted metabolomic experiment (Fig. 5B and Dataset EV4).

We considered whether the OMA1-DELE1 mt-ISR may compensate for mitochondrial stress by upregulation of the key enzymes identified by transcriptomics and mitochondrial proteomics. To assess this in adult C10 G58R hearts, we used *Oma1* ASO knockdown strategy we employed previously (Shammas et al, 2022), as *Dele1* KO and *Oma1* KO cause early death in this model. Among the named metabolites in the untargeted metabolomics dataset, 3-phosphoserine (3-PS) had the greatest fold-change, increasing >100 fold in the hearts of C10 G58R mice compared to control mice (Fig. 5C; Appendix Fig. S8A and Dataset EV4). This increase was blocked by *Oma1* KD, indicating it was dependent on the OMA1-DELE1 mt-ISR. This same pattern was also seen by targeted steady-state metabolomics in the hearts of P28 C10 G58R mice, and, to a lesser degree, in P56 *Tfam* mKO mice, although it did not reach significance in these models after correction for multiple comparisons (Fig. 5C; Appendix Fig. S8B,C and Dataset EV5).

Notably, 3-PS is produced by two enzymes, PHDGH and PSAT1, the gene expression of which was strongly upregulated by the DELE1 mt-ISR in all models (Fig. 4B). These enzymes lie at a branch point in glycolysis, in which glucose-derived carbons are directed to pyruvate, the TCA cycle, and ultimately OXPHOS or toward the biosynthesis of serine and its derivates, such as glycine and 1C metabolism intermediates. The upregulation of PHDGH and PSAT1 and consequent increase in 3-PS are consistent with the hypothesis that a principal outcome of the OMA1-DELE1 mt-ISR is to shunt carbon from the glycolysis intermediate 3-PG toward biosynthesis.

Another potential carbon source for 3-PS synthesis is glutamine through the TCA cycle and gluconeogenesis. It is notable that two mitochondrial matrix enzymes, GPT2 (promoting glutamine entry into the TCA cycle via glutamate and oxaloacetate) and PCK2 (promoting gluconeogenesis from the TCA intermediate oxaloacetate) were increased by the DELE1 mt-ISR in C10 G58R hearts, and PCK2 was also increased in *Tfam* mKO hearts (Fig. 4C,E,F). Thus, the DELE1 mt-ISR may additionally push glutamine toward serine biosynthesis, as has been observed in rapidly dividing cancer cells with high biosynthetic needs (Vincent et al, 2015).

3-PS is the precursor for serine biosynthesis (Fig. 5A). Serine, in turn, is used for glycine biosynthesis and is the chief 1C donor for 1C metabolism (Ducker and Rabinowitz, 2017). Under basal conditions, most of the 1C metabolism flux from serine goes through the mitochondrial 1C metabolism pathway, with

mitochondria exporting formate used for de novo purine and thymidine biosynthesis and the regeneration of methionine (after additional transformations) (Ducker et al, 2016). Mitochondrial 1C metabolism is also critical for generating the mitochondrial NAPDH pool needed for glutathione regeneration and proline synthesis, through the DELE1-responsive enzyme ALDH18A1. We, therefore, asked whether the OMA1-DELE1 mt-ISR helps maintain the mitochondrial 1C metabolism pathway in the setting of mitochondrial stress. To do so, we evaluated the levels of two metabolites AICAR and S-AICAR, precursors in purine synthesis, that are known to increase following blocks in mitochondrial 1C flux (particularly at MTHFD2) (Ducker et al, 2016). Consistent with the OMA1-DELE1 mt-ISR maintaining mitochondrial 1C flux in the setting mitochondrial stress, AICAR and S-AICAR levels remained baseline in C10 G58R mice but increased significantly when the OMA1-DELE1 mt-ISR was silenced, suggesting a partial block in the pathway in the absence of the mt-ISR (Fig. 5D). Thus, likely by upregulating mitochondrial 1C metabolism enzymes such as MTHFD2 and SHMT2 (and the upstream serine biosynthesis pathway), the OMA1-DELE1 mt-ISR may circumvent a block in the mitochondrial 1C metabolism pathway in the setting of mitochondrial stress.

In addition to serine and glycine biosynthesis, the DELE1 mt-ISR upregulates several amino acid biosynthesis and uptake pathways, including proline synthesis and asparagine biosynthesis, which connect directly with mitochondrial metabolism (Fig. 5A). We next evaluated whether these DELE1-dependent gene expression changes resulted in a steady-state change in amino acid levels in the hearts of adult C10 G58R (from untargeted metabolomics) and *Tfam* mKO mice (from targeted metabolomics). In both models, "aminoacyl-tRNA biosynthesis" was the most enriched KEGG metabolite set among mt-ISR-dependent metabolites (Appendix Fig. S8A,B). DELE1-dependent amino acids included proline, asparagine, and threonine in both models, and glycine, serine, leucine, phenylalanine, and glutamate in at least one of the models (Fig. 5E,F; Appendix Fig. S8A,B). These DELE1-dependent amino acids all increased with stress.

We also examined amino acid levels in juvenile (P28) C10 G58R mice with one or no copies of *Dele1* (Fig. 5G; Appendix Fig. S8C). In contrast to the adult C10 G58R mice, there were no significant changes in the abundance of steady-state amino acids levels, except for glycine (which significantly increased with stress) and glutamine (which decreased with stress). The change in glutamine was significantly DELE1-dependent, whereas the change in glycine trended in the direction of DELE1 dependence but did not reach significance after correcting for multiple comparisons. The relatively small effect on the steady-state metabolome in the rapidly growing animal could be due to increased utilization of amino acids, particularly if amino acid abundance is one of the factors limiting growth in the C10 G58R animals. Interestingly, the C10 G58R; *Dele1* KO animals also showed high variability in steady-state tissue amino acid levels, which may reflect loss of homeostasis in animals that are at or nearing the end-stage of their life.

Considered together, these data suggest that the OMA1-DELE1 mt-ISR may upregulate amino acid levels in the adult heart under mitochondrial stress, including four that are synthesized through pathways that are closely connected to mitochondrial metabolism: proline, asparagine, serine, and glycine. We speculate that the steady-state increase of these four amino acids and 3-PS is driven by the DELE1 mt-ISR-dependent upregulation of the enzymes responsible

for their biosynthesis, which we found were also upregulated in a DELE1-dependent manner at the transcript and/or protein level. However, as flux through many different pathways affect levels of 3-PS and these amino acids, this attribution remains speculative. That the upregulation of these amino acids is coordinated with the upregulation of their corresponding amino acid tRNA synthetase (i.e., proline with *Eprs*, asparagine with *Nars*, serine with *Sars*, glycine with *Gars*) additionally raises the possibility that they may be upregulated to maintain adequate levels of tRNA-charged amino acids for protein synthesis (Figs. 4D and 5A).

## DELE1 mediates the mt-ISR in mitochondrially enriched tissues, including brown adipose tissue and skeletal muscle

Having defined that the DELE1 mt-ISR is activated in several preclinical models of mitochondrial disease that exhibit progressive and chronic pathology, we next investigated its tissue-specific modulation during acute metabolic stress (Fig. 6A and Dataset EV6). Cold stress has been shown to activate OMA1 and, in separate reports, to activate the ISR (Quirós et al, 2012; Jena et al, 2023; Flicker et al, 2019; Levy et al, 2023). We, therefore, hypothesized that DELE1 may mediate the ISR in BAT under cold stress.

In BAT, a 9-h cold stress successfully decreased lipid stores (Appendix Fig. S9A,B) and activated OMA1 to cleave L-OPA1 (Fig. 6A; Appendix Fig. S9C). Examining the transcriptional response, no difference was detected between *Dele1* KO and WT in mice kept at room temperature (Appendix Fig. S9D). Following cold stress, however, 11 DELE1-dependent DEGs were detected—*Mthfd2*, *Cth*, *Fgf21*, *Psat1*, *Chac1*, *Slc7a1*, *Slc7a5*, *Trib3*, *Cars*, *Tars*, and *Phf10*—demonstrating that DELE1 activates the ISR in BAT in response to physiologic cold stress (Fig. 6B; Appendix Fig. S9E). Upregulation of these DELE1-dependent genes was similarly dependent on *Oma1* following cold stress (Appendix Fig. S9E). Notably, the DELE1-dependent mt-ISR in BAT involved a similar (albeit more limited) set of genes compared to what was observed in the heart under pathological mitochondrial stress.

We next asked whether the DELE1 mt-ISR identified in the heart and BAT is conserved across additional mitochondria-rich mouse tissues, under stress, including skeletal muscle and liver. In contrast to heart and BAT, the mt-ISR transcriptional response was surprisingly weak in the livers of C10 G58R mice, despite strong OMA1 activation (Shammas et al, 2022). No DELE1-dependent DEGs were detected in liver at P28 and only a small number of OMA1-dependent DEGs at ~1 year, following knockdown of *Oma1* with an ASO (Fig. 6C; Appendix Fig. S10A,B).

Altogether, two OMA1 or DELE1-dependent genes, *Fgf21* and *Psat1*, were shared among all tissues, suggesting that these may represent the most robust markers for the OMA1-DELE1 mt-ISR across mouse tissues (Fig. 6C). Of the four tissues, the heart and gastrocnemius shared most DELE1-dependent genes, demonstrating that the mito-ISR is most similar in heart and skeletal muscle (Appendix Fig. S11A–D).

To examine differences among skeletal muscle and heart muscle in greater detail, we next performed RNAseq on tissues from three striated muscles: gastrocnemius (Dataset EV7), heart (Dataset EV8), and tibialis anterior (Dataset EV9) (Appendix Figs. S12, S13, S14, S15A,B). RNAseq measures transcript abundance by sequencing and has a higher dynamic range than microarray, providing an

 

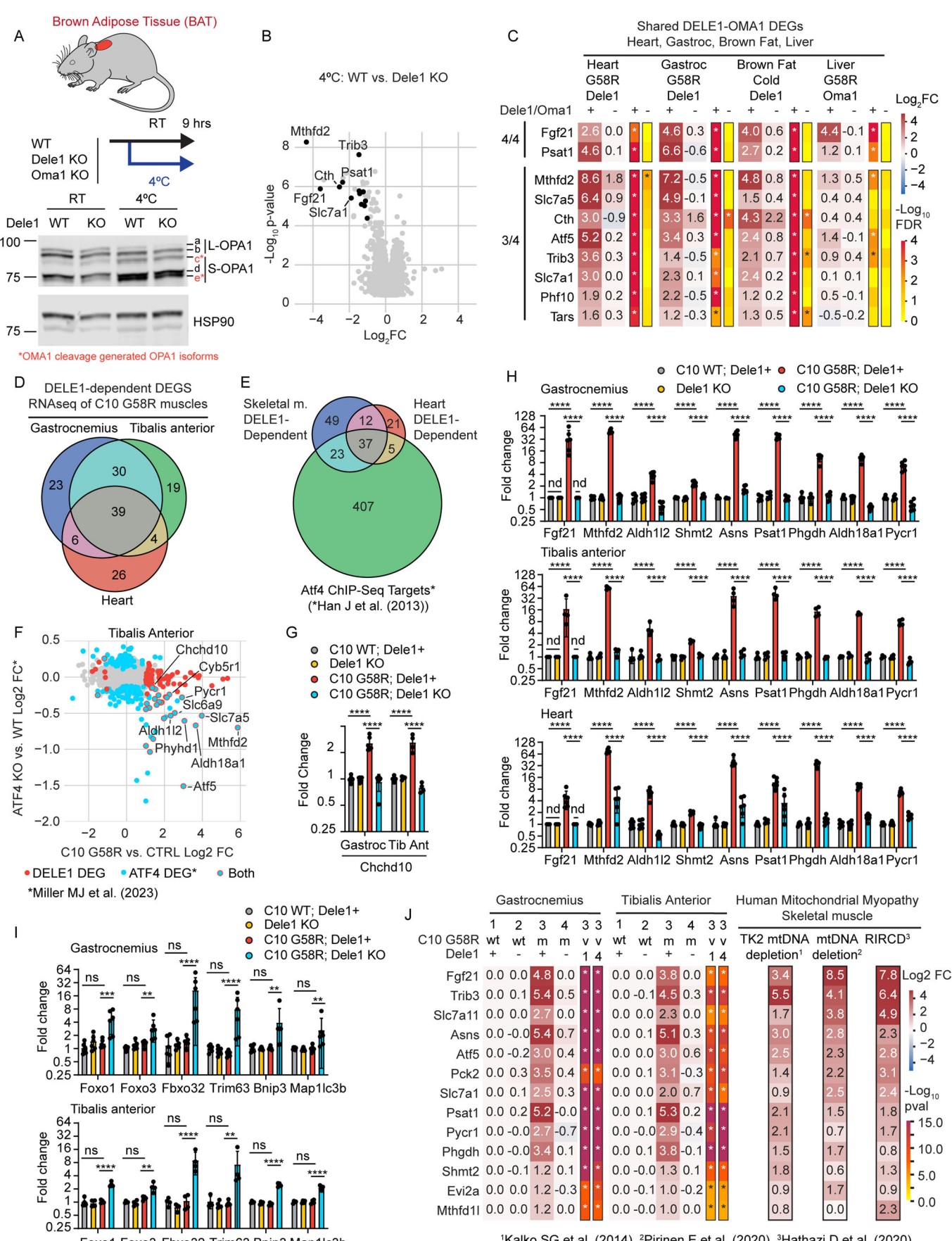

◄

**Figure 6. The DELE1 mt-ISR is observed in several mitochondria-rich tissues and has greatest overlap between heart and skeletal muscle.**

(A) *Dele1* KO and WT littermates were challenged with cold stress for 9 h and interscapular BAT was analyzed by immunoblotting for OPA1 cleavage by OMA1. OMA1 cleavage generated OPA1 isoforms are in indicated in red as c* and e*. (B) Volcano plot of global gene expression changes in cold stressed *Dele1* KO vs. WT littermates measured by microarray; significant genes (FDR < 0.05 and |Log$_2$FC| > 1) are in black. Statistics were performed as described in Methods for all microarray-based transcriptomics. $N = 7$ mice per group. (C) Heat map shows overlap among DELE1-dependent DEGS in four mitochondrial-rich tissues: heart, gastrocnemius (gastroc) skeletal muscle, and liver from C10 G58R mice, in addition to BAT from mice subjected to cold stress. BAT microarray data from same dataset that also appears in Appendix Fig. S1B. C10 G58R heart microarray data are from the same dataset that also appears in Appendix Fig. S1B, Fig. 2E, and Fig. 4B. Statistics on all microarray data (which also appear in Dataset EV6) were performed as described in Methods for all microarray-based transcriptomics. FDR values were corrected for multiple comparisons across all transcripts in dataset. $N = 4$ for all groups except for C10 G58R; Dele1+ liver, which was $N = 3$. (D) Venn diagram shows intersection of C10 G58R DELE1-dependent DEGs from three muscles (gastrocnemius, tibialis anterior, and heart), measured by RNA-Seq at P28 (data also available in Dataset EV7–EV9). (E) Venn diagram shows intersection of skeletal and and heart muscle DELE1-dependent DEGs as in (D) with ATF4 target genes, from previously published ATF4 ChIP-Seq dataset (Han et al, 2013). (F) Scatterplot shows correlation between genes that increase in tibialis anterior of P28 C10 G58R mice vs. CTRL and those that decrease in abundance in tibialis anterior from 6-month-old ATF4 skeletal muscle knockout mice vs. CTRL, from a previously published dataset (Miller et al, 2023). (G, H) Individual data is shown for select DELE1-dependent genes detected from skeletal muscle and/or heart from P28 C10 G58R mice, by RNA-Seq (data for all detected genes in Dataset EV7–EV9). Fgf21 was below detection limit (not detected, "nd") in all but the C10 G58R; Dele1+ condition. Significance was first tested with a two-way ANCOVA including genotype and sex as variables. For post hoc testing after ANCOVA (shown in Figure), the general linear hypotheses was used in conjunction with the multiple comparisons of means ("mcp") function to test for all possible two-way comparisons via "Tukey" method. For (G), p-values were 1.98E−08 and 6.98E−07 (for bottom row) and 7.12E−10 and 6.62E−08 (for top row). For (H, top graph, gastrocnemius) p-values for C10 WT; Dele1+ vs. C10 G58R; Dele1+ (for genes left to right) were 0, 0, 2.58E−09, 1.89E−15, 0, 0, 0, 0, and 1.11E−16, and for C10 G58R; Dele1+ vs. C10 G58R; *Dele1* KO (for genes left to right) were 0, 0, 1.19E−12, 2.50E−13, 0, 0, 0, and 0. For (H, middle graph, tibialis anterior) p-values for C10 WT; Dele1+ vs. C10 G58R; Dele1+ (for genes left to right) were 4.37E−07, 0, 4.63E−06, 7.99E−09, 4.97E−11, 0, 2.33E−15, 3.42E−13 and for C10 G58R; Dele1+ vs. C10 G58R; Dele1 KO (for genes left to right) were 8.51E−08, 0, 2.24E−05, 1.37E−08, 1.24E−12, 0, 2.22E−16, 7.77E−16, and 0. For (H, top graph, heart) p-values for C10 WT; Dele1+ vs. C10 G58R; Dele1+ (for genes left to right) were 1.57E−10, 0, 1.42E−11, 1.63E−12, 0, 2.71E−09, 0, 0, and 0, and for C10 G58R; Dele1+ vs. C10 G58R; Dele1 KO (for genes left to right) were 6.21E−10, 1.22E−12, 1.55E−15, 3.22E−15, 1.16E−13, 0.0001011, 0, 0, 0. In all graphs, **** indicates $p \le 0.0001$, respectively, and "ns" not significant. Error bars represent SD. $N \ge 4$ mice per group (genotype). (I) individual data is shown for select genes significantly elevated in skeletal muscles of only C10 G58R; *Dele1* KO and no other genotypes relative to control. Significance was first tested with a two-way ANCOVA including genotype and sex as variables. For post hoc testing after ANCOVA (shown in Figure), the general linear hypotheses was used in conjunction with the multiple comparisons of means ("mcp") function to test for all possible two-way comparisons via "Tukey" method. For gastrocnemius (top graph), p-values for C10 WT; Dele1+ vs. C10 G58R; Dele1+ (for genes left to right) were 0.7386, 0.3720, 0.8302, 0.8152, 0.9998, and 0.9999, and for C10 G58R; Dele1+ vs. C10 G58R; Dele1 KO (for genes left to right) were 0.0007080, 0.001020, 7.28E−05, 2.80E−05, 0.003794, 0.005845. For tibialis anterior (bottom graph), p-values for C10 WT; Dele1+ vs. C10 G58R; Dele1+ (for genes left to right) were 0.9919, 0.3088, 0.9999, 0.9984, 0.2705, 0.9920, and for C10 G58R; Dele1+ vs. C10 G58R; Dele1 KO (for genes left to right) were 3.52E−05, 0.002405, 1.66E−05, 0.001349, 1.16E−07, 1.30E−08. In all graphs, **, ***, **** indicates $p \le 0.01, 0.001, 0.0001$, respectively, and "ns" not significant. Error bars represent SD. $N \ge 4$ mice per group (genotype). (J) Heat map of 13 mouse genes that are DELE1-dependent in C10 G58R gastrocnemius and tibialis anterior model (as in D) and have human orthologs that are upregulated by >2-fold (on average) in three previously published datasets from mitochondrial myopathy patients (Hathazi et al, 2020; Pirinen et al, 2020; Kalko et al, 2014). P-values were corrected for multiple comparisons across all transcripts in dataset. Source data are available online for this figure.

orthogonal method for measuring global gene expression. We also included more samples per group for heart and gastrocnemius muscle to assess for potential sex differences among DELE1-dependent genes. As expected, gene expression fold-changes were highly correlated between the RNAseq and microarray data (r = 0.78 for C10 G58R; Dele1+ vs. CTRL comparison in heart), cross-validating results obtained on the two platforms (Appendix Fig. S12A). As seen in all other transcriptomic datasets above, there were no significant DEGs comparing Dele1 KO to control animals in any muscle tissue, excepting *Dele1* itself (Appendix Fig. S12B). Sex affected expression of some DELE1-dependent genes in striated muscle, however, DELE1-dependent genes were regulated by DELE1 in the same pattern for male and female mice (Appendix Figs. S13 and S14), demonstrating that the DELE1 mt-ISR is similar for both sexes. ANCOVA adjusted fold changes were used to correct for small but significant sex effects for some genes in the subsequent analysis.

Notably, the two skeletal muscles shared the majority of their DELE1-dependent DEGs (69/98 (70%) for gastrocnemius and 69/92 (75%) for tibialis anterior) (Fig. 6D; Appendix Fig. S15A,B). About half of these (39/69) were also shared with heart muscle and most were established ATF4 targets (Fig. 6E), based on comparison to a previously published ChIP-Seq dataset (Han et al, 2013). DELE1-dependent genes in the tibialis anterior additionally overlapped with ATF4-regulated genes detected from the same muscle in conditional *Atf4* KO mice (Fig. 6F), from a previously published dataset (Miller et al, 2023). This included *Chchd10*, suggesting that

*Chchd10* is a DELE1-dependent gene in skeletal muscle (Fig. 6F,G). Among the DELE1-dependent pathways common to skeletal and heart muscle were those identified to be shared among MM/CM models: mitochondrial 1C metabolism; serine, asparagine, and proline biosynthesis; amino acid transport across the plasma membrane; the tRNAs nuclear export protein XPOT; and cytosolic aaRSs (Fig. 6H; Appendix Fig. S15A). Notably, marker genes for both stage 1 (*Fgf21*, *Mthfd2*, and *Asns*) and stage 2 (*Psat1* and *Phghd*) mitochondrial stress responses identified previously by (Forsström et al, 2019) were elevated in the IMMD model by P28 and were DELE1-dependent. Together these data suggest that there are overlapping but distinct DELE1-dependent responses in skeletal muscle and heart muscle, involving genes targeted by ATF4 and pathways aimed at increasing precursors for protein synthesis.

In the absence of the mt-ISR, C10 G58R upregulated a separate gene expression program involving genes associated with hypoxia and semi-starvation-induced atrophy (De Theije et al, 2013). These included the transcriptional factors *Foxo1* and *Foxo3*, and atrogenes *Fbxo32* (also known as, *MAFbx*) and *Trim63* (also known as, *MuRF1*) encoding muscle-specific E3 ubiquitin ligases, and the autophagy-related genes *Bnip3* and *Map1l* (Fig. 6I). These data suggest that a catabolic program may be activated in skeletal muscle, following loss of the pro-anabolic transcriptional response mediated by the DELE1 mt-ISR. We speculate that both loss of the pro-anabolic DELE1 mt-ISR and activation of a catabolic in skeletal muscle may contribute to the growth failure in the IMMD early-onset MM model lacking the DELE1 mt-ISR.

We next asked whether the DELE1 mt-ISR signature in skeletal muscle is conserved to humans. We compared the DELE1-dependent genes detected in the two skeletal muscles from the IMMD mouse model with the transcriptional signature of muscle from MM patients from three previously published datasets (Hathazi et al, 2020; Pirinen et al, 2020; Kalko et al, 2014). The patients in these datasets were of different ages and had different causes of their MM, but 13 genes DELE1-dependent DEGs shared in the two mouse muscles were upregulated by at least 2-fold averaged across the three studies (Fig. 6J). These included genes involved in biosynthesis of serine (*PHGDH* and *PSAT1*), proline (*PYCR1*), and asparagine (*ASNS*), as well as gluconeogenesis (*PCK2*), which may facilitate additional carbon sources for serine biosynthesis, as discussed above. The signature additionally included the amino acid transporters, *SLC7A11* and *SLC7A1*, for cystine and positively charged amino acids, respectively, and the mitokine *FGF21*. 77% (10/13) of these genes are human ATF4 targets, with ATF4 ChIP-Seq peaks detected near the transcriptional start site in most human cell lines (Appendix Fig. S12C), based on a meta-analysis of human ATF4 ChIP-Seq data (Örd et al, 2023). Together our findings suggest that key components of the DELE1 mt-ISR promoting anabolism in the setting of mitochondrial stress are conserved to humans and upregulated by diverse causes of MMs. These may also serve as biomarkers to monitor the DELE1 mt-ISR in MM.

## The DELE1 mt-ISR maintains protein synthesis and proteostasis in striated muscle under mitochondrial stress

The findings so far suggest that the DELE1 mt-ISR upregulates pathways to maintain protein synthesis intermediates in the setting of mitochondrial stress. This would be predicted to promote net protein synthesis by facilitating translation elongation (Fig. 7A). However, acutely the ISR also limits the availability of the ternary complex required for translation initiation, which would be predicted to slow the net protein synthesis rate (Pakos-Zebrucka et al, 2016). While these two actions of the ISR may seem opposed, they have in common the promotion of well-regulated protein synthesis under metabolic stress. By simultaneously restricting the number of ribosomes that can initiate translation and increasing protein synthesis intermediates to maintain translation elongation, the mt-ISR may minimize ribosome stalling and/or mistranslation, thereby limiting translation-associated protein misfolding (Stein and Frydman, 2019; Stein et al, 2022). This suggests that the overall effect of the mt-ISR in mitochondrial myopathy may not be to slow net protein translation per se but to maintain the quality of protein translation in the setting of mitochondrial stress.

To directly assess the net effect of the DELE1 mt-ISR on protein synthesis in striated muscle, we performed the SUnSET assay in the heart and gastrocnemius muscle of the C10 G58R mice on the *Dele1*+ or *Dele1* KO backgrounds. Mice in each group were injected with puromycin, a tRNA aminoacyl analog that is incorporated into the elongating polypeptide, 30 min prior to sacrifice, and puromycylated peptides were detected by immunoblotting (Schmidt et al, 2009) (Fig. 7B,C). In C10 G58R mice, protein synthesis trended toward increased in the heart and decreased in gastrocnemius but only differed significantly from control mice in the hearts at day 21 (Fig. 7B,C). Importantly,

although *Dele1* KO normalized pS51-eIF2α in C10 G58R mice, as expected (Appendix Fig. S15C), it did not increase net protein synthesis in the heart or gastrocnemius (Fig. 7B,C). We also considered the ratio of protein synthesis in the gastrocnemius to the heart for individual mice, in order to control for sources of interindividual variability. Notably, the C10 G58R; *Dele1* KO had the lowest heart-to-gastrocnemius ratio among groups (Fig. 7C, bottom graph). Together these data suggest that the DELE1 mt-ISR does not have a net inhibitory effect on protein synthesis in stressed striated muscle, despite elevated pS51-eIF2α.

Having found the DELE1 mt-ISR did not decrease the net protein synthesis rate, we next considered its effect on proteostasis in skeletal muscle, using confocal microscopy. Notably, C10 G58R; *Dele1* KO but not C10 G58R; *Dele1*+ gastrocnemius muscle contained aggregates of ubiquitinated proteins that co-localized with p62/SQSTM1, a ubiquitin binding protein that sequesters misfolded proteins and can function as an autophagy adapter (Fig. EV5A,B). In some fibers from C10 G58R; *Dele1* KO, these aggregates were so numerous that they appeared confluent by diffraction-limited light microscopy, suggesting proteostatic collapse within the myofiber (Fig. 7D,E).

Newly translated proteins are particularly vulnerable to protein misfolding, and the rate of protein translation is controlled through regulation of translation initiation and varying of elongation rates along the transcript (e.g., through codon selection) to minimize protein misfolding. To determine if protein aggregation within myofibers is associated with changes in protein synthesis, we compared protein translation rates in individual muscle fibers with and without aggregates of ubiquitinated proteins, in gastrocnemius muscle for C10 G58R; Dele1 KO mice that were injected with puromycin (Fig. 7F–H). Myofibers with aggregates were again observed in C10 G58R; *Dele1* KO mice but not the other genotypes (Fig. EV5C). Notably, the muscle fibers with numerous aggregates had higher protein synthesis rates than other fibers in the same tissue (with 171.2% higher puromycin intensity on average), suggesting loss of translation control (Fig. 7G). These myofibers also had a significantly smaller cross-sectional area (CSA) (mean CSA: 330.0 vs. 446.9 μm$^2$), consistent with growth failure or atrophy (Figs. 7H and EV5D). Interestingly, individual aggregates also co-localized with puromycylated polypeptides, suggesting that the aggregates of ubiquitinated proteins may form near sites of active translation (Fig. 7F, arrows). The puromycin immunostaining was likely specific for puromycylated polypeptides, as no signal was present in a negative control C10 G58R; *Dele1* KO mouse not injected with puromycin (Fig. EV5C, right most panel). These findings suggest that the DELE1 mt-ISR is important for maintaining translation-associated proteostasis in striated muscle under mitochondrial stress.

Notably, muscle fibers of each major type (type 1, type 2a, type 2b, and type 2x), exhibited loss of proteostasis and diminished cross-sectional area, demonstrating that this phenotype was not restricted to a single fiber type (Appendix Fig. S16; Fig. 7I,J). More oxidative fibers (type 1 and type 2a) trended toward having a greater propensity to accumulate ubiquitin protein aggregates compared to the more glycolytic 2b fibers, reaching significance for the comparison between type 1 and type 2b (Fig. 7K). Fibers with aggregates also tended to cluster toward the anterior surface of the gastrocnemius, where the mixed fiber population was located (Appendix Fig. S16). Thus, all fiber types appeared to be affected,

   

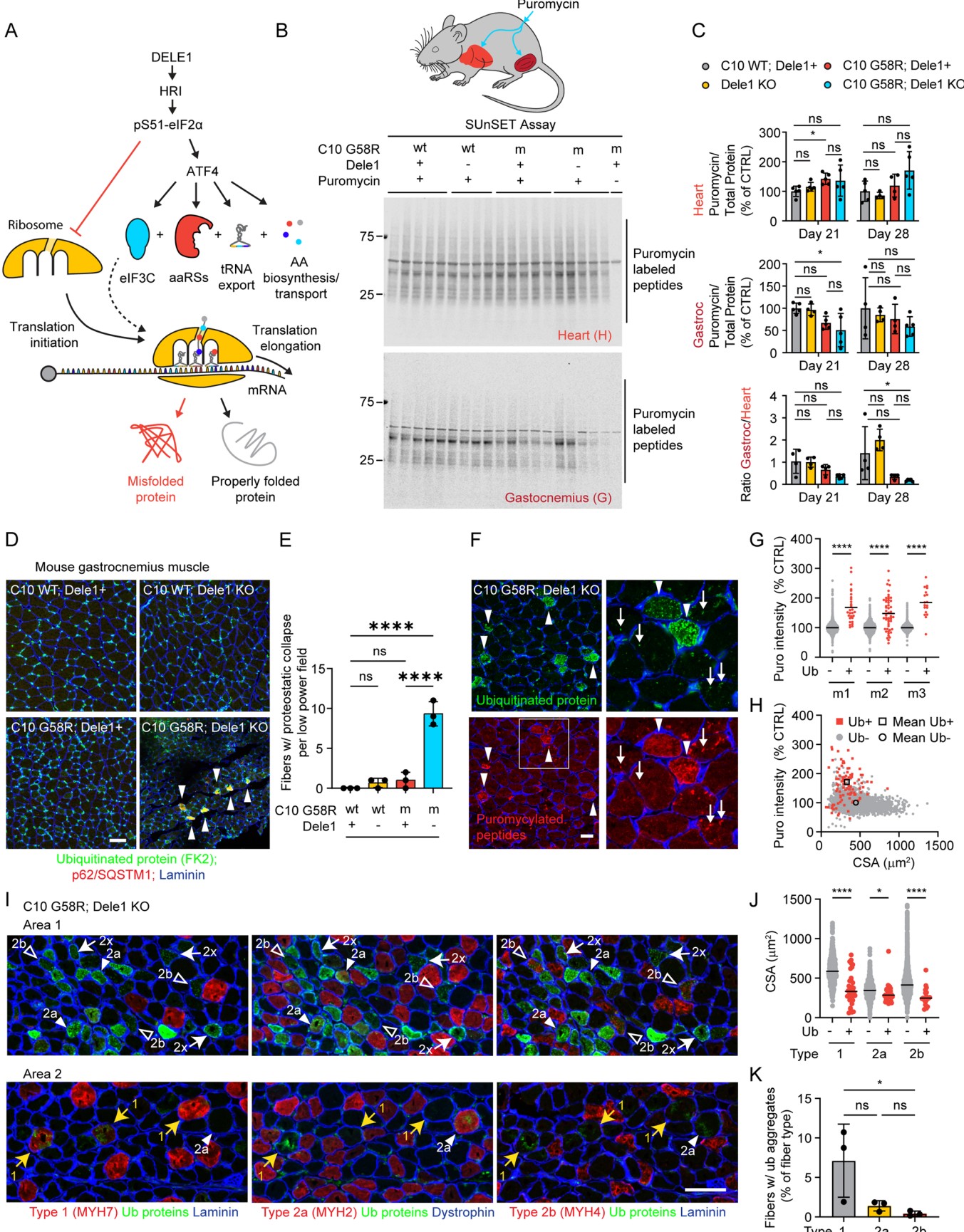

**Figure 7.  DELE1 mt-ISR promotes translation-associated proteostasis in striated muscle.**

(A) Model depicts predicted effects of the DELE1 mt-ISR on protein translation including acute inhibition of translation initiation by pS51-eIF2α, resumption of protein initiation following the transcriptional upregulation of *Eif3c*, and facilitation of translation elongation through increased production of protein synthesis intermediates by the coordinated upregulation of (1) genes related amino biosynthesis and transport, (2) Xpot to mediate tRNA export from the nucleus to the cytosol, and (3) aminoacyl-tRNA synthases (aaRSs) to promote aminoacyl conjugation to tRNAs. Protein translation dynamics and fidelity can affect the proportion of newly translated proteins that fold properly vs. misfold. (B, C) Measurement of in vivo protein synthesis in C10 G58R; *Dele1* KO mice and indicated littermates at P21 and P28, using the SUnSET assay. Blot for P21 timepoint is shown in (B). Mice were injected with the aminoacyl-tRNA ortholog, puromycin, sacrificed 30 min later, and puromycin incorporation into newly synthesized polypeptides in heart and gastrocnemius tissue lysates was measured by immunoblotting. Statistical analysis performed using Welch ANOVA and Dunnett's T3 multiple comparisons test. (Top graph, heart, Day 21) adjusted *p*-values were 0.4894 for C10 WT; Dele1+ vs. Dele1 KO, 0.0236 for C10 WT; Dele1+ vs. C10 G58R; Dele1+, 0.5341 for C10 WT; Dele1+ vs. C10 G58R; Dele1 KO, and 0.9978 for C10 G58R; Dele1+ vs. C10 G58R; Dele1 KO. (Top graph, heart, Day 28) adjusted *p*-values were 0.8419 for C10 WT; Dele1+ vs. Dele1 KO, 0.8923 for C10 WT; Dele1+ vs. C10 G58R; Dele1+, 0.2250 for C10 WT; Dele1+ vs. C10 G58R; Dele1 KO, and 0.4979 for C10 G58R; Dele1+ vs. C10 G58R; Dele1 KO. (Middle graph, gastroc, Day 21) adjusted *p*-values were 0.9990 for C10 WT; Dele1+ vs. Dele1 KO, 0.1338 for C10 WT; Dele1+ vs. C10 G58R; Dele1+, 0.0136 for C10 WT; Dele1+ vs. C10 G58R; Dele1 KO, and 0.7086 for C10 G58R; Dele1+ vs. C10 G58R; Dele1 KO. (Middle graph, gastroc, Day 28) adjusted *p*-values were 0.9841 for C10 WT; Dele1+ vs. Dele1 KO, 0.9399 for C10 WT; Dele1+ vs. C10 G58R; Dele1+, 0.7107 for C10 WT; Dele1+ vs. C10 G58R; Dele1 KO, and 0.8419 for C10 G58R; Dele1+ vs. C10 G58R; Dele1 KO. (Top graph, gastroc/heart ratio, Day 21) adjusted *p*-values were 0.9999 for C10 WT; Dele1+ vs. Dele1 KO, 0.6361 for C10 WT; Dele1+ vs. C10 G58R; Dele1+, 0.2390 for C10 WT; Dele1+ vs. C10 G58R; Dele1 KO, and 0.1941 for C10 G58R; Dele1+ vs. C10 G58R; Dele1 KO. (Top graph, gastroc/heart ratio, Day 28) adjusted *p*-values were 0.8820 for C10 WT; Dele1+ vs. Dele1 KO, 0.4304 for C10 WT; Dele1+ vs. C10 G58R; Dele1 +, 0.3415 for C10 WT; Dele1+ vs. C10 G58R; Dele1 KO, and 0.1941 for C10 G58R; Dele1+ vs. C10 G58R; Dele1 KO. In all graphs, * indicates $p \leq 0.05$ and "ns" not significant. $N \geq 4$ mice per group (genotype). Error bars represent SD. (D) Representative immunofluorescence images of gastrocnemius muscle from P28 C10 G58R; Dele1 KO mice and indicated littermates, showing fibers with many or confluent aggregates of ubiquitinated proteins co-localized with the aggregate-forming adapter protein p62, suggesting proteostatic collapse (arrowheads) in low power (20×) images (top panels). Scale bars = 50 μm. (E) Quantification of (D). $N = 3$ mice per genotype with 30 low power fields counted per genotype. Low-power field size was 397.75 μm × 397.75 μm. Statistical analysis performed using ordinary one-way ANOVA with Šidák's multiple comparisons test, with a single pooled variance. $N = 3$ in each group. $P = 0.8857$ for C10 WT; Dele1+ vs. C10 WT; Dele1 KO. $P = 0.6605$ for C10 WT; Dele1+ vs. C10 G58R; Dele1 +. ns = non-significant. **** = < 0.0001. Error bars represent SD. (F) Representative immunofluorescence images of gastrocnemius muscle from P28 C10 G58R; *Dele1* KO mice injected with puromycin 30 min prior to sacrifice as in (B). Muscle cross-sections were immunostained for ubiquitinated proteins (using the FK2 antibody) (green), puromycin (red), and laminin (blue). Arrowheads indicate muscle fibers containing many or confluent aggregates of ubiquitinated proteins that were also co-stained for elevated puromycylated polypeptides. Arrows indicate individual aggregates of ubiquitin proteins that also contain puromycylated polypeptides. Scale bars = 20 μm. (G, H) Quantification of (F). Individual myofibers were automatically segmented using the Laminin immunofluorescence to define the myofiber border. Average puromycin fluorescence intensity and cross-sectional area (CSA) were measured for each myofiber. Separately myofibers were manually scored as containing many or confluent aggregates of ubiquitinated proteins (Ub+ or Ub- muscle fibers, respectively). The average puromycin intensity for Ub+ and Ub- myofiber is shown in graph separately for three mice (m1–3). $N = 3$ mice with 10 high power fields counted per mouse. The scatterplot in (H) shows the relationship between myofiber CSA and puromycin intensity for all myofibers analyzed in (G). Fibers with CSA < 50 μm were excluded from analysis. Statistical analysis was performed using Kruskal–Wallis test with Dunn's multiple comparisons test for (G). (I) Representative confocal images show three consecutive sections through two areas of gastrocnemius from C10 G58R; Dele1 KO mice triple stained for muscle fiber type 1, 2a, or 2b (red) and FK2 to detect ubiquitin protein aggregates (green) and a sarcolemma marker protein laminin or dystrophin (blue). Fiber type 2x is defined by the absence of any other fiber marker in the consecutive sections. Areas 1 and 2 are magnifications of the boxed areas in Appendix Fig. S16, which shows the whole tissue section. Scale bar = 100 μm. Yellow arrow = type 1 fiber, solid white arrowhead = type 2a fiber, open white arrowhead = 2b fiber, and solid white arrow = 2x fiber. (J) Graph quantifying the CSA of fibers with and without aggregates of ubiquitinated proteins, separated by fiber type as in (I). Gastrocnemius from three C10 G58R; Dele1 KO mice were analyzed. Statistical analysis was performed using Kruskal–Wallis test with Dunn's multiple comparisons test as data was non-parametrically distributed. (K) Graph quantifying the proportion of each fiber types that contained aggregates of ubiquitinated proteins from tissue stained as in (I). Gastrocnemius from three C10 G58R; Dele1 KO mice were analyzed. Statistical analysis was performed using Kruskal–Wallis test with Dunn's multiple comparisons test as data was non-parametrically distributed. $N = 3$ in each group. $P = 0.4081$ for percent of type 1 myofibers with FK2 aggregates vs. percent of type 2a myofibers with FK2 aggregates. $P = 0.0338$ for percent of type 1 myofibers with FK2 aggregates vs. percent of type 2b myofibers with FK2 aggregates. $P = 0.8902$ for percent of type 1 myofibers with FK2 aggregates vs. percent of type 2b myofibers with FK2 aggregates. ns = non-significant. * = <0.05. Error bars represent SD. Source data are available online for this figure.

with a trend toward greater propensity for aggregates of ubiquitinated proteins in more oxidative fibers.

Considered together, these findings suggest that the DELE1 mt-ISR reshapes the metabolic network in stressed striated muscle to promote anabolism, including the continued production of intermediates for protein synthesis. We speculate that loss of the DELE1 mt-ISR may lead to loss of protein translation control and underlie the observed proteostatic collapse, decreased growth, and decreased survival in models of early-onset MM, such as IMMD (Fig. EV5E).

## Discussion

By comparing four MM/CM models resulting from diverse mitochondrial stresses, including protein unfolding and mtDNA depletion, we identified that the OMA1-DELE1 pathway is the predominant pathway for mt-ISR activation in striated muscle. Although the primary source of mitochondrial stress was distinct in each model, they converged on disruption of the IMM, a central component of the

OXPHOS system, which was sensed by OMA1 to activate the mt-ISR through DELE1 (Fessler et al, 2020; Guo et al, 2020). As OXPHOS disruption can affect the biosynthetic function of mitochondria, through decreased turnover of redox equivalents such as NADH (Luengo et al, 2021), the integrity of the IMM may be an important predictor of disruptions in mitochondrial metabolism. By sensing disruption of the IMM, the OMA1-DELE1 pathway may anticipate these disruptions to mitochondrial metabolism.

Notably, the DELE1 mt-ISR did not correct the underlying mitochondrial structural or respiratory chain defects but rather compensated for OXPHOS dysfunction to maintain biosynthesis in the setting of stress, particularly through the upregulation of pathways for the biosynthesis of aminoacyl-tRNAs. It is notable that the ISR in yeast is largely a homeostatic response, in which uncharged tRNAs are sensed by GCN2, which, in turn, limits translation initiation, and triggers a transcriptional response to increase the synthesis of aminoacyl-tRNAs (Postnikoff et al, 2017). In this context, the OMA1-DELE1-HRI may be thought of as a predictive homeostatic response; it anticipates a limitation of biosynthetic intermediates in the

setting of diverse forms of mitochondrial stress and responds by upregulating metabolic pathways to maintain biosynthesis. This response may not slow net protein synthesis per se but may promote the overall translation fidelity by reducing the chance of ribosome stalling or mistranslation.

Consistently, we found that the DELE1 mt-ISR did not slow net protein synthesis in the stressed heart and gastrocnemius, and, in fact, showed a trend toward promoting net protein synthesis in the gastrocnemius. This is also consistent with recent data from the Dars2 cKO model, in which the ISR only slowed translation transiently in the setting of an exaggerated ISR response to chronic mitochondrial stress in the heart but did not mediate a prolonged decrease in translation (Kaspar et al, 2021). It is tempting to speculate that loss of translational control following Dele1 KO may also explain the proteostatic collapse we observed in the striated muscle of C10 G58R; Dele1 KO mice, as ribosome stalling and mistranslation are known to promote protein misfolding (Stein and Frydman, 2019; Stein et al, 2022). Consistently, myofibers with the most severely disrupted proteostasis in C10 G58R; Dele1 KO mice had the highest rates of translation.

The DELE1 mt-ISR was particularly critical for survival in two models, C2/10 DKO and C10 G58R, that experienced mitochondrial stress onset during the rapid period of growth in the first weeks of life. The survival deficit was most dramatic in C2/10 DKO mice. We speculate that the DELE1 mt-ISR is particularly important for maintaining the biosynthetic capacity of stressed striated muscle during periods of rapid hypertrophic muscle growth. It is notable in this context that demands for protein synthesis intermediates are especially high in the first weeks of life in the mouse, as hypertrophic muscle growth accounts for half of the eight-fold increase in body mass during the first three weeks of life (White et al, 2010; Gokhin et al, 2008). This may explain why many of the pathways upregulated by the DELE1 mt-ISR in postmitotic striated muscle (such as serine/glycine synthesis, mitochondrial 1C metabolism, and proline synthesis) are also frequently upregulated in rapidly dividing cancer cells (Westbrook et al, 2022; Ducker et al, 2016; Geeraerts et al, 2021; Nilsson et al, 2014). In both cases, there is a strong demand for protein synthesis intermediates to increase biomass. Although striated muscle may be able to meet this need without upregulation of these pathways under basal conditions, in the setting of mitochondrial dysfunction, metabolic pathways may need to be rebalanced to maintain an uninterrupted supply of biosynthetic intermediates. We thus hypothesize that the importance of the DELE1 mt-ISR depends on the developmental stage of the animal, with the DELE1 mt-ISR being most important during periods of rapid hypertrophic growth of post-mitotic cells. This model would predict that inhibiting the DELE1 mt-ISR in adult mice under mitochondrial stress would be better tolerated than inhibiting the DELE1 mt-ISR in neonatal mice.

Consistent with this hypothesis, the dramatic survival benefit observed in the C10 G58R and C2/10 DKO models contrasted with the more modest survival benefit seen in the Tfam mKO and C10 S59L models. We speculate that the DELE1 mt-ISR is less important for survival in these models because the biosynthetic demands in striated muscle of juvenile and adult mice are less than those of neonatal mice. These demands are still present in the adult, however, and the DELE1 mt-ISR may similarly protect the Tfam mKO and C10 S59L models by maintaining biosynthesis in the setting of mitochondrial stress. The biosynthetic needs in the adult heart may be related to protein synthesis or they may be related to

intersecting metabolic pathways. Glycine and cysteine, for instance, are important for glutathione synthesis and redox homeostasis. In the Cox10 cKO, the DELE1 mt-ISR has been suggested to be important for redox homeostasis and prevention of ferroptosis (Ahola et al, 2022). In this context, it is noteworthy that among the 51 highly conserved genes across the models were the transporter for cystine, Slc7a11, which tends to increase intracellular cysteine levels (which contributes to glutathione synthesis together with glycine and glutamate), and the mitochondrial glutathione transporter, Slc25a39, which may also help prevent damage to membranes from lipid peroxidation and suppress ferroptosis. Likewise, serine, a 1C donor, is important for the de novo synthesis of nucleotides and the production of mitochondrial NADPH through the mitochondrial 1C metabolism pathway. In some models of mtDNA maintenance disorders like the "deletor" mouse, nucleotide synthesis through the 1C metabolism pathway may be particularly critical (Nikkanen et al, 2016). Thus, although the transcriptional response is similar in each model, the metabolic flexibility afforded by increased levels of key enzymes such as the NAD$^+$-dependent PSAT1 and MTHFD2, may serve different biosynthetic needs at different developmental stages, and this may be reflected in downstream differences in the metabolome and proteome among the models.

Our comparison of these four models also suggests that the mt-ISR likely persists as long as the mitochondrial stress remains. Consistently, we previously found that C10 G58R mice, which have onset of the DELE1 mt-ISR prior to P28 (Fig. 1M,N), still had persistent activation of the mt-ISR at around 1 year of age (Shammas et al, 2022). We speculate that if the underlying mitochondrial stress were relieved, the DELE1 mt-ISR would subside, as OMA1 activation can be reversed within minutes, at least in cultured cells (Murata et al, 2020). The persistence of the DELE1 mt-ISR over months and perhaps years also raises the question of whether it remains adaptive in the chronic setting. As the DELE1 mt-ISR was found to be protective also in models with late-onset mitochondrial stress, such as the C10 S59L model, it seems the DELE1 mt-ISR can be protective on the chronic time scale (as in the C10 S59L model) as well as the subacute time scale (as in the C10 G58R and C2/C10 DKO models).

Notably, the DELE1 mt-ISR transcriptional signature identified here overlapped extensively with that in other models of disrupted mtDNA maintenance or expression. Indeed, a similar response, identified first in skeletal muscle of the "deletor" mouse model of MM (Tyynismaa et al, 2010), has been documented with disruptions at each step of mtDNA maintenance and expression, including mtDNA replication (Twinkle KO), mtDNA maintenance (Tfam KO), mtDNA transcription (Polrmt KO), mtRNA stability and processing (Lrpprc KO), and mtRNA translation (Mterf4 KO and Dar2 KO) (Dogan et al, 2014; Kühl et al, 2017) in heart muscle. Our results from the Tfam mKO model strongly suggest that the DELE1 mt-ISR broadly mediates transcriptional mitochondrial stress responses resulting from decreased mtDNA expression or maintenance. Thus, it is likely that the DELE1 mt-ISR underlies a transcriptional signature that it is seen in striated muscle across diverse forms of MM. As mtDNA mutations and disorders of mitochondrial maintenance are the most common causes of primary mitochondrial disorders (Gorman et al, 2015), the DELE1 mt-ISR may be an important response in most forms of MM. Consistently, we found extensive overlap between the DELE1

   

mt-ISR signature in the skeletal muscle of the IMMD mouse model of myopathy and genes upregulated in three cohorts of patients with MM, which included patients with mutations in *Twinkle*, *TK2*, and mtDNA (Hathazi et al, 2020; Pirinen et al, 2020; Kalko et al, 2014). This signature included genes involved in serine, proline, and asparagine biosynthesis, as well as the gene encoding the mitokine FGF21.

Using a model of early-onset MM, we found the DELE1-mediates transcriptional mitochondrial stress responses in skeletal muscle in addition to heart muscle. The transcriptional mitochondrial stress response in skeletal muscle has been best described in the "deletor" mouse model, where it was recently shown to occur in stages, with stage 1 defined by marker genes *Fgf21* and *Mthfd2* (observed starting around 9 months) and stage 2 defined by marker genes *Psat1* and *Pghdh* (starting around 16 months) (Forsström et al, 2019). Expression of stage 2 genes in the "deletor" mice was found to depend on *Fgf21*, suggesting they are activated via an autocrine or paracrine effect of the mitokine FGF21. Notably, we found that in our IMMD model both stage 1 and stage 2 marker genes were elevated in skeletal muscle in the first month of life and both depended on *Dele1* for activation. There are notable differences between the IMMD and "deletor" MM models that may account for these differences in the timing of transcriptional stress responses. The "deletor" mice develop myopathy as adults and mitochondrial dysfunction is highly variable between skeletal muscle fibers, because of the stochastic mtDNA deletion process occurring in each muscle fiber (Tyynismaa et al, 2005). By contrast, the IMMD model has early-onset of MM and mitochondrial dysfunction likely occurs more uniformly, as evidenced by globally reduced COX staining (but few truly COX negative fibers) in striated muscle (Shammas et al, 2022). The paracrine effects FGF21 may be more critical in the setting of mosaic OXPHOS dysfunction across fibers. Alternatively, local FGF21 levels may reach a critical level for stage 2 gene activation more quickly in the setting of uniform mitochondrial dysfunction. It is not yet clear if there are stages in the IMMD model that are compressed in time or if this model lacks the staged response observed in the "deletor" mice. Examination of whether *Fgf21* KO blocks activation of stage 2 genes in the IMMD model might help address this question.

Our findings have several potentially important clinical implications for the treatment of MM and MC. First, they suggest that inhibiting the ISR in most cases of MM will likely be harmful, as it is adaptive in response to diverse mitochondrial stresses. This contrasts with other conditions like Vanishing White Matter Disease, where ISR inhibition is being actively pursued as a therapeutic strategy (Wong et al, 2019). Second, our work establishes that common biomarkers of MM, such as FGF21 (Lehtonen et al, 2016), likely reflect signaling through the OMA1-DELE1 pathway. Understanding the mechanism for biomarker regulation is important for interpreting how they are affected by disease progression and their response to therapies for MM. In addition, our work raises the possibility that enhancing the mt-ISR may be protective. Notably, in three of the models that we tested, the mt-ISR was activated later in the disease course. Future work might test whether therapeutically activating the ISR early in the disease course, in anticipation of progression, or above physiologic levels may be beneficial. Several pharmacological activators of the ISR have been described, including halofuginone (targeting EPRS to increase eIF2α phosphorylation by GCN2) and Sephin1 (inhibiting

an eIF2α phosphatase) (Das et al, 2015; Keller et al, 2012), the latter of which was recently shown to augment the DELE1 mt-ISR in an AFG3L2 mutant model (Franchino et al, 2024). Finally, as DELE1 exerts a strong myoprotective effect in models of MM/CM without attenuating most ultrastructural mitochondrial defects or OXPHOS deficiency, it suggests that cellular stress response to mitochondrial dysfunction should be monitored in addition to changes to mitochondrial structure and OXPHOS in assessing potential interventions for mitochondrial disorders.

Together our findings demonstrate a stereotyped DELE1-dependent response to diverse forms of mitochondria stress in MM. These mitochondrial stressors likely converge on disruption of the IMM and IMM-dependent OXPHOS. The response circumvents disruptions to biosynthetic pathways coupled to OXPHOS by rebalancing the metabolic network through a coordinated transcriptional program. This response is particularly critical during periods of rapid growth when biosynthetic demands are high. These findings suggest that interventions to promote the biosynthetic functioning of stressed striated muscle may be protective and cautions against inhibition of the ISR in MM.

## Limitations

Our study has several limitations that should be considered in interpreting the findings. One is the use of a constitutive (germline) *Dele1* KO allele, which may allow for compensation during development. Arguing against developmental compensation, however, the unstressed *Dele1* KO mouse did not exhibit any differences in its transcriptome in the tissues assayed (excepting decreased *Dele1* expression in some datasets) (Appendix Figs. S9D and S12B). Similarly, heart mitochondria appeared to be unaltered both in terms of the expression of OXPHOS complex subunits and mitochondrial ultrastructure (Appendix Fig. S1D–G). In addition, eIF2α failed to be phosphorylated in the absence of *Dele1*, despite mitochondrial stress (Appendix Fig. S15C), indicating that HRI and the three other eIF2α kinases are not activated by mitochondrial stress in the absence of *Dele1*. Consistently, the majority of ATF4 targeted transcripts that were upregulated by mitochondrial stress were silenced by *Dele1* KO (Fig. 4B). Thus, it seems that activation of the ISR in striated muscle by mitochondrial stress is heavily dependent on DELE1 signaling. It should be noted, however, that while our assessment finds *Dele1* KO is largely aphenotypic at steady-state, more subtle phenotypes may have been overlooked. For instance, the IMPC project has previously generated and phenotyped a separate *Dele1* KO model, which found significant differences in the morphology of the vertebral arch, ovaries, vibrissa, increased circulating potassium, and enhanced contextual conditioning at steady-state, demonstrating that there may be specific abnormalities resulting from loss of *Dele1* (https://www.mousephenotype.org/data/genes/MGI:1914089). The connection of these phenotypes to mitochondrial stress responses are not obvious at present, except for potentially enhanced contextual conditioning behavior given the role of the ISR in learning. Another potential limitation is that the experiments were performed on the C57BL/6J background, and it is recognized that genetic background can modify mitochondrial phenotypes. Nonetheless, we speculate that we would observe the same general relationships between models and among genotypes on other backgrounds. Finally, while there are similarities between in the muscle physiology of mice and humans, there are also important differences, and caution is warranted in translating what is learned in these preclinical models to patients with MM.

# Methods

### Reagents and tools table

| Reagent/Resource | Reference or Source | Identifier or Catalog Number |
|---|---|---|
| **Experimental Models** | | |
| CHCHD10 G58R mouse | Shammas et al, 2022 | MGI:7532640 |
| CHCHD10 S59L mouse | Liu et al, 2020 | |
| CHCHD2/CHCHD10 DKO mouse | Liu et al, 2020 | |
| OMA1 KO mouse | Quirós et al, 2012 | MGI:5426689 |
| Dele1 KO mouse | This study | |
| OPA1 Δs1 mouse | This study | |
| OPA1 Δs1 primary neonatal fibroblasts | This study | |
| Ckmm-cre mouse | Brüning et al, 1998 | JAX stock #006475 |
| Tfam^fl/fl mouse | Hamanaka et al, 2013 | JAX stock #026123 |
| **Recombinant DNA** | | |
| OMA1 ASOs and control ASOs | Shammas et al, 2022 Ionis Pharmaceuticals | |
| **Antibodies** | | |
| Mouse monoclonal anti-OPA1 | BD Bioscience | 612606 |
| Mouse monoclonal anti-OMA1 | Santa Cruz | SC-515788 |
| Rabbit polyclonal anti-HSP90 | Proteintech | 13171-1-AP |
| Mouse monoclonal anti-β-ACTIN | Sigma | A2228 |
| Rabbit monoclonal anti-eIF2α | Cell Signaling | 5324 |
| Rabbit monoclonal anti-p-eIF2α | Cell Signaling | 3597 |
| Rabbit polyclonal anti-MTHFD2 | Proteintech | 12270-1-AP |
| Mouse monoclonal anti-Puromycin | Sigma | MABE343 |
| IRDye 800CW Goat anti-Rabbit IgG (H + L) | LI-COR Biosciences | 926-32211 |
| IRDye 680RD Goat anti-Mouse IgG (H + L) | LI-COR Biosciences | 926-68070 |
| IRDye 800CW Goat anti-Mouse IgG2a-Specific | LI-COR Biosciences | 926-32351 |
| Mouse monoclonal anti-P62 | Abcam | AB56416 |
| Mouse monoclonal anti-ubiquitinylated proteins antibody, clone FK2 | Sigma | 04-263 |
| Rabbit monoclonal anti-Ubiquitin FK2 | StressMarq | SMC-550D |
| Rabbit polyclonal anti-Laminin | Abcam | AB11575 |
| Mouse monoclonal anti-Dystrophin, exons 31/32. | DSHB | MANDYS1, 3B7/ AB_528206 |
| Mouse monoclonal anti-Myosin heavy chain (slow, alpha- and beta-) (type 1) | DSHB | BA-F8/ AB_10572253 |
| Mouse monoclonal anti-Myosin heavy chain Type IIA | DSHB | SC-71/AB_2147165 |
| Mouse monoclonal anti- Myosin heavy chain Type IIB | DSHB | 10F5/AB_1157896 |
| Alexa 488 Goat anti-Mouse IgG1 | Invitrogen | A-21121 |
| Alexa 647 Goat anti-Mouse IgG2a | Invitrogen | A-21241 |

| Reagent/Resource | Reference or Source | Identifier or Catalog Number |
|---|---|---|
| Alexa 555 Goat anti-Rabbit IgG (H + L) | Invitrogen | A-32732 |
| Alexa 488 Goat anti-mouse IgM | Invitrogen | A-21042 |
| Alexa 488 Goat anti-mouse IgG2a | Invitrogen | A-21131 |
| Alexa 488 Goat anti-mouse IgG2b | Invitrogen | A-21141 |
| Alexa 647 Goat anti-mouse IgG1 | Invitrogen | A-21240 |
| Alexa 647 Goat anti-rabbit (H + L) | Invitrogen | A-32728 |
| Alexa 555 Goat anti-mouse IgG1 | Invitrogen | A-21127 |
| **Oligonucleotides and other sequence-based reagents** | | |
| Genotyping primer 1 of Dele1 KO mice | This study | TGATGGTGTGACTGTCCCTT |
| Genotyping primer 2 of Dele1 KO mice | This study | GGTTGTCTGAGACCCATCTGA |
| Genotyping primer 1 of OPA1 Δs1 mice | This study | CGTGCACCCCACTTACTTTT |
| Genotyping primer 2 of OPA1 Δs1 mice | This study | ATGCAAAGTCCCAGGAAAGA |
| **Chemicals, Enzymes and other reagents** | | |
| Paraformaldehyde 16% aqueous, EM grade | Electron Microscopy Sciences | 15710 |
| Glutaraldehyde 50% aqueous, EM grade | Electron Microscopy Sciences | 16320 |
| Phosphate Buffered Saline | Lonza, Fisher, or Corning | |
| Sodium Cacodylate Buffer, 0.2 M, pH 7.4 | Electron Microscopy Sciences | 11653 |
| Calcium Chloride, 2 M | Quality Biological, Inc. | 351-130-721 |
| Osmium Tetroxide 4% aqueous solution | Electron Microscopy Sciences | 19150 |
| Potassium hexacyanoferrate (II) trihydrate | Sigma-Aldrich | P3289-100G |
| Sodium Acetate Trihydrate | Electron Microscopy Sciences | 21120 |
| Uranyl Acetate | Electron Microscopy Sciences | 22400 |
| Ethyl Alcohol, 200 proof | The Warner-Graham Co. | 6505001050000 |
| Embed 812 Embedding Kit with Dmp-30 | Electron Microscopy Sciences | 14120 |
| BanII Endonuclease | NEB | R0119S |
| BSA fraction V | MP Biomedicals | 02160069 |
| BCA kit | Thermo Scientific | 23225 |
| Direct-zol RNA Miniprep Kit | Zymo | R2051 |
| RNeasy Fibrous Tissue Mini Kit | QIAGEN | 74704 |
| RNeasy Lipid Tissue Mini Kit | QIAGEN | 74804 |
| Carbonyl cyanide 3-chlorophenylhydrazone (CCCP) | Sigma-Aldrich | C2759-100MG |
| Dimethyl Sulfoxide (DMSO) | Corning | 25-950-CQC |
| NEBNext® Ultra™ II Directional RNA Library Prep Kit | New England Biolabs | E7660 |
| RIPA buffer | Cell Signaling Technology | 9806 |
| Protease/Phosphatase Inhibitor Cocktail | Cell Signaling Technology | 5872 |

| Reagent/Resource | Reference or Source | Identifier or Catalog Number |
|---|---|---|
| ProLong Diamond Antifade Mountant | Invitrogen | P36965 |
| Triton X-100 | Sigma-Aldrich | X100-5ML |
| Dithiothreitol (DTT) | Bio-Rad Laboratories | 1610611 |
| 4x Laemmli Sample Buffer | Bio-Rad Laboratories | 1610747 |
| Dry milk powder | Dot Scientific | DSM17200-1000 |
| 4–15% Criterion TGX Precast Midi Protein Gel | Bio-Rad Laboratories | 5671084 |
| 7.5% Criterion TGX Precast Midi Protein Gel | Bio-Rad Laboratories | 5671024 |
| MyTaq Red Mix, | Bioline | BIO-25044 |
| SYBR Safe DNA Gel Stain | Invitrogen | S33102 |
| Clariom S Mouse microarray | Affymetrix/Applied Biosystems | 902931 |
| Seahorse XF Pro M FluxPak Mini | Agilent Technologies | 103777-100 |
| **Software** | | |
| Adobe Illustrator 2023 | Adobe | |
| Adobe Photoshop 2023 | Adobe | |
| Fiji/ImageJ | NIH | RRID:SCR_002285 |
| Cellpose 2.0 | Pachitariu and Stringer, 2022 | |
| LabelsToROIs of FIJI plugin | Waisman et al, 2021 | |
| Transcriptome Analysis Console Software, version 4.0.1 | Affymetrix | RRID:SCR_018718 |
| Perseus | MaxQuant | RRID:SCR_015753 |
| Image Studio, v5.2 | LI-COR | RRID:SCR_0157 95 |
| GraphPad Prism, version 10.1.0 (264) | https://www.graphpad.com | RRID:SCR_0027 98 |
| Miniconda3 | https://docs.anaconda.com/miniconda/ | |
| Empanada-napari | Conrad and Narayan, 2023, Cell Systems 14, 58-71 | |
| MitoNet plugin | Conrad and Narayan, 2023, Cell Systems 14, 58-71 | |
| Proteome Discoverer 2.4 | Thermo Fisher Scientific | |
| Binner | Kachman et al, 2020 | |
| MetaboAnalyst 6.0 | https://www.metaboanalyst.ca/MetaboAnalyst/ | |
| g:Profiler | Raudvere et al, 2019 https://biit.cs.ut.ee/gprofiler/gost | |
| **Other** | | |
| Mouse Heart Slicer Matrix, 1 mm coronal section | Zivic Instruments | HSMS001-1 |
| Grip strength instrument | BIOSEB | EB1-BIOGS3 |
| 1.5H coverslips | Thorlabs | CG15KH |
| Confocal laser scanning microscope | Olympus | FLUOVIEW FV3000 |
| Seahorse XF Pro Analyzer | Agilent Technologies | |
| UltiMate 3000 high pressure liquid chromatography instrument | Thermo Fisher Scientific | |
| Orbitrap Lumos mass spectrometer | Thermo Fisher Scientific | |

## Methods and protocols

### Mouse models

Mice were maintained on a 12-h light/12-h dark cycle, with food and water provided ad libitum unless otherwise stated. The $Opa1^{\Delta s1/+}$ and $Dele1^{+/-}$ mice were generated using CRISPR/Cas9 endonuclease-mediated genome editing on a C57Bl6J background by the NHLBI Transgenic Core. To generate $Dele1^{+/-}$ mice, sgRNAs targeting exon 2 (TGGGTTTGGGCGTTATCCCA) and exon 3 (sequence: 5' ATGGCGTCCCGCAGTGTGTT) were used; the indel was obtained in exon 2. To generate $Opa1^{\Delta s1/+}$, exon 5 of mouse $Opa1$ isoform 1, was targeted with two sgRNA sequences (GCTCGAAATGCTGTT TCTCC and CTGAAAGTGACAAGCATTAC) to generate a small deletion together with a small donor single stranded oligonucleotide with the sequence: GGGTTGTTTACATCATTGGAGAATTTAAA-CAAGTGTGTTTATCTTTAAGGTTCACCTGGAGAAAGTGACA AGCATTACAGGAAGGTAAGTATAAAACAGAACTCATCTCA-CAGGGAGTCC.

The deletions were confirmed by Sanger sequencing (Eurofins). Genotyping oligos for $Dele1$ KO had the following DNA sequences: TGATGGTGTGACTGTCCCTT and GGTTGTCTGAGACCCATC TGA. Genotyping oligos for $Opa1^{\Delta s1/\Delta s1}$ had the following DNA sequences: CGTGCACCCCACTTACTTTT and ATGCAAAGTCC-CAGGAAAGA. $Ckmm$-Cre (JAX stock #006475) and $Tfam^{fl/fl}$ mice (JAX stock #026123) were obtained from Jackson laboratory (Brüning et al, 1998; Hamanaka et al, 2013). The generation of C10 G58R (MGI:7532640), C10 G59L (no MGI registration), C2/C10 DKO (no MGI registration), and OMA1 KO (MGI:5426689) mice were described previously (Liu et al, 2020; Shammas et al, 2022; Quirós et al, 2012). In all cases, the mice were maintained on a C57BL/6J background. The Oma1 KO was originally on a mixed background including 129S6/SvEvTac, C57BL/6, and C57BL/6NCr and were kind gift from Dr. Carlos Lopez-Otin and Dr. Pedro M. Quirós (Quirós et al, 2012). The specific crosses that were used to generate the mice used in comparisons are shown in Appendix Fig. S17. All animal studies were approved by the Animal Care Use Committee at the NINDS, NIH intramural research program. Both sexes were used in all studies.

### Nutritional support

Mice in the nutritional support group were weaned on P28 instead of P21. In addition to the regular chow on the racks, they received KMR milk replacer using a small syringe twice daily from Monday to Friday and once daily on Saturday and Sunday. Other dietary supplements, including plain soft chow, Dietgel 76A (ClearH$_2$O), and bacon-flavored treats, were also provided on the cage floor ad libitum.

### Cell culture

Primary neonatal fibroblasts were generated using methods described previously (Liu et al, 2020). Primary neonatal fibroblasts generated from $Opa1^{\Delta s1/\Delta s1}$ mice and from the wild-type littermates, described above, were treated with 20 µM CCCP or DMSO vehicle only for 16 h. Cell lines were not tested for mycoplasma.

### Cold experiments

Adult mice in the experimental group were transferred to individual plastic cages with pre-chilled water but without bedding

or food in a 4 °C cold room while mice in the control group remain room temperature. After 9 h of exposure, mice were anesthetized with isoflurane and euthanized by cervical dislocation. Interscapular BATs were harvested immediately.

### Behavioral tests

Mouse forelimb grip strength was assessed using a BIOSEB instrument with grid attachment (catalog EB1-BIO-GS3). Three grip strength measurements were taken per mouse with 15-s resting periods between the trials. Composite phenotype scores were assigned based on the protocol described previously (Guyenet et al, 2010). All tests were performed by the same tester, who was blinded to the genotypes of the mice.

### RNA microarray

RNA microarrays were performed and analyzed as described previously (Shammas et al, 2022). In brief, RNA was extracted from frozen mouse hearts, gastrocnemius muscles, livers, and BATs using the Direct-zol RNA Miniprep Kit (Zymo, catalog R2051), the RNeasy Fibrous Tissue Mini Kit (QIAGEN, catalog 74704), or the RNeasy Lipid Tissue Mini Kit (QIAGEN, catalog 74804). RNA expression was measured using the Clariom S Mouse microarray (Affymetrix) by the NHGRI Microarray Core. Transcriptome Analysis Console software (Affymetrix, version 4.0.1) was used to analyze the data with the default settings.

### RNA-Seq

RNA was extracted as described above. The library was prepped with NEBNext® Ultra™ II Directional RNA library prep kit with sample purification beads (New England Biosciences, Inc.), following the manufacturer's instructions. RNA was sequenced on the NextSeq 2000 platform (Illumina), using a P3 flow cell. For RNA-Seq data analysis, counts per gene were normalized by gene length first, followed by normalization for differences in sequencing depth, using the counts per million procedure. Normalized values were pedestalled by (+2) then Log2 transformed. Cross-sample normalization was then performed to correct for differences in expression of distribution spread and location, using the "cyclicloess" procedure. After inspection of results in boxplots, noise was modeled to identify and remove noise-biased genes. Values for each gene were subset per cohort and fit via lowess to produce trend lines for each experiment group, in order to determine the dynamic range. Noise-cutoffs were defined where the trends deviated from linearity. Expression values less than cut-off were floored to the cut-off and genes not having at least one sample greater than the cut-off were removed as noise-biased and non-informative. A similar procedure was used for those genes with expression with values higher than the ceiling cut-off. ~12,870 genes remained after noise floor and filter processing. ANOVA (Expression ~ Group + Sex) was applied to these followed by post hoc testing. Key results from these processing steps are preserved in Dataset EV7–EV9 in tabs moving from left to right.

### Immunoblotting

Immunoblotting and densitometric measurements were performed as described previously (Liu et al, 2020). In brief, mouse frozen hearts, BATs, and fibroblasts were lysed in RIPA buffer (Cell Signaling Technology, catalog 9806) with Protease/Phosphatase Inhibitor Cocktail (Cell Signaling Technology, catalog 5872). A

buffer containing 20 mM Tris pH 7.8, 137 mM NaCl, 2.7 mM KCl, 1 mM MgCl$_2$, 1% Triton X-100, 10% glycerol, 1 mM ethylenediamine tetraacetic acid and 1 mM dithiothreitol was used instead to lyse mouse frozen skeletal muscle. OPA1 bands on the blot were quantified using FIJI software (Schindelin et al, 2012). All other protein bands and total protein stains (LI-COR Biosciences, catalog 926-11016) were quantified using Image Studio v5.2 (LI-COR Biosciences).

### Histological analysis

BATs were fixed in 4% paraformaldehyde in phosphate-buffered saline (PBS) overnight and washed in PBS three times. Fixed tissues were sent to Histoserv (Germantown, MD) for paraffin embedding, microtome sectioning, and haematoxylin and eosin (H&E) staining by standard procedures.

### Immunofluorescent staining

Mice were anesthetized with isoflurane and transcardially perfused with PBS. Mouse tissues were dissected and frozen in chilled isopentane. 10 μm cross cryosections of the heart and of gastrocnemius mid-belly region were collected onto glass slides. Sections were washed with 0.1% Triton X-100 in PBS (PBST), blocked with 5% BSA in PBST for 1 h at room temperature, and incubated in primary antibodies with 0.5% BSA in PBST overnight at 4 °C. Then, sections were washed in PBST and incubated in secondary antibody in PBST for 1–2 h at room temperature. Slides were washed with PBST and PBS before coverslipped with ProLong Diamond Antifade Mountant (Invitrogen, catalog P36965) and sealed with nail polish. Images were obtained using Olympus FLUOVIEW FV3000 confocal laser scanning microscope.

### Quantification of CSA of muscle fibers

Muscle fiber segmentation was performed using Cellpose 2.0 (Pachitariu and Stringer, 2022), deep learning program that utilizes human-in-the-loop training models to further automate the segmentation process, on laminin-labeled immunofluorescence images. After segmentation, CSA was quantified using LabelsToROIs of FIJI plugin (Waisman et al, 2021). Scoring of fibers containing FK2 and/or P62 positive aggregates was performed by a rater blinded to the genotype. The rater was not blind for additional analysis focusing on correlations between puromycin staining, FK2 positive fibers, and CSA analysis from images in the C10 G58R; Dele1 KO samples.

### Muscle fiber type immunofluorescence

The gastrocnemius of C10 G58R; *Dele1* KO mice was harvested, and flash-frozen in isopentane for muscle fiber typing. Serial sections were obtained to allow for tracking and comparison of different muscle fiber type. Each section was either stained for Type 1 (DSHB, catalog AB_10572253), Type 2a (DSHB, AB_2147165), or Type 2b (DSHB catalog AB_1157896) muscle fibers, in combination with Laminin or Dystrophin to stain muscle fiber membranes and FK2 for ubiquitinated protein. Quantification of fiber type CSA was performed as previously stated. Percentage of relative muscle fiber type exhibiting FK2 aggregates was quantified.

### Mitochondrial isolation

Mitochondria were isolated from mouse heart tissues as described previously (Frezza et al, 2007).

## Proteomics

Label-free quantitative proteomics of mitochondria isolated from mouse heart tissue was performed by the NINDS Proteomics Core Facility as described previously for most samples (Shammas et al, 2022). In brief, liquid chromatography-tandem mass spectrometry (LC-MS/MS) data acquisition was performed on an Orbitrap Lumos mass spectrometer (Thermo Fisher Scientific) coupled with a 3000 Ultimate high-pressure liquid chromatography instrument (Thermo Fisher Scientific). Peptides were separated on an ES802 column (Thermo Fisher Scientific) with the mobile phase B (0.1% formic acid in acetonitrile) increasing from 3 to 22% over 70 min. The LC MS/MS data were acquired in data-dependent mode. For the survey scan, the mass range was 400–1500 $m/z$; the resolution was 120 k; the automatic gain control (AGC) value was 8e5. For C10 G58R and related genotype samples, a FAIMS interface was used. In this case the gradient MPB was increased from 3 to 20% over 63 min and the mass range of the MS1 scan was 375–1500 $m/z$. In all cases, the MS1 cycle time was set to 3 s. As many MS2 scans as possible were acquired within the cycle time. MS2 scans were acquired in ion-trap with an isolation window of 1.6 Da. Database search and mutant/WT ratio calculation were performed using Proteome Discoverer 2.4 (Thermo Fisher Scientific) against Sprot Mouse database. Proteins were annotated as mitochondrial if they appeared in mouse MitoCarta3.0, and were additionally annotated with MitoPathways from MitoCarta3.0 (Rath et al, 2021).

## Seahorse assays

Seahorse oxygen consumption-based measurements of CI and CIV activities from frozen mitochondria isolated from hearts were performed according to the protocol described previously (Osto et al, 2020). 0.4 μg/well of frozen mitochondria were used with Seahorse XF Pro Analyzer (Agilent Technologies).

## ASO experiment

ASOs were synthesized at Ionis Pharmaceuticals as previously described (Seth et al, 2010; Shammas et al, 2022). For the OMA1 knockdown in liver for transcriptomics, twelve C10 G58R; $Oma1^{+/-}$ mice were given weekly subcutaneous injections of either control ASOs or OMA1 ASOs for total of 12 weeks as described previously (Shammas et al, 2022). For the OMA1 metabolomics experiment, adult $C10^{wt/wt}$; $Oma1^{+/-}$ mice were given weekly subcutaneous injections of PBS. Adult $C10^{G58R/wt}$; $Oma1^{+/-}$ mice were given weekly subcutaneous injections of either a nontargeting (control) ASO or an OMA1-targeting ASOs. The mice were between 10.5 months and 13.5 months old. The ASOs were at a concentration of 5 mg/mL, and injections were dosed at 50 mg/kg. A total of 6 injections were administered per mouse over 6 weeks. Then, the mice were anesthetized with isoflurane and transcardially perfused with PBS, and their tissues were collected and flash-frozen in liquid nitrogen.

## Metabolomics

Mice were anesthetized with isoflurane and transcardially perfused with PBS. Mouse hearts were harvested and freeze-clamped immediately. Metabolomics data of the ASO experiment was obtained by the Metabolomics Core at the University of Michigan using their non-targeted metabolomics platform. Metabolites were extracted as previously described (Overmyer K et al, 2015). In brief, frozen pulverized mouse heart tissue samples were homogenized with chilled extraction solvent by pulsed sonication in tubes in an ice bath. Samples were centrifuged for 10 min at $16,000 \times g$ at 4 °C after resting on ice for 5 min. The supernatants obtained were transferred for LC-MS analysis. The tissue pellets were dried, and their masses were measured for normalization. Manual peak integration was performed on named compounds, and automated peak detection was performed for other mass spectrometer signals. The abundances of these compounds were processed through Binner for data reduction (Kachman et al, 2020). RefMet (https://doi.org/10.1093/bioinformatics/btz798) was the database used. Metabolomics data of the C10 G58R and Tfam mKO experiments were obtained by Metabolomics Core Resource Laboratory at NYU Langone Health. Metabolites used for assessment of quality control standards during data acquisition include phenylalanine-13C,15N-9,1, isoleucine-13C,15N-6,1, leucine-13C,15N-6,1, methionine-13C,15N-5,1, valine-13C,15N-5,1, proline-13C,15N-5,1, tyrosine-13C,15N-9,1, threonine-13C,15N-4,1, histidine-13C,15N-6,3, alanine-13C,15N-3,1, glutamate-13C,15N-5,1, aspartate-13C,15N-4,1, glycine-13C,15N-2,1, serine-13C,15N-3,1, lysine-13C,15N-6,2, and arginine-13C,15N-6,4.

## SUnSET assays

Measuring protein synthesis in mouse tissues was performed according to the protocol described previously (Ravi et al, 2020). In brief, puromycin 40 nmol/g of body weight was injected intraperitoneally to C10 G58R; Dele1 KO mice and their littermates. Mice were scarified after 30 min from the injection. Their tissues were collected and flash-frozen in liquid nitrogen. Immunoblotting was performed as described above. For the P21 timepoint, one litter of mice was excluded from analysis as no puromycin labeling was observed in the control animals suggesting a technical problem with the puromycin administration. This criterion for exclusion was not prespecified. All animals injected for the P28 timepoint were used in analysis.

## Transmission electron microscopy of mouse tissues

Mice were deeply anaesthetized and transcardially perfused with PBS. Hearts were rapidly dissected, and sub-millimeter pieces of tissue were excised from the left ventricle and immersed in freshly prepared fixative containing 4% glutaraldehyde and 2 mM calcium chloride in 0.1 M sodium cacodylate buffer, pH 7.4 (Electron Microscopy Sciences, Hatfield, PA, USA [EMS]). In the case of tissue obtained for the C2/C10 DKO mutant, mice were anaesthetized with isoflurane and then transcardially perfused with 2% paraformaldehyde, 2.5% glutaraldehyde, and 2 mM calcium chloride in 0.1 M sodium cacodylate buffer. Perfusion-fixed hearts were excised and submerged in storage fixative containing 2% glutaraldehyde in 0.1 M cacodylate buffer. Hearts were immediately further dissected by cutting a 1 mm-thick coronal section at the midpoint of the heart using a mouse heart slicer matrix (Zivic Instruments, Pittsburgh, PA, USA). The wall of the left ventricle in the coronal slice was cut into sub-millimeter pieces with a razor blade, submerged in storage fixative and stored at 4 °C until processing for EM. Brown adipose tissue was initially fixed and stored in 4% paraformaldehyde in PBS at 4 °C long-term. Prior to EM processing, the tissue was rinsed with 0.1 M cacodylate buffer and fixed overnight at 4 °C with 2% glutaraldehyde in 0.1 M cacodylate buffer.

For EM processing, pieces of tissue were rinsed with 0.1 M sodium cacodylate buffer (cacodylate buffer) three times and then postfixed on ice with reduced osmium containing 1% osmium

tetroxide (EMS) and 1% ferrocyanide (Fisher Scientific, Pittsburgh, PA, USA) in cacodylate buffer for one hour. After three, 5-min rinses in cacodylate buffer, tissue was rinsed in 0.1 N acetate buffer (pH 5.0–5.2) at room temperature and then en bloc stained overnight with 1% uranyl acetate (EMS) in acetate buffer at 4 °C. Tissue was then rinsed in acetate buffer and dehydrated through a graded series of 10-min ethanol rinses prior to infiltration in EmBED812 epoxy resin using the manufacturer's hard resin formulation (EMS). Finally, tissue was embedded in flat embedding molds and resin was polymerized in a 60 °C oven for 48 h.

Ultrathin sections were cut to a thickness of 60–70 nm using an ultramicrotome (EM UC7, Leica Microsystems, Wetzlar, Germany) equipped with a diamond knife (DiATOME, Hatfield, PA, USA). Sections were picked up on formvar coated 200-mesh or 1 mm-slot copper grids (EMS), and post-stained with 3% Reynold's lead citrate (EMS) for 1 min. Sections were viewed using a JEOL1400 Flash transmission electron microscope (JEOL USA, Inc., Peabody, MA, USA) operated at 120 KV. Images were acquired using a 29 Mpix CMOS detector (Advanced Microscopy Techniques, AMT, Danvers, MA, USA). Images from serial sections were aligned using the TrakEM2 plugin of Fiji image analysis software (Schindelin et al, 2012) which is a version of the open-source image analysis software, Image J2 (Rueden et al, 2017). For display, linear adjustments to light and dark levels in greyscale images were made Adobe Photoshop 2023 and Figures were prepared using Adobe Illustrator 2023 (Adobe Inc., San Jose, CA, USA).

### Semi-automated analysis of mitochondria size, shape, and ultrastructural features in TEM images

Mitochondria were auto-segmented using MitoNet, a deep learning segmentation model, operated with Python software, using the napari plungin, called empanda (Conrad and Narayan, 2023). Structural features of mitochondria in TEM images were analyzed from two mice of each genotype (except for the *Tfam* mKO for which only one littermate of control and *Tfam* mKO, Dele1 KO were available). To carry out the analysis, images of at least two areas of tissue for each littermate were recorded at 2000× direct magnification which encompassed a 550 μm² field of view (examples for each genotype are shown in Fig. 4). These areas were selected based on being located towards the interior of the tissue away from edges cut during dissection, and areas where the plane of the thin section was semi-longitudinal with respect to the muscle fibers, rather than transversely oriented. Within these areas, images of at least five subareas were acquired at 5000× direct magnification, each of which encompassed an 87 μm² field of view (examples are shown in Fig. 4). Two or more of these 5000× images were randomly selected for structural analysis using MitoNet segmentation. For each image, labels were automatically assigned to mitochondria detected in the TEM image by empanada-napari. The segmentation labels were manually proofread and corrected as needed and then the set of labels for each image was used to log the size features of each mitochondria including area, major and minor axis lengths, and aspect ratio into Excel (Microsoft Corporation, Redmond, WA, USA). A minimum of 600 mitochondria per genotype were analyzed. The same labels created in empanada-napari were used to tabulate ultrastructural features of individual mitochondria in the control and mutant genotypes. With labels overlayed on the TEM image, each mitochondrion was examined and manually scored as normal or abnormal, and structural features noted in the Excel file that also contained the shape features logged for each mitochondrion. The percent of mitochondria with various structural features per genotype were graphed using GraphPad Prism 10.1.0 (316) version for Windows (GraphPad Software, Boston, MA, USA, www.graphpad.com). Raters were not blind to genotype for these analyses.

### Statistics

For microarray transcriptomic datasets, reported *p*-values and FDR values were calculated in TAC v4.0.3 software across all genes in dataset, using the default settings, which uses a one-way ANOVA corrected with the Benjamini-Hochberg procedure for multiple procedures. For proteomics and metabolomics datasets, two-sided Student's t-tests were performed, using SciPy 1.14.0 library in Python, and values were corrected for multiple comparisons across all gene groups/features by calculating an FDR with the Benjamini–Hockberg procedure, using the statsmodels 0.14.1 library in Python.

For these datasets, DELE1-dependent genes/metabolites met the following conditions:

(1) Significant in the stress; Dele1+ vs. control comparison (FDR ≤ 0.05 and abs(log2FC) ≥ 1) and

(2) Significant in the stress; Dele1 KO vs. stress; Dele1+ comparison and

(3) FC is in the opposite direction for (1) and (2).

Statistical analysis for the RNA-Seq data was similar except that significance was first tested with a two-way ANCOVA including genotype and sex as variables. For post hoc testing after ANCOVA, the general linear hypotheses test ("glht") supported in R was used in conjunction with the multiple comparisons of means ("mcp") function to test for all possible two-way comparisons via "Tukey" method. To be DELE1-dependent in the RNA-Seq analysis, the gene needed to meet the following conditions:

(1) have an ANCOVA corrected *p*-value < 0.05.

(2) be significant in the stress; Dele1+ vs. control pairwise comparison (post hoc *p*-value < 0.05 and abs(Log2FC > 1))

(3) be significant in the Dele1 KO vs. stress; Dele1+ comparison and

(4) FC is in the opposite direction for (2) and (3).

When individual data have been graphed from these datasets, we have displayed the adjusted FDR levels or *p*-values calculated across the whole dataset, unless otherwise indicated. For quantification from imaging either parametric or non-parametric testing was used depending on the distribution of data in that dataset. In these cases, the statistical testing was performed in GraphPad Prism 10. The specific tests used are indicated in the Figure legends.

Set analysis for metabolomics data was performed using MetaboAnalyst 6.0 (https://www.metaboanalyst.ca/MetaboAnalyst/). Gene enrichment analysis was performed using g:Profiler (https://biit.cs.ut.ee/gprofiler/gost) (Raudvere et al, 2019). In all Figures, *, **, ***, and **** correspond to *p*-values or FDR values of ≤0.05, ≤0.01, ≤0.001, ≤0.0001, respectively. Sample sizes were estimated based on prior experience with transcriptional mitochondrial stress responses in the C10 G58R mice with and without *Oma1* (Shammas et al, 2022). All statistical tests were two-sided.

## Data availability

Microarray and RNA-Seq datasets have been submitted to the NCBI's Gene Expression Omnibus (GEO) Database (accession numbers: GSE273310, GSE271792, GSE273187, GSE273191, GSE273192, GSE273193, GSE273199). Analyzed data is available in Datasets EV1–EV9. The corresponding author will fulfill requests for novel reagents, including transgenic mouse lines, generated for the research article that have not been deposited in a repository. This fulfillment may be subject to Material Transfer Agreement required by the institution and costs related to shipping, testing, and rederivation as needed.

The source data of this paper are collected in the following database record: biostudies:S-SCDT-10_1038-S44318-024-00242-x.

## Peer review information

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

## Acknowledgements

We thank Sandra Lara, Dr. Jung-Hwa Tao-Cheng, and the NINDS EM Facility for technical assistance with TEM. We thank Dr. Abdel Elkahloun and the NHGRI/DIR Microarray Core for technical assistance with RNA expression studies. We thank the NINDS Proteomics Core Facility for the label-free quantitative proteomics data acquisition. We thank Dr. Chengyu Liu and the NHLBI Transgenic Core for assistance in generating transgenic mice. We thank Drs. Pedro M. Quirós and

Carlos Otin Lopez for providing the *OMA1* KO mice. We thank Dr. James J. Faust (Evident) for technical assistance with confocal microscopy. We thank Maureen Kachman and the Michigan Regional Comprehensive Metabolomics Resource Core for help with metabolic experiments of adult C10 G58R mice treated with *Oma1* ASOs. We thank Ionis Pharmaceuticals for providing control and *Oma1*-targeted ASOs that were used in in vivo experiments. We thank Dr. Richard Youle for critical reading of the manuscript and insightful comments. The linked image (https://en.wikipedia.org/wiki/Laboratory_mouse#/media/File:Vector_diagram_ of_laboratory_mouse_(black_and_white).svg) was used in a modified form in some of the Figures and is covered under a CC BY-SA 4.0 license. This work was supported by the Intramural Research Program of the National Institute of Neurological Disorders and Stroke, National Institutes of Health (grant: 1ZIANS003169).

## Author contributions

**Hsin-Pin Lin**: Conceptualization; Data curation; Formal analysis; Supervision; Investigation; Visualization; Methodology; Writing—original draft; Writing—review and editing. **Jennifer D Petersen**: Conceptualization; Data curation; Formal analysis; Supervision; Investigation; Visualization; Methodology; Writing—original draft; Writing—review and editing. **Alexandra J Gilsrud**: Formal analysis; Investigation; Visualization; Methodology; Writing—review and editing. **Angelo Madruga**: Investigation; Visualization; Methodology; Writing—review and editing. **Theresa M D'Silva**: Formal analysis; Investigation. **Xiaoping Huang**: Formal analysis; Investigation; Writing—review and editing. **Mario K Shammas**: Investigation; Writing—review and editing. **Nicholas P Randolph**: Investigation. **Kory R Johnson**: Formal analysis. **Yan Li**: Formal analysis; Investigation; Writing—review and editing. **Drew R Jones**: Formal analysis; Investigation; Writing—review and editing. **Michael E Pacold**: Formal analysis; Investigation; Writing—review and editing. **Derek P Narendra**: Conceptualization; Resources; Data curation; Formal analysis; Supervision; Funding acquisition; Investigation; Visualization; Methodology; Writing—original draft; Writing—review and editing.

Source data underlying figure panels in this paper may have individual authorship assigned. Where available, figure panel/source data authorship is listed in the following database record: biostudies:S-SCDT-10_1038-S44318-024-00242-x.

## Funding

## Disclosure and competing interests statement

The authors declare no competing interests.

# Expanded View Figures

**Figure EV1.   TEM of myocardium and ultrastructural features of mitochondria in C10 G58R on *Dele1*$^{+/-}$ and *Dele1* KO backgrounds.**

(A) Kernel density plots showing distribution of mitochondrial areas for indicated genotypes, measured from TEM images of heart mitochondria. Median values and, in parentheses, interquartile ranges are reported adjacent to curves. $N = 2$ animals per genotype except for *Tfam* mKO; *Dele1* KO, where only 1 animal was available. >600 mitochondria were measured per animal. *P* value was <0.0001. Bar and error bars represent mean and SD, respectively. (B) Bar graph comparing the areas of segmented and non-segmented types of electrolucent mitochondria that were obtained from analysis of C10 G58R animals and littermates in (A). Statistics were performed using Mann–Whitney test, as the data had a non-parametric distribution. **** indicates $p \leq 0.0001$. (C–E) Representative TEM images acquired at 2000× direct magnification show areas of myocardium of indicated genotype used for analysis of mitochondria. Scale bar = 5 μm. (F) Image of the subarea boxed yellow in (D), acquired at 5000× direct magnification and representative of the images used to quantify ultrastructural features of mitochondria detailed in Fig. 3. Scale bar = 2.5 μm. (G) Examples of inclusions observed in C10 G58R mutant mitochondria (black arrows). Scale bar = 500 nm. (H) Examples of two types of electrolucent mitochondria characterized by an enlarged matrix area absent of electron-dense substance and fewer cristae. (Top) A uniformly electrolucent mitochondrion. (Bottom) a segmented mitochondrion that has an electrolucent part (white arrow) separated from a portion of normal-looking matrix and cristae by a cut-through cristae. Open black arrowhead indicates the junction between electrolucent and normal portions of the segmented mitochondria. Scale bar = 500 nm. Please note: mitochondrion in (H, bottom) also appears in bottom right corner of image (J, top). (I) Mitochondria that are fully wrapped by electron-dense phagosome membranes (black arrows). Scale bar = 500 nm. (J) Mitochondria with ruptured OMMs. Open white arrowheads indicate sites where the intact IMM is visible, but OMM is absent. Scale bar = 500 nm. Please note: mitochondrion in image (H, bottom) also appears in the lower right corner of (J, top). Source data are available online for this figure.

     

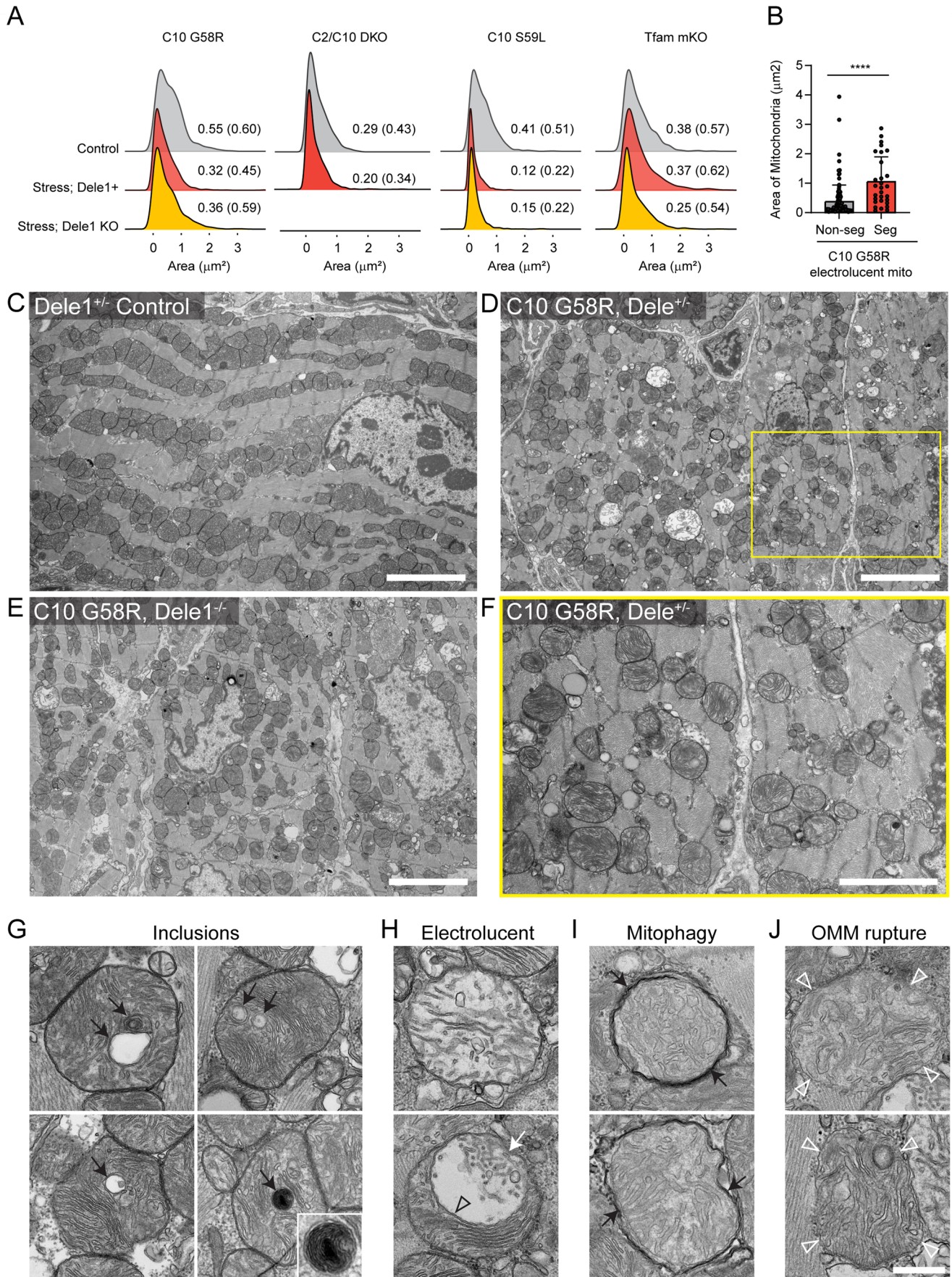

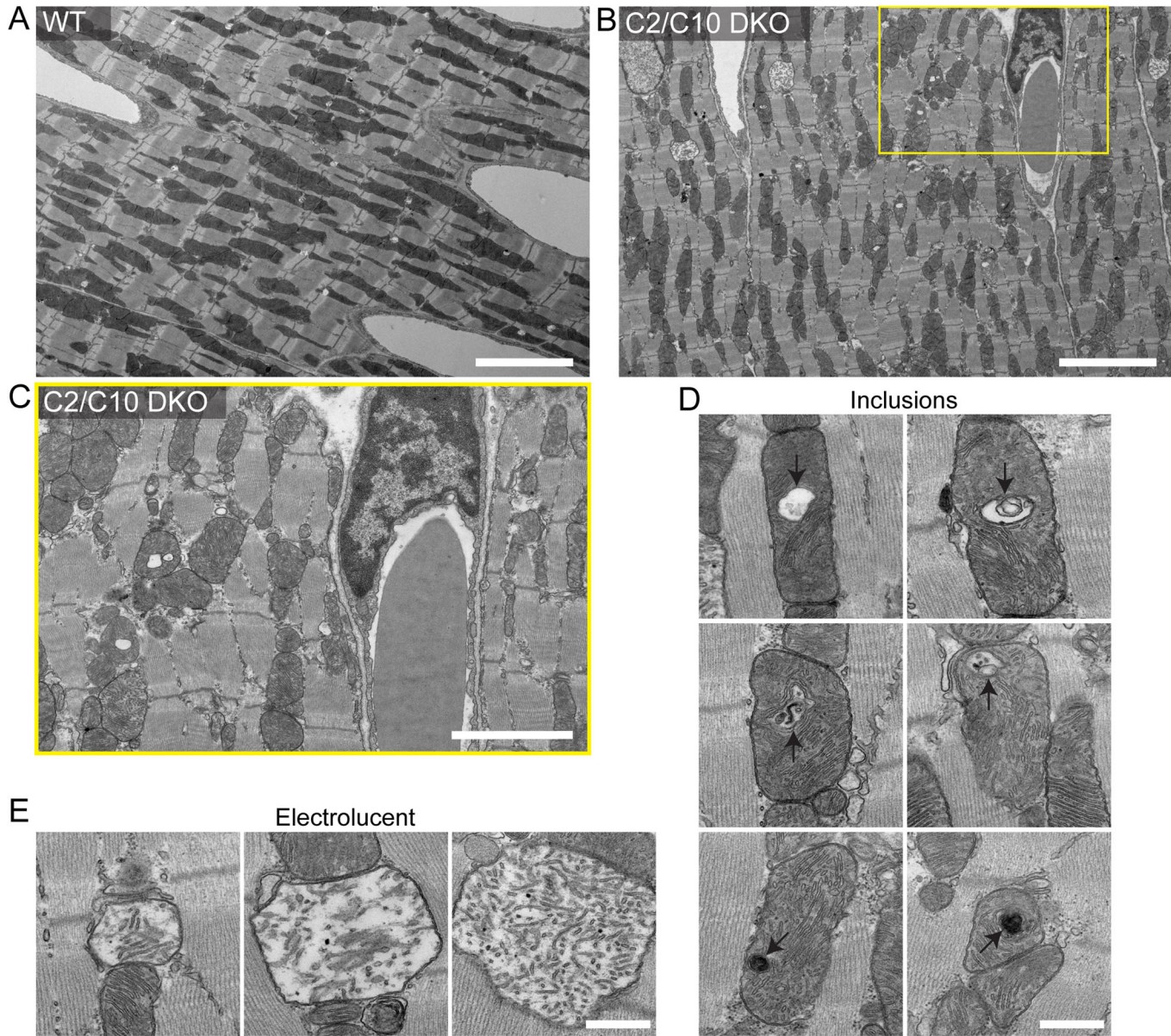

**Figure EV2. TEM of myocardium and ultrastructural features of mitochondria in C2/C10 DKO.**

(A, B) Representative TEM images acquired at 2000× direct magnification show areas of myocardium of indicated genotype used for analysis of mitochondria. Scale bar = 5 μm. (C) Image of the subarea boxed yellow in (B), acquired at 5000× direct magnification and representative of the images used to quantify ultrastructural features of mitochondria detailed in Fig. 3. Scale bar = 2.5 μm. (D) Examples of inclusions observed in C2/C10 DKO mitochondria (black arrows). Scale bar = 500 nm. (E) Examples of electrolucent mitochondria characterized by an enlarged matrix area absent of electron-dense substance and fewer cristae. Scale bar = 500 nm. Source data are available online for this figure.

 

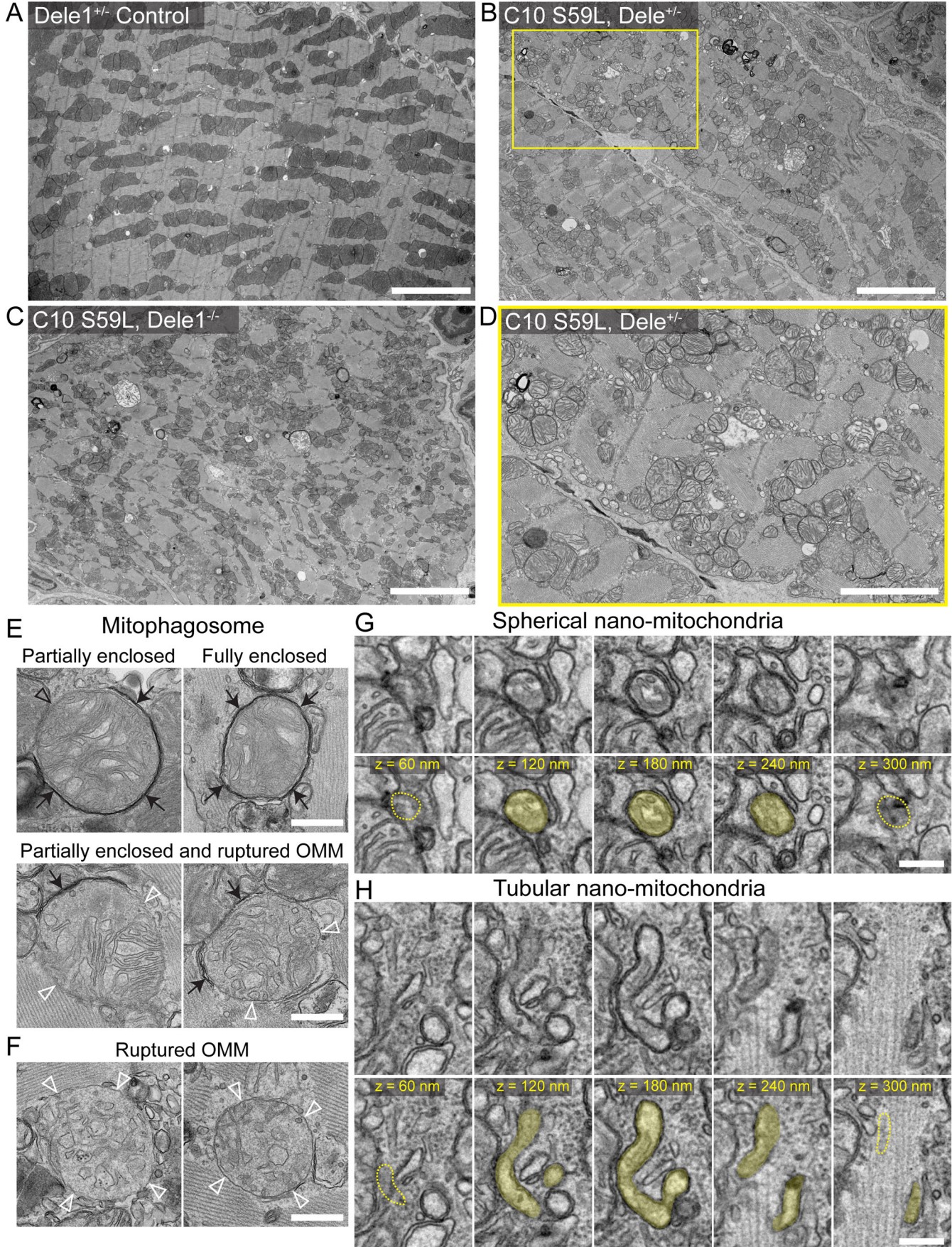

**Figure EV3. TEM of myocardium and ultrastructural features of mitochondria in C10 S59L on Dele1+/− and Dele1 KO backgrounds.**

(A–C) Representative TEM images acquired at 2000× direct magnification show areas of myocardium of indicated genotype used for analysis of mitochondria. Scale bar = 5 µm. (D) Image of the subarea boxed yellow in (B), acquired at 5000× direct magnification and representative of images used to quantify ultrastructural features of mitochondria detailed in Fig. 3. Scale bar = 2.5 µm. (E) Examples of mitochondria that are partially or fully enclosed by electron-dense phagosome membranes (black arrows). Open black arrowhead indicates a portion of the mitochondria that is not enclosed. Partially enclosed mitochondria with ruptured OMMs were also observed. Open white arrowheads indicate sites where the intact IMM is visible, but the OMM is absent. Scale bar = 500 nm. (F) Examples of mitochondria with ruptured OMMs. Open white arrowheads indicate sites where the intact IMM is visible, but an OMM is absent. Scale bar = 500 nm. (G) Serial sections through a 250 nm diameter mitochondrion show that it is a spherical nano-mitochondrion spanning fewer than five 60-nm sections (< 300 nm in (Z)). Top row shows the five serial sections without colorization, bottom row shows the same serial sections with the nano-mitochondrion shaded yellow. Yellow dotted lines indicate absence of the mitochondrion in neighboring serial sections. Scale bar = 200 nm. (H) Five serial sections of 60-nm thickness show a 100 nm-wide tubule-shaped mitochondrion. Top row shows five serial sections through the tubular nano-mitochondrion, bottom row shows the same serial sections with the tubular nano-mitochondrion shaded yellow. The yellow dotted lines indicate absence of the mitochondrion in the neighboring section. Scale bar = 200 nm. Source data are available online for this figure.

 

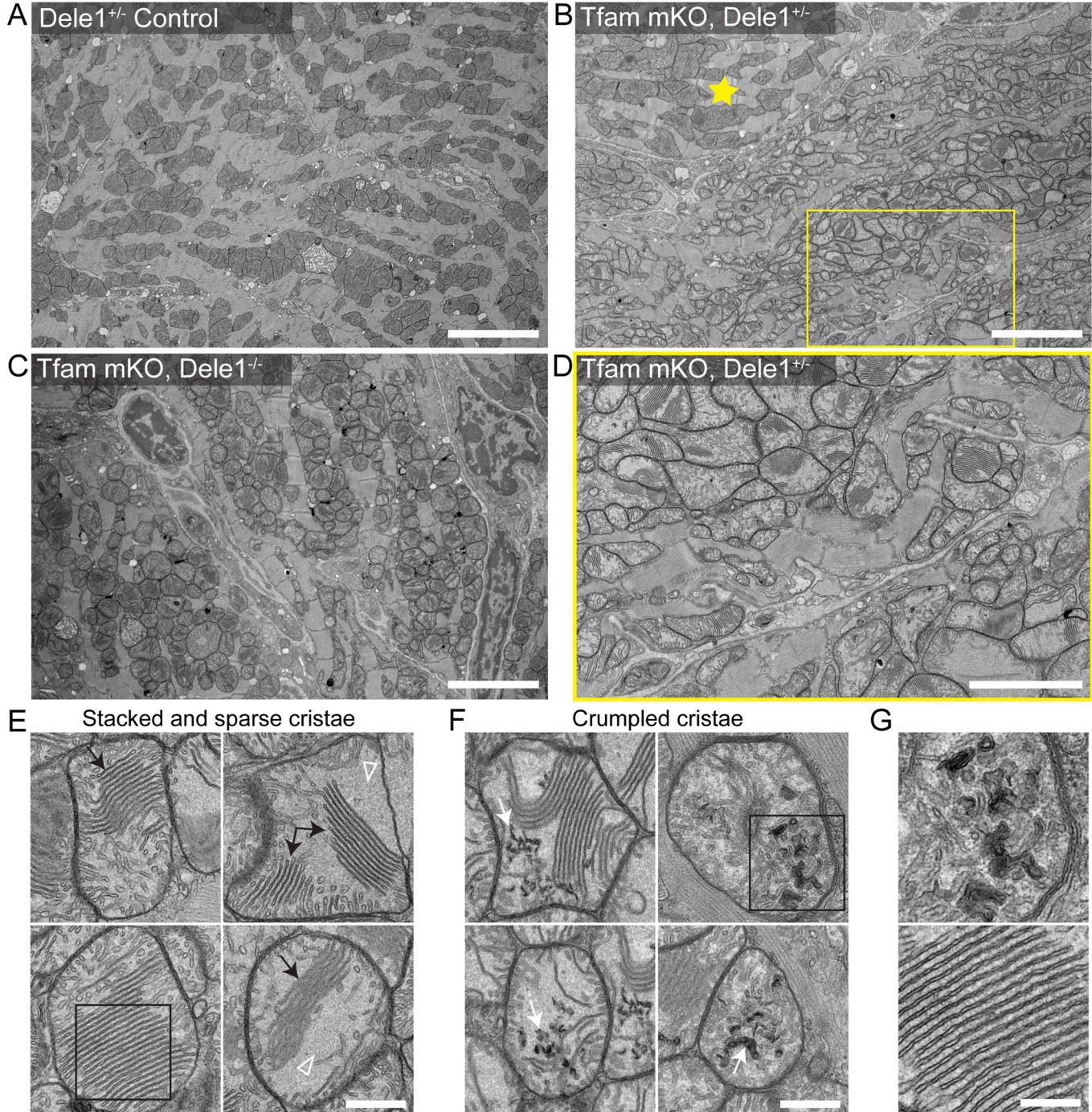

**Figure EV4.  TEM of myocardium and ultrastructural features of mitochondria in *Tfam* mKO on *Dele1*+/− and *Dele1* KO backgrounds.**

(A–C) Representative TEM images acquired at 2000× direct magnification show areas of myocardium of indicated genotype used for analysis of mitochondria. Yellow star in (B) indicates a myocyte with milder structural phenotype compared to neighboring myocytes, illustrating the observed mosaicism of the phenotype. Scale bar = 5 μm. (D) Image of the subarea boxed yellow in (B), acquired at 5000× direct magnification and representative of images used to quantify ultrastructural features of mitochondria detailed in Fig. 3. Scale bar = 2.5 μm. (E) *Tfam* mKO mitochondria displayed populations of closely aligned "stacked" cristae (black arrows) and sparse areas filled with a granular matrix material and few cristae (open white arrowheads). Scale bar = 500 nm. The mitochondrion in the bottom left image is cropped from same cell as the mitochondrion shown in (Fig. 3F, middle). At least some Images of mitochondria are taken from the same cell. (F) Examples of crumpled cristae (white arrows) that occurred in *Tfam* mKO mitochondria. Scale bar = 500 nm. At least some Images of mitochondria are taken from the same cell. (G) This panel shows enlargements of images that are also shown in (Fig. EV4E, boxed area with "stacked cristae") and (Fig. EV4F, boxed area with "crumpled cristae"). Scale bar = 250 nm. Source data are available online for this figure.

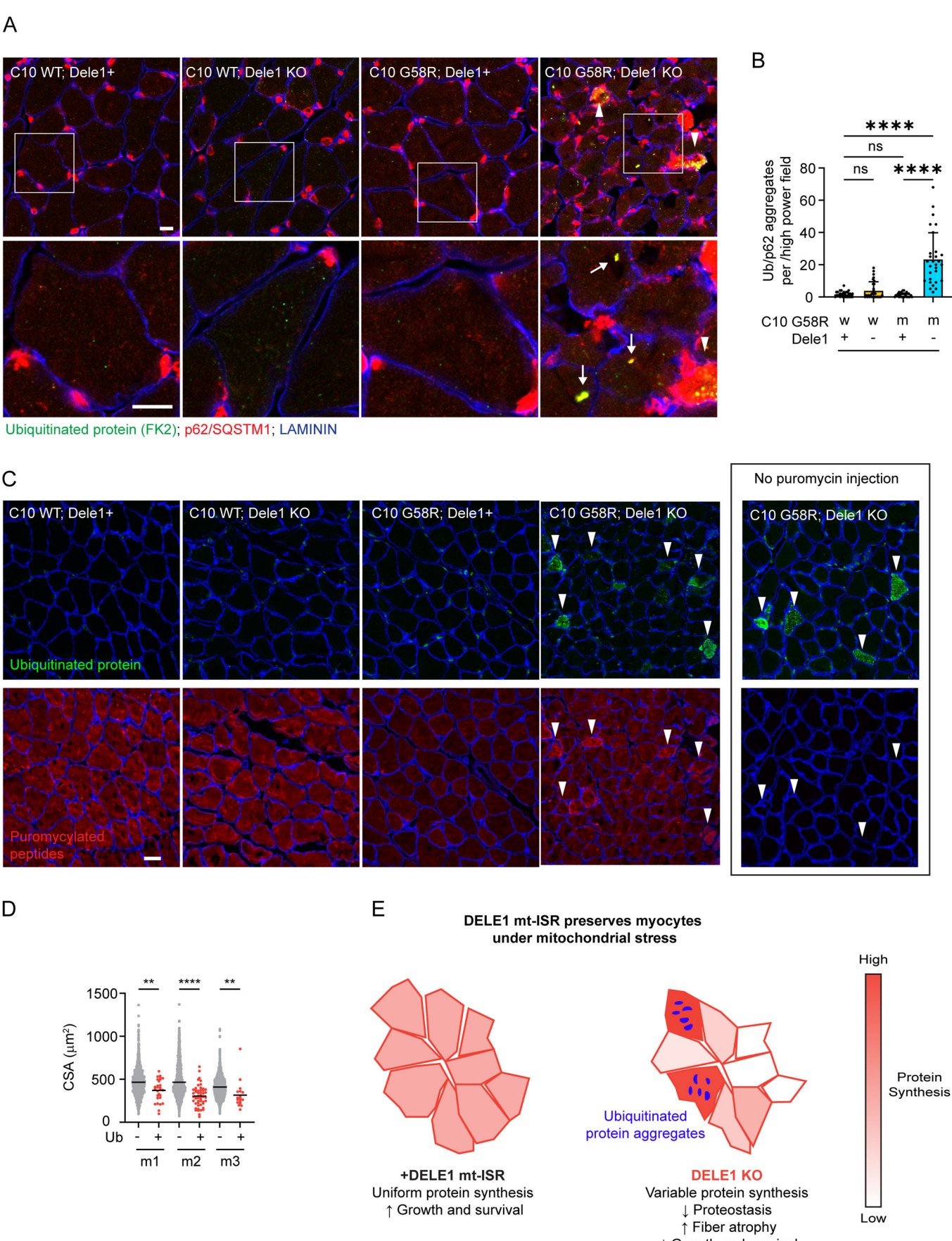

A

Ubiquitinated protein (FK2); p62/SQSTM1; LAMININ

B

C

No puromycin injection

D

E

DELE1 mt-ISR preserves myocytes
under mitochondrial stress

+DELE1 mt-ISR
Uniform protein synthesis
↑ Growth and survival

DELE1 KO
Variable protein synthesis
↓ Proteostasis
↑ Fiber atrophy
↓ Growth and survival

Ubiquitinated
protein aggregates

◀

**Figure EV5. The *Dele1* mt-ISR prevents disruptions in translation-associated proteostasis in skeletal muscle.**

(A) Representative immunofluorescence images of gastrocnemius muscle from C10 G58R; *Dele1* KO and indicated littermates triple-stained for the ubiquitinated protein marker, FK2 (green), the aggregate adapter protein p62 (red) and laminin (blue). In the C10 G58R; *Dele1* KO genotype, a subset of fibers displayed FK2 and p62 colocalized in individual (white arrows) or confluent aggregates (arrowheads), suggesting proteostatic collapse. High power (60X) views of the boxed areas are shown in bottom panels. Scale bars = 10 μm. Note: animals were not injected with puromycin in this experiment. (B) Quantification of (A). Aggregates positive for both FK2 and p62 were counted in 10 high power (60X) fields. $N = 3$ mice per genotype with 29 or 30 fields counted total per sample. High-power (60X) field size is 132.58 μm × 132.58 μm. Statistical analysis was performed using the Kruskal–Wallis test with Dunn's multiple comparisons test, as the data distribution was non-parametric. **** indicates $p \leq 0.0001$ and "ns" not significant. Adjusted *p*-values are >0.9999 for WT vs. *Dele1* KO, >0.9999 for WT vs. C10 G58R, <0.0001 for WT vs. C10 G58R; *Dele1* KO, and <0.0001 for C10 G58R vs. C10 G58R; *Dele1* KO. (C) Representative immunofluorescence images of gastrocnemius muscle from P28 C10 G58R; *Dele1* KO mice and indicated littermates injected with puromycin 30 min prior to sacrifice as in (Fig. 7E). Muscle cross-sections were immunostained for the ubiquitinated protein marker, FK2 (green), puromycin (red), and laminin (blue). Arrowheads indicate muscle fibers containing many or confluent aggregates of ubiquitinated protein that were also co-stained for elevated puromycylated polypeptides. $N = 1$ mouse for each genotype except for P28 C10 G58R; *Dele1* KO mice for which $N = 3$ mice. Scale bars = 20 μm. (D) Quantification of myofiber cross-sectional area (CSA) as in (Fig. 7E). The average CSA for Ub+ and Ub- myofiber is shown in graph separately for three G58R; *Dele1* KO mice (m1–3) in graph. $N = 3$ mice with 10 high power fields counted per mouse. Statistical analysis was performed using the Kruskal–Wallis test with Dunn's multiple comparisons test, as the data distribution was non-parametric. ** and **** indicates $p \leq 0.01$ and 0.0001, respectively, and "ns" not significant. *P*-values (from left to right) are 0.0055, <0.0001, and 0.0016. (E) Schematic summarizing data showing that *Dele1* KO results in variable protein synthesis, decreased proteostasis, and increased muscle fiber atrophy within skeletal muscle undergoing mitochondrial stress. We hypothesize that these changes are responsible for the decreased growth and survival in MM models with early mitochondrial stress, in the absence of the *Dele1* mt-ISR.

 