## [Peer Review File · The EMBO Journal]

DELE1 maintains muscle proteostasis to promote growth and survival in mitochondrial myopathy

Hsin-Pin Lin, Jennifer Petersen, Alex Gilsrud, Angelo Madruga, Theresa D'Silva, Xioping Huang, Mario Shammas, Nicholas Randolph, Kory Johnson, Yan Li, Drew Jones, Michael Pacold, and Derek Narendra

Corresponding author(s): Derek Narendra (derek.narendra@nih.gov)

Review Timeline:

Submission Date:	27th Feb 24
Editorial Decision:	28th Mar 24
Revision Received:	8th Jul 24
Editorial Decision:	5th Aug 24
Revision Received:	11th Aug 24
Accepted:	22nd Aug 24

Editor: William Teale

Transaction Report:

Dear Derek,

Thank you again for the submission of your manuscript entitled "DELE1 promotes translation-associated homeostasis, growth, and survival in mitochondrial myopathy" and for your patience during the review process. Your manuscript was sent to three referees; we have now received reports from all of them, which I copy below.

As you can see from their comments, while the referees need a more carefully framed context for the data you present (that will require the inclusion of some additional experiments), all point out the potential value of your work to the scientific community.

Based on the overall interest expressed in the reports, therefore, I would like to invite you to address the comments of all referees in a revised version of the manuscript. I should add that it is The EMBO Journal policy to allow only a single major round of revision and that it is therefore important to resolve the main concerns at this stage. I believe the concerns of the referees are reasonable and addressable, but please contact me if you have any questions, need further input on the referee comments or if you anticipate any problems in addressing any of their points. I am happy to arrange a Zoom call once you have had a chance to digest the reports. Please, follow the instructions below when preparing your manuscript for resubmission.

I would also like to point out that as a matter of policy, competing manuscripts published during this period will not be taken into consideration in our assessment of the novelty presented by your study ("scooping" protection). We have extended this 'scooping protection policy' beyond the usual 3 month revision timeline to cover the period required for a full revision to address the essential experimental issues. Please contact me if you see a paper with related content published elsewhere to discuss the appropriate course of action.

Again, please contact me at any time during revision if you need any help or have further questions.

Thank you very much again for the opportunity to consider your work for publication. I look forward to your revision.

Best regards,

William

William Teale, Ph.D.
Editor
The EMBO Journal

When submitting your revised manuscript, please carefully review the instructions below and include the following items:

- 1) a .docx formatted version of the manuscript text (including legends for main figures, EV figures and tables). Please make sure that the changes are highlighted to be clearly visible.
- 2) individual production quality figure files as .eps, .tif, .jpg (one file per figure).
- 3) a .docx formatted letter INCLUDING the reviewers' reports and your detailed point-by-point response to their comments. As part of the EMBO Press transparent editorial process, the point-by-point response is part of the Review Process File (RPF), which will be published alongside your paper.
- 4) a complete author checklist, which you can download from our author guidelines ([https://wol-prod-cdn.literatumonline.com/pb-assets/embo-site/Author Checklist%20-%20EMBO%20J-1561436015657.xlsx](https://wol-prod-cdn.literatumonline.com/pb-assets/embo-site/Author%20Checklist%20-%20EMBO%20J-1561436015657.xlsx)). Please insert information in the checklist that is also reflected in the manuscript. The completed author checklist will also be part of the RPF.
- 5) Please note that all corresponding authors are required to supply an ORCID ID for their name upon submission of a revised manuscript.
- 6) We require a 'Data Availability' section after the Materials and Methods. Before submitting your revision, primary datasets produced in this study need to be deposited in an appropriate public database, and the accession numbers and database listed under 'Data Availability'. Please remember to provide a reviewer password if the datasets are not yet public (see <https://www.embopress.org/page/journal/14602075/authorguide#datadeposition>). If no data deposition in external databases is

needed for this paper, please then state in this section: This study includes no data deposited in external repositories. Note that the Data Availability Section is restricted to new primary data that are part of this study.

Note - All links should resolve to a page where the data can be accessed.

8) For data quantification: please specify the name of the statistical test used to generate error bars and P values, the number (n) of independent experiments (specify technical or biological replicates) underlying each data point and the test used to calculate p-values in each figure legend. The figure legends should contain a basic description of n, P and the test applied. Graphs must include a description of the bars and the error bars (s.d., s.e.m.).

9) We would also encourage you to include the source data for figure panels that show essential data. Numerical data can be provided as individual .xls or .csv files (including a tab describing the data). For 'blots' or microscopy, uncropped images should be submitted (using a zip archive or a single pdf per main figure if multiple images need to be supplied for one panel). Additional information on source data and instruction on how to label the files are available at .

10) We replaced Supplementary Information with Expanded View (EV) Figures and Tables that are collapsible/expandable online (see examples in <https://www.embopress.org/doi/10.15252/embj.201695874>). A maximum of 5 EV Figures can be typeset. EV Figures should be cited as 'Figure EV1, Figure EV2" etc. in the text and their respective legends should be included in the main text after the legends of regular figures.

12) Our journal encourages inclusion of *data citations in the reference list* to directly cite datasets that were re-used and obtained from public databases. Data citations in the article text are distinct from normal bibliographical citations and should directly link to the database records from which the data can be accessed. In the main text, data citations are formatted as follows: "Data ref: Smith et al, 2001" or "Data ref: NCBI Sequence Read Archive PRJNA342805, 2017". In the Reference list, data citations must be labeled with "[DATASET]". A data reference must provide the database name, accession number/identifiers and a resolvable link to the landing page from which the data can be accessed at the end of the reference. Further instructions are available at .

Additional instructions for preparing your revised manuscript:

At EMBO Press we ask authors to provide source data for the main manuscript figures. Our source data coordinator will contact you to discuss which figure panels we would need source data for and will also provide you with helpful tips on how to upload

and organize the files.

We realize that it is difficult to revise to a specific deadline. In the interest of protecting the conceptual advance provided by the work, we recommend a revision within 3 months (26th Jun 2024). Please discuss the revision progress ahead of this time with the editor if you require more time to complete the revisions. Use the link below to submit your revision:

Referee #1:

The manuscript by Lin et al. implicates the DELE1 pathway in a generalizable mitochondrial integrated stress response. Using several transgenic mouse models, the authors provide a comprehensive report of the mt-ISR in both cardiac and skeletal muscle mitochondrial myopathies. The manuscript is well-written, highly informative, and will make a meaningful contribution to the literature.

General comments:

- 1) What measures were taken to verify embryonic (constitutive) DELE1 knockout does not induce any compensatory adaptation that might otherwise confound the data reported here?
- 2) While highly insightful, the findings reported here do not offer much clinical relevance. How might targeting the DELE1 pathway be useful at improve outcome measures in mitochondrial myopathies?
- 3) The authors report activation of mt-ISR is a protective mechanism during mitochondrial dysfunction. However, data reported here show DELE1 activation does not majorly affect OXPHOS or mitochondrial structure. Is mt-ISR activation temporary, and does it restore homeostasis in the absence of direct effects on OXPHOS and mitochondrial structure? Can chronically activated mt-ISR lead to unfavorable cellular conditions?
- 4) Increased 3-phosphoserine, and DELE1-dependent amino acid synthesis is highly descriptive and no causal effect of these metabolites are shown in the mt-ISR response.

Specific comments:

- 1) The authors studied mice, and specimens thereof, from both sexes but no major sex-dependent readouts were reported. Does biological sex differentially influence the DELE1 mt-ISR response?
- 2) Does nutritional support influence metabolomics reported here?
- 3) Not all figure legends describe statistical tests used. And did the data follow a parametric or non-parametric distribution?

Referee #2:

In this paper Lin et al. perform an in vivo characterization of DELE1 knock out model in the context of mitochondrial stress. OMA1-DELE1 signaling cascade elicits a protective mitochondrial integrated stress response (mt-ISR) by activating a specific gene signature. The impact of DELE1 mtISR has been characterized in different in vivo models of mitochondrial myopathies. In the last part of the manuscript, the authors highlight the metabolic impact of DELE1 mediated mt-ISR on tissue metabolism at the level of heart and skeletal muscle.

Although the findings are very interesting, the paper has an overall confused structure that should be amended. Also, some

minor experimental points shall be addressed.

The DELE1 total knockout (KO) model does not exhibit any gross phenotypic alterations. However, under stress conditions, such as cold exposure, the authors demonstrate a blockage of the mitochondrial integrated stress response (mt-ISR) activation in brown adipose tissue (BAT) (refer to Fig. 1H, I, and J).

In Fig. 1E, a transmission electron microscopy (TEM) image of cardiac mitochondria is presented. While no differences in the area or aspect ratios are reported (see Fig. 1F), the image shows enlarged mitochondria and the presence of small, electron-transparent mitochondria in DELE1 KO hearts. The authors are requested to comment on these observations.

If the only stress tested in this mouse model was cold exposure in BAT, it is misleading to present TEM data on mitochondrial status in the heart while placing BAT analysis in Supplementary Fig. 1. To establish robust evidence of whether mitochondria undergo structural alterations under stress conditions in the DELE1 KO model, BAT mitochondria should be reported and measured. Specifically, do they change under room temperature (RT) or cold exposure conditions?

In Fig. 2, the authors explore the effect of DELE1 deletion in the context of concomitant mitochondrial-related myopathy or cardiomyopathy. DELE1 KO mice were crossed with models of downregulation of CHCHD2/10 and TFAM, as well as up-regulated dominant negative mutants of CHCHD10 protein. The worsening in lifespan and mouse activity indicates that blocking mt-ISR in vivo has detrimental effects in the presence of compromised mitochondrial fitness.

The biochemical analysis in Fig. 2M is related to cardiac tissue. The cardiac mitochondrial phenotype (Fig. 1E) should be relocated in this context, addressing the concerns mentioned earlier.

To dissect the role of OMA1 and DELE1 in mt-ISR response activation (Fig. 3), the authors bred the two mouse strains with the C10G58R mutant. OMA1 gene deletion has an additive effect on the lifespan of mice compared to DELE1 KO alone in the context of unhealthy mitochondria. Crossing with OPA1 Δ s1/ Δ s1 mice excludes the concomitant effect of the other main substrate of OMA1 protease. This experiment does not directly address whether blocking OPA1 cleavage is responsible for the additional effect due to OMA1 deletion. How is the mt-ISR in these mice? The analysis of the expression of MTHFD2, mirroring that performed in the OMA1 knockout mice, is a pivotal experiment. The data presented in Supplementary Fig. 2, Panel I, is crucial for substantiating the direct influence of DELE1 in activating this response. MTHFD2 activation is observed in C10 G58R/OMA1KO mice but not in C10 G58T/DELE1 KO mice. However, only one sample of C10 G58R/DELE1KO mice has been presented. To enhance the robustness of the analysis, additional samples with this genotype should be included, and improvements to the OMA1 western blot are warranted.

The experiment supports the transcriptomic profile in Fig. 3E, where only DELE1 downregulation is involved in the inactivation of mt-ISR gene profile, without affecting the level of oxidative phosphorylation (OXPHOS) content or oxygen consumption rate.

In Fig. 4, the authors examine mitochondrial ultrastructure in different mutant/DELE1KO mice, encountering various ultrastructure abnormalities that were only partially and differently exacerbated by the absence of DELE1. There are no common features underlying a general effect of DELE1 downregulation. However, it would be interesting to check if in OMA1KO mice crossed with OPA1 Δ s1/ Δ s1 mice there is no alteration of cristae ultrastructure, providing clarity that DELE1 downregulation alone does not impact the cristae alteration in OMA1KO mice.

The transcriptome signature performed in mitochondria from compromised hearts highlights that in the absence of the DELE1 gene, the mt-ISR response is silenced (Fig. 5).

Figure 2S panel I: the quality of the OMA1 western blot should be improved.

Referee #3:

general summary and opinion about the principle significance of the study, its questions and findings

Lin and colleagues report striking phenotypes that are interesting in the context of mt-ISR regulation. However, the sheer density of data and multiple narratives obstruct its clarity, making the manuscript difficult to read at times. The overall flow is disjointed, and the manuscript requires greater streamlining to be more accessible. The authors show that genetic ablation of DELE1 does not affect normal physiology, but does have physiological relevance during specific contexts of extreme mitochondrial stress. However, the lack of conceptual clarity makes it challenging to distinguish the novelty of this manuscript from their 2022 paper.

specific major concerns essential to be addressed to support the conclusions

The manuscript presents numerous paradigms, but they are only superficially detailed. The transitions between models and approaches are not clearly articulated. For instance, the authors start with BAT, but the manuscript is clearly muscle-focused, and the switch to muscle is very abrupt. The authors need to better craft this switch and thread the most important results together with solid rationales for the experimental paradigms chosen. Furthermore, the statistical descriptions lack sufficient detail. To ensure a clear interpretation, it is necessary to include effect sizes and appropriate tests.

A major concern is the repeated primary assertion that 'the mt-ISR has not been examined in skeletal muscle'. However, this statement is incorrect as the group of Anu Suomalainen originally profiled stage-wise mt-ISR regulation in disease models for over a decade. In a recent 2019 paper published in *Cell Metabolism*, Forsström et al examined the mt-ISR in mitochondrial myopathy. They reported that FGF21 is a local and systemic signal of mitochondrial stress in both preclinical mouse and human models. The authors previously cited this work in their 2022 paper, Shamma et al, but did not do so in this present submission. It is contentious to claim such novelty in light of the pre-existing body of literature. This omission is a very unfortunate oversight. If this manuscript is revised, a significant rewrite with greater nuance, precision, and appropriate attribution is needed. The idea of distinct mitochondrial defects triggering the mt-ISR has been around for some time now. See work by the Mootha group (Mick et al. 2020). Taken together, although a lot of work is collated in this submission, it is difficult to see the conceptual novelty of this paper as a whole.

The survival phenotypes are compelling. To what extent are these a function of developmental maturation? The authors should discuss and speculate whether conditional *Dele1* inactivation in later life could induce pathology. Much more detail is also required on the genetic background of the mice. How has the breeding been performed? This needs to be explicitly described in the methods. How does the genetic background impact the survival phenotypes described?

The overall phenotyping focuses on the survival effects, which the reviewer acknowledges are very nice - but without drilling down into the in vivo pathomechanisms at play. Striated muscle is not homogeneous, and details on the muscle fiber types affected are lacking from the manuscript. Does *Dele1* protect integrity of specific muscle fiber types, or all? This question is important and essential to understand the protective mechanistic functions of *DELE1* in vivo, given the fiber-specific mitochondrial features that mice and humans exhibit. Put simply, the authors should analyse their muscle samples to assess if ablation of *Dele1* accelerates the demise of specific fiber types.

The RNA data is derived from microarray analysis, which is still a valid method for some clinical applications and has greater sensitivity than RT-qPCR. However, its dynamic range is inferior to that of whole genome transcriptomics. This method is less sensitive than more widely used contemporary approaches such as RNA-seq/whole genome transcriptomics, and is not aligned with the current state-of-the-art in the field. The authors' rationale for selecting this method is unclear. Ideally, these findings should be authenticated using RNA-seq/whole genome transcriptomics. To gain more insight and profile the RNA-seq data, Ingenuity, gProfiler or similar pipelines could be used. Additionally, it is important to identify the promoter motifs common to differentially modified transcripts overlapping with the three mouse models. This can be easily assessed by MEME-ChIP analysis of RNA-seq data. Additionally, providing greater detail on metabolomics methodology, including metabolite extraction and quantification standards, would enhance the reproducibility and interpretation of the data.

minor concerns that should be addressed

Including a detailed limitations section would be prudent, outlining inherent caveats in the findings and delineating avenues for further exploration.

We thank the reviewers for the substantial time they invested in reviewing our manuscript, and their thoughtful suggestions on how to make it stronger. We have sought to address all the reviewers concerns and suggestions often with new data, extensive rewriting of the Introduction and Discussion, and restructuring of the narrative, within the time and scope typically afforded a revision for The EMBO Journal. (Note: we added bolding to text below to help clarify the concerns guiding our response.)

Referee #1:

The manuscript by Lin et al. implicates the DELE1 pathway in a generalizable mitochondrial integrated stress repose. Using several transgenic mouse models, the authors provide a comprehensive report of the mt-ISR in both cardiac and skeletal muscle mitochondrial myopathies. The manuscript is well-written, highly informative, and will make a meaningful contribution to the literature.

General comments:

1) What measures were taken to verify embryonic (constitutive) DELE1 knockout does not induce any compensatory adaptation that might otherwise confound the data reported here?

This is an excellent question. We have added to the discussion this caveat:

“A limitation of our study is the use of a constitutive (germline) *Dele1* KO allele, which may allow for compensation during development. Arguing against developmental compensation, however, the unstressed *Dele1* KO did not exhibit any differences in its transcriptome in the tissues assayed (excepting decreased *Dele1* in some datasets) (Fig. S9D and S12B). Similarly, heart mitochondria appeared to be unaltered both in terms of the expression of OXPHOS complex subunits and mitochondrial ultrastructure (Fig. S1D). Additionally, eIF2 α failed to be phosphorylated in the absence of *Dele1*, despite mitochondrial stress (Fig. S15C), indicating that HRI and the three other eIF2 α kinases are not activated by mitochondrial stress in the absence of *Dele1*. Consistently, the majority of ATF4 targeted transcripts that were upregulated by mitochondrial stress were silenced by *Dele1* KO (Fig. 4B). Thus, it seems that activation of the ISR in striated muscle by mitochondrial stress is heavily dependent on DELE1 signaling.”

We are planning future studies that involve conditional knockout of DELE1 in adult animals that may address this point experimentally, but we feel this is beyond the scope of the present manuscript.

2) While highly insightful, the findings reported here do not offer much clinical relevance. How might targeting the DELE1 pathway be useful at improve outcome measures in mitochondrial myopathies?

This is an important question that, as the Reviewer suggests, we could address more clearly in the manuscript. We have done so in the Discussion of the revised manuscript:

“Our findings have several potentially important clinical implications for the treatment of MM [mitochondrial myopathy] and MC [mitochondrial cardiomyopathy]. First, they suggest that inhibiting the ISR in most cases of MM will likely be harmful, as it appears to be adaptive in response to diverse mitochondrial stresses. This contrasts with other conditions like Vanishing White Matter Disease, where ISR inhibition is being actively pursued as a therapeutic strategy (Wong et al, 2019). Second, our work

establishes that common biomarkers of MM, such as FGF21 (Lehtonen et al, 2016), likely reflect signaling through the OMA1-DELE1 pathway. Understanding the mechanism for biomarker regulation is important for interpreting how they are affected by disease progression and their response to therapies for MM. Finally, our work raises the possibility that enhancing the mt-ISR may be protective. Notably, in many of the models we tested, the mt-ISR is activated later in the disease course. Future work might test whether therapeutically activating the ISR early in the disease course, in anticipation of progression, or above physiologic levels may be beneficial. Several pharmacological activators of the ISR have been described, including halofuginone (targeting EPRS to increase eIF2 α phosphorylation by GCN2) and Sephin1 (inhibiting an eIF2 α phosphatase) (Das et al, 2015; Keller et al, 2012), the latter of which was recently shown to augment the DELE1 mt-ISR in an AFG3L2 mutant model (Franchino et al, 2024).”

3) The authors proposit activation of mt-ISR is a protective mechanism during mitochondrial dysfunction. However, data reported here show DELE1 activation does not majorly affect OXPHOS or mitochondrial structure. **Is mt-ISR activation temporary, and does it restore homeostasis in the absence of direct effects on OXPHOS and mitochondrial structure? Can chronically activated mt-ISR lead to unfavorable cellular conditions?**

These are important questions that we now address in the Discussion of the revised manuscript:

“Our comparison of these four models also suggests that the mt-ISR likely persists as long as the mitochondrial stress remains. Consistently, we previously found that C10 G58R mice, which have onset of the DELE1 mt-ISR prior to P28 (Fig. 1M and N), still had persistent activation of the mt-ISR at around 1 year of age (Shammas et al, 2022). We speculate that if the underlying mitochondrial stress were relieved, the DELE1 mt-ISR would subside, as OMA1 activation can be reversed within minutes, at least in cultured cells (Murata et al, 2020). The persistence of the DELE1 mt-ISR over months and perhaps years also raises the question of whether it remains adaptive in the chronic setting. As the DELE1 mt-ISR was found to be protective also in models with late-onset mitochondrial stress, such as the C10 S59L model, it seems the DELE1 mt-ISR can be protective on the chronic time scale (as in the C10 S59L model) as well as the subacute time scale (as in the C10 G58R and C2/C10 DKO models).”

4) Increased 3-phosphoserine, and DELE1-dependent amino acid synthesis is highly descriptive and no causal effect of these metabolites are shown in the mt-ISR response.

Our interpretation of metabolomics data follows the same logic as our transcriptomic and proteomic data, but we agree with the reviewer that it is more tenuous to assert causality in the case of metabolomics, given the steady-state level of a given metabolite is often influenced by fluxes through a complex network of pathways. We have added this caveat to our conclusion of the relevant section of the results, making it clear that the attribution of causality remains speculative. This revised passage reads as follows:

“Considered together, these data suggest that the OMA1-DELE1 mt-ISR may upregulate amino acid levels in the adult heart under mitochondrial stress, including four that are synthesized through pathways that are closely connected to mitochondrial metabolism: proline, asparagine, serine, and glycine. We speculate that the steady-state increase of these four amino acids and 3-PS is driven by the

DELE1 mt-ISR dependent upregulation of the enzymes responsible for their biosynthesis, which we found were also upregulated in a DELE1-dependent manner at the transcript and/or protein level. However, as flux through many different pathways affect levels of 3-PS and these amino acids, this attribution remains speculative. That the upregulation of these amino acids is coordinated with the upregulation of their corresponding amino acid tRNA synthetase (i.e., proline with *Eprs*, asparagine with *Nars*, serine with *Sars*, glycine with *Gars*) additionally raises the possibility that they may be upregulated to maintain adequate levels of tRNA-charged amino acids for protein synthesis (Fig. 4D and 5A).”

Specific comments:

1) The authors studied mice, and specimens thereof, from both sexes but no major sex-dependent readouts were reported. **Does biological sex differentially influence the DELE1 mt-ISR response?**

This is an excellent question. To address this as well as a concern raised by Reviewer 3, we have performed heart and gastrocnemius transcriptomics using RNA-Seq on larger groups of CHCHD10 G58R mice, including roughly equal numbers of each biological sex. We find both sexes show the same pattern of gene expression changes across genotypes for DELE1-dependent DEGs. This is now shown for individual animals divided by sex in Fig. S13 and S14. The data from heart samples are reproduced here:

DELE1 dependent DEGs common to gastroc, tibalis anterior, and heart

C10 G58R Dele1	Female Hearts												Male Hearts												Log2 FC		
	wt				m				wt				m				wt				m						
	F1	F2	F3	F4	F1	F2	F3	F4	F1	F2	F3	F4	M1	M2	M3	M4	M1	M2	M3	M4	M1	M2	M3	M4		M1	M2
Mthfd2	-0.0	0.1	0.1	-0.1	0.1	-0.1	-0.1	6.2	6.2	6.4	3.0	2.2	2.2	-0.0	-0.0	0.1	0.3	0.3	6.1	6.5	6.0	3.1	1.5	-0.0			
Asns	-0.1	0.6	-0.2	-0.2	0.0	0.1	-0.1	5.2	5.0	5.4	2.4	1.5	1.6	-0.2	-0.1	0.0	-0.2	-0.2	5.0	5.9	5.1	2.3	0.9	0.0			
Trib3	0.1	-0.2	-0.0	0.0	0.2	0.4	-0.2	5.4	5.0	5.3	1.7	1.8	1.2	-0.3	0.3	-0.1	0.1	-0.2	5.0	4.7	4.5	1.4	0.9	2.2			
Psat1	0.0	0.0	-0.0	-0.2	0.2	0.3	0.1	3.9	3.3	3.6	3.0	1.3	1.6	0.2	-0.1	0.5	0.4	0.1	3.3	4.1	2.9	2.4	0.7	-0.0			
Slc7a5	-0.2	0.2	0.0	0.1	0.1	0.1	-0.2	4.8	4.4	5.0	2.0	1.7	1.7	-0.2	-0.1	0.2	0.4	-0.0	4.2	3.9	4.2	1.8	1.3	1.5			
Phgdh	0.2	0.1	-0.1	0.0	0.1	0.5	-0.2	5.3	5.2	5.4	0.6	0.2	0.5	-0.2	-0.0	-0.1	-0.2	-0.1	5.2	5.1	4.5	0.7	0.0	-0.3			
Fgf21	0.0	0.0	0.0	0.0	0.0	0.0	0.0	3.2	1.9	2.3	0.0	0.0	0.0	0.0	0.0	0.0	0.0	0.0	1.4	1.5	2.4	0.0	0.0	0.0			
Aldh18a1	0.1	-0.0	-0.2	-0.1	0.4	0.3	-0.4	3.5	3.2	3.5	0.6	1.0	0.6	0.2	0.0	-0.2	-0.0	-0.3	3.0	2.8	3.2	0.6	0.6	-0.1			
Aff5	0.2	-0.1	-0.1	-0.2	0.2	0.1	-0.2	4.2	3.9	4.2	0.3	0.6	0.4	-0.1	0.2	0.0	-0.0	0.0	3.9	3.9	4.1	0.4	0.5	1.8			
N4bp211	0.5	0.2	-0.1	-0.3	-0.2	-0.1	-0.3	3.0	2.3	2.2	-0.1	-0.2	-0.3	-0.2	-0.2	0.2	-0.3	-0.3	3.1	2.6	2.0	0.2	-0.3	2.3			
Pycr1	0.1	-0.0	-0.0	-0.0	0.0	0.4	-0.0	2.9	2.8	3.1	1.1	0.7	0.6	-0.0	-0.0	-0.0	-0.0	-0.0	2.5	2.6	2.3	0.6	0.6	0.2			
Pck2	0.1	-0.0	0.2	-0.3	0.3	-0.0	-0.0	1.7	1.5	2.0	-0.1	-0.3	-0.0	0.1	-0.2	0.6	0.3	0.2	1.8	1.9	1.3	0.2	-0.7	0.2			
Sesn2	-0.0	-0.1	-0.1	0.1	0.2	0.2	0.0	3.1	2.7	3.0	0.1	0.7	0.6	-0.0	-0.1	0.0	0.2	-0.0	3.0	2.4	2.4	0.5	0.4	0.4			
Nupr1	-0.1	0.3	-0.2	0.2	0.0	0.0	-0.1	2.8	2.2	2.5	0.1	-0.3	0.4	-0.2	-0.2	-0.0	0.0	-0.0	2.2	2.8	2.5	0.2	-0.4	-0.4			
Snd1	0.0	0.1	-0.1	-0.1	0.1	0.0	-0.1	3.1	2.4	2.6	0.7	0.7	0.5	-0.1	0.1	0.0	0.0	0.2	3.1	1.7	2.1	0.9	0.5	0.5			
Slc6a9	0.1	-0.1	-0.2	0.1	-0.1	0.2	-0.1	2.3	2.0	2.5	0.4	0.5	0.7	0.0	0.1	0.1	0.2	0.1	2.4	2.0	2.2	0.4	0.1	1.6			
Cyb5r1	0.0	0.1	-0.1	-0.1	0.1	0.1	0.0	3.0	2.4	2.9	1.5	0.6	0.8	-0.0	-0.1	0.1	0.1	-0.2	2.5	2.7	2.5	0.9	0.5	-0.4			
Tenm4	0.5	-0.2	-0.2	0.1	0.3	0.5	-0.2	3.7	1.8	3.2	-0.2	-0.2	-0.2	-0.2	-0.2	-0.2	-0.2	-0.2	2.1	2.5	2.5	-0.2	-0.2	-0.2			
Aldh11l2	-0.3	-0.2	-0.0	0.3	0.0	0.1	-0.2	2.9	2.8	3.2	-0.8	-0.5	-0.8	0.2	-0.0	-0.6	0.1	0.2	2.6	1.6	2.2	-1.0	-1.1	-1.5			
Slc3a2	0.0	0.1	-0.0	-0.0	0.1	0.1	-0.0	2.2	2.0	2.2	1.0	0.7	1.2	-0.1	-0.0	0.4	0.2	0.1	1.9	2.0	2.1	0.9	0.8	1.0			
Gadd45a	0.3	0.1	-0.0	-0.2	0.0	0.3	-0.2	1.2	1.6	1.4	-1.5	-1.1	-1.9	-0.1	-0.3	0.2	-0.4	-0.6	1.4	2.0	1.7	-0.9	-1.2	-0.1			
Gars	-0.0	0.1	0.0	0.0	0.1	-0.1	0.2	2.3	2.0	2.5	0.9	0.8	0.8	-0.1	-0.1	-0.0	-0.1	-0.1	2.1	2.3	2.2	1.1	0.6	0.7			
Soat2	0.0	0.0	0.0	0.0	0.0	0.0	0.0	2.8	1.6	2.1	0.0	0.0	0.0	0.0	0.0	0.0	0.0	0.0	1.7	1.6	1.7	0.0	0.0	0.0			
Slc7a11	0.0	0.0	0.0	0.0	0.0	0.0	0.0	1.2	0.9	1.4	0.0	0.0	0.0	0.0	0.0	0.0	0.0	0.0	1.1	1.2	0.7	0.0	0.0	0.0			
Arhgef2	-0.4	-0.2	-0.0	0.1	0.1	0.1	-0.3	2.0	1.7	2.1	0.1	0.2	0.1	0.3	0.1	0.4	0.3	0.4	1.6	1.6	1.5	0.1	0.5	-0.3			
Brca2	-0.1	-0.1	-0.1	-0.0	-0.1	0.3	-0.1	3.8	3.0	2.9	-0.1	-0.1	-0.1	-0.1	0.3	-0.1	-0.1	-0.1	3.2	3.2	2.7	-0.1	-0.1	-0.1			
Phf10	0.2	0.2	-0.0	-0.3	0.0	0.0	0.6	1.5	1.4	1.6	0.5	0.0	0.2	-0.0	-0.1	-0.1	-0.1	-0.1	1.4	2.6	1.6	0.3	0.2	0.1			
Ddit3	-0.1	-0.3	0.2	0.2	-0.2	-0.2	-0.0	2.2	2.1	2.4	0.5	0.5	0.1	-0.2	0.1	-0.3	-0.0	0.1	1.8	2.2	1.5	0.0	0.3	0.5			
Slc7a3	0.0	0.0	0.0	0.0	0.0	0.0	0.0	2.3	1.5	2.1	0.0	0.0	0.0	0.0	0.0	0.0	0.0	0.0	1.5	1.0	1.5	0.0	0.0	0.0			
Ostn	0.0	0.0	0.0	0.0	0.0	0.0	0.0	3.2	1.6	2.7	0.0	0.0	0.0	0.0	0.0	0.0	0.0	0.0	1.0	1.4	1.1	0.0	0.0	0.0			
Extl1	-0.2	-0.2	-0.1	-0.1	0.1	-0.1	-0.2	2.1	1.7	2.0	-1.2	-0.6	-1.2	0.1	0.4	-0.1	0.2	-0.2	1.6	1.3	1.8	-0.9	-0.4	-1.0			
Cebpg	0.3	0.0	-0.1	-0.2	0.2	-0.2	0.1	1.9	1.9	2.1	0.3	0.0	-0.2	0.2	-0.2	0.5	0.2	0.4	1.6	2.0	1.8	0.1	0.1	0.6			
Xpot	-0.1	-0.1	0.0	0.0	-0.2	0.1	-0.0	2.2	1.8	2.2	0.6	0.4	0.4	-0.0	-0.1	-0.3	-0.0	-0.1	1.7	1.7	1.6	0.5	0.6	0.2			
Yars	0.0	-0.1	-0.0	-0.0	-0.0	0.1	-0.0	2.0	1.9	2.1	0.7	0.6	0.5	0.0	0.1	0.1	0.1	-0.1	1.7	1.9	1.8	0.8	0.7	0.7			
Adm2	0.0	0.0	0.0	0.0	0.0	0.0	0.0	2.5	1.8	2.3	0.0	0.0	0.0	0.0	0.0	0.0	0.0	0.0	1.5	1.7	1.7	0.0	0.0	0.0			
Ghitm	0.0	0.0	-0.0	0.0	-0.1	0.0	0.1	1.4	1.2	1.4	-0.0	-0.2	-0.1	0.0	-0.0	-0.0	-0.0	-0.0	1.3	1.3	1.1	-0.2	-0.1	-0.2			
Stbd1	0.1	0.3	-0.4	0.2	0.0	0.1	-0.7	2.2	1.5	2.4	0.2	-0.6	-0.0	0.1	-0.3	-0.1	0.3	0.5	1.2	1.8	1.4	-0.1	-0.9	-0.6			
Stard5	0.4	-0.2	-0.2	0.1	-0.2	-0.3	0.0	1.8	1.6	2.2	-0.4	-0.2	0.4	-0.0	-0.2	-0.2	-0.1	0.1	1.4	0.9	1.1	-0.2	-0.5	-0.5			
Slc25a39	0.2	-0.0	-0.2	-0.1	0.1	0.1	-0.0	1.7	1.2	1.4	0.2	0.2	0.3	0.1	0.0	-0.0	-0.1	-0.0	1.3	1.2	1.1	0.2	0.2	-0.2			

This also allowed us to use a two-way ANCOVA to correct for sex effects on expression of some of the DELE1 dependent genes.

2) Does nutritional support influence metabolomics reported here?

This is an excellent question. We did find that nutritional support modestly extended the lifespan of *Chchd10* G58R; *Dele1* KO mice in data that we included as a panel in a Supplemental Figure in the initial submission (now Fig. S3B in the revised manuscript). Unfortunately, no tissue could be collected in this survival study. While we would like to do a further exploration in metabolomics in future work, we feel that an additional metabolomics study, which would take several months and require many additional mice, is beyond of the scope of the present work.

3) Not all figure legends describe statistical tests used. And did the data follow a parametric or non-parametric distribution?

This is an important point. We have made this clearer in the figure legends and methods throughout, also, in response to Reviewer 3.

In the transcriptomics, proteomics, and metabolomics datasets, the data were generally normally distributed and parametric testing was most appropriate. To be consistent we have used parametric tests for all the data in these datasets.

For quantification of data from immunoblots or images either parametric or non-parametric testing was used depending on the distribution of data in the datasets. In these cases, the statistical testing was performed in GraphPad Prism 10. The specific tests are now indicated in the Figure Legends.

In conducting this review and additionally gathering and reviewing the Source Data to meet the Journal's requirements we noted errors in four panels that have been corrected in the Revised Figures, none of which changes the interpretation of the data (numbering of revised Figures is used):

- (1) Fig. 1F. The original figure panel showing survival analysis from C10 G58R; *Dele1* KO mice mistakenly contained the combined data from two experiments involving the mice. This has been corrected. The data in Fig. 1F and S3B are now non-overlapping as had been originally intended.
- (2) Fig. 2I. Placing the graph in the Illustrator file from Prism lead to loss of some data points. This has been fixed. This does not change the underlying statistical analysis.
- (3) Fig. 2J. Review of the statistical analysis performed in this experiment showed that inadvertently different statistical tests had been used for some of the graphs. These have all been reanalyzed using the same statistical test. This does not change the underlying interpretation of this graph, namely, that *Dele1* KO does not affect CI or CIV activities in C10 G58R or *Tfam* mKO mice.
- (4) Fig. 6C. The y-axis of the middle graph was incorrect. It has been corrected. The right graph had an extra datapoint for one of the conditions. This has been corrected. These do not change the statistical analysis or the interpretation.

Referee #2:

In this paper Lin et al. perform an in vivo characterization of DELE1 knock out model in the context of mitochondrial stress. OMA1-DELE1 signaling cascade elicits a protective mitochondrial integrated stress response (mt-ISR) by activating a specific gene signature. The impact of DELE1 mtISR has been characterized in different in vivo models of mitochondrial myopathies. In the last part of the manuscript, the authors highlight the metabolic impact of DELE1 mediated mt-ISR on tissue metabolism at the level of heart and skeletal muscle.

Although the findings are very interesting, the paper has an overall confused structure that should be amended. Also, some minor experimental points shall be addressed.

The DELE1 total knockout (KO) model does not exhibit any gross phenotypic alterations. However, under stress conditions, such as cold exposure, the authors demonstrate a blockage of the mitochondrial

integrated stress response (mt-ISR) activation in brown adipose tissue (BAT) (refer to Fig. 1H, I, and J).

In Fig. 1E, a transmission electron microscopy (TEM) image of cardiac mitochondria is presented. While no differences in the area or aspect ratios are reported (see Fig. 1F), the image shows enlarged mitochondria and the presence of small, electron-transparent mitochondria in DELE1 KO hearts. **The authors are requested to comment on these observations.**

In retrospect, we realize our representative image choice and labelling made it difficult to compare organelles between the samples. The round electrolucent structures are not mitochondria but lipid droplets. They can be identified at higher magnification by the absence of a phospholipid bilayer (the bilayer has a characteristic “railroad track” appearance, while the phospholipid monolayer surrounding the lipid droplet does not). These were present in both samples but happened to be captured in the initial representative image from the DELE1 KO hearts only. We have exchanged the WT image with one that captures lipid droplets and have labelled these in both to avoid confusion. The new panel is reproduced here:

E Heart Mitochondria

If the only stress tested in this mouse model was cold exposure in BAT, **it is misleading to present TEM data on mitochondrial status in the heart while placing BAT analysis in Supplementary Fig. 1.**

We welcome the Reviewer’s suggestion to improve the flow and avoid confusion. What was Fig. 1 has now been spilt up. A description of the unstressed Dele1 KO model has moved to Fig. S1 and data from BAT has been deemphasized and moved to Fig. 6A-B and S9. These changes also allow us to accommodate new data requested by Reviewer 3 to expand our analysis of skeletal muscle.

The data on the status of Dele1 KO mitochondria in the heart now appears in Fig. S1 and is immediately followed by discussion of the phenotypes resulting from crossing the four myopathy/cardiomyopathy models with Dele1 KO mice.

We hope that this restructuring improves the flow and avoids confusion, which was also a concern of Reviewer 3.

To establish robust evidence of whether mitochondria undergo structural alterations under stress conditions in the DELE1 KO model, **BAT mitochondria should be reported and measured. Specifically, do they change under room temperature (RT) or cold exposure conditions?**

We have now attempted TEM of samples from this experiment. The quality of the data, however, is limited as the available samples were PFA fixed and had been stored at 4°C for more than a year. We now include representative TEM images from this material in Fig. S9B. These demonstrate similar morphological changes in WT and DELE1 KO following cold stress, including shrinking lipid droplets, as had also been observed in the H&E. It is reproduced here:

We have not attempted quantification of mitochondrial morphology or ultrastructure from these samples given their poor quality. As we did not make a claim about structural changes in the BAT and this experiment has been deemphasized, we believe repeating the experiment to obtain better quality samples specifically for structural analysis by EM is beyond the scope of the present study. We have therefore limited our claim regarding BAT to demonstrating that DELE1 mediates the ISR induced by cold stress.

In Fig. 2, the authors explore the effect of DELE1 deletion in the context of concomitant mitochondrial-related myopathy or cardiomyopathy. DELE1 KO mice were crossed with models of downregulation of

CHCHD2/10 and TFAM, as well as up-regulated dominant negative mutants of CHCHD10 protein. The worsening in lifespan and mouse activity indicates that blocking mt-ISR in vivo has detrimental effects in the presence of compromised mitochondrial fitness.

The biochemical analysis in Fig. 2M is related to cardiac tissue. **The cardiac mitochondrial phenotype (Fig. 1E) should be relocated in this context, addressing the concerns mentioned earlier.**

We thank the reviewer for this suggestion to improve the flow and avoid confusion. We have moved these data to what is now Fig. S1. As noted above, we have removed discussion of BAT from this section, which we believe this Reviewer and Reviewer 3 recognized as most impeding the flow and causing confusion.

The narrative now starts with a general description of the Dele1 KO. It then compares WT and Dele1 KO mice in the absence of stress, including preservation of normal mitochondrial ultrastructure in the heart. This is immediately followed by discussing the cross of Dele1 KO to four myopathy/cardiomyopathy models.

This progression avoids the interruption of the BAT experiment and keeps the focus of the first five figures on the heart.

To dissect the role of OMA1 and DELE1 in mt-ISR response activation (Fig. 3), the authors bred the two mouse strains with the C10G58R mutant. OMA1 gene deletion has an additive effect on the lifespan of mice compared to DELE1 KO alone in the context of unhealthy mitochondria. **Crossing with OPA1 Δ s1/ Δ s1 mice excludes the concomitant effect of the other main substrate of OMA1 protease. This experiment does not directly address whether blocking OPA1 cleavage is responsible for the additional effect due to OMA1 deletion. How is the mt-ISR in these mice?**

As expected, the mt-ISR is activated in hearts of C10 G58R; OPA1 Δ s1/ Δ s1 mice. This is evidenced by MTHFD2 upregulation by C10 G58R in the presence or absence of the OPA1 Δ s1 allele. We had previously left these data out of the immunoblot of what is now Fig. 2B. We now include them to make this point clear. The new panel from the revised figure is reproduced here:

*OMA1 cleavage generated OPA1 isoforms

The analysis of the expression of MTHFD2, mirroring that performed in the OMA1 knockout mice, is a pivotal experiment. **The data presented in Supplementary Fig. 2, Panel I, is crucial for substantiating the direct influence of DELE1 in activating this response. MTHFD2 activation is observed in C10 G58R/OMA1KO mice but not in C10 G58T/DELE1 KO mice. However, only one sample of C10 G58R/DELE1KO mice has been presented. To enhance the robustness of the analysis, additional samples with this genotype should be included, and improvements to the OMA1 western blot are warranted.** The experiment supports the transcriptomic profile in Fig. 3E, where only DELE1 downregulation is involved in the inactivation of mt-ISR gene profile, without affecting the level of oxidative phosphorylation (OXPHOS) content or oxygen consumption rate.

We agree this is important to demonstrate clearly. We now include analysis of MTHFD2 and OMA1 for four CHCHD10 G58R; Dele1 KO mice and other relevant genotypes in what is now Fig. S3J. The OMA1 blot from these samples show that OMA1 is decreased by C10 G58R stress in the presence or absence of DELE1, likely due to its cleavage on activation. These data are also consistent with the mitochondrial proteomics (which quantified MTHFD2 in addition to other DELE1-dependent enzymes) that appear in what is now Fig. 4E. The new data suggested by the Reviewer is reproduced here:

In Fig. 4, the authors examine mitochondrial ultrastructure in different mutant/DELE1KO mice, encountering various ultrastructure abnormalities that were only partially and differently exacerbated by the absence of DELE1. There are no common features underlying a general effect of DELE1 downregulation. **However, it would be interesting to check if in OMA1KO mice crossed with OPA1 Δ s1/ Δ s1 mice there is no alteration of cristae ultrastructure, providing clarity that DELE1 downregulation alone does not impact the cristae alteration in OMA1KO mice.**

This is an interesting suggestion. The generation of C10 G58R; OMA1 KO; OPA1 Δ s1/ Δ s1 triple mutants would require considerable time (~1 year) and resources. Additionally, OMA1 KO; OPA1 Δ s1/ Δ s1 would be equivalent to OMA1 KO with respect to proportion of OPA1 isoforms, as OMA1 is the sole protease cleaving at the Δ s1 site. In both models, there would be increased *a* and *b* combined with absent *c* and

e, due to lack of OMA1 cleavage at $\Delta s1$. They would also be equivalent in terms of their effect on the mt-ISR as both would lack OMA1.

Therefore, to address the question of what C10 G58R phenotypes are dependent on OMA1-DELE1 signaling vs. OMA1 independent of DELE1 in the time allotted for a revision for the EMBO Journal, we have elected to compare the C10 G58R; *Dele1* KO with previous samples that we generated from C10 G58R; *Oma1* KO mice that were originally examined in our previous paper (Shammas et al. 2022).

To do so, we have cut sections from the prior epon-embedded blocks from these animals and analyzed them using the same pipeline that was used for C10 G58R; *Dele1* KO. This comparison makes clear that the formation of cristae inclusions is independent of both *Dele1* KO and *OMA1* KO as they appear in both. It additionally makes clear that the increased formation of electrolucent mitochondria is downstream of DELE1, as it is observed in both C10 G58R; *Dele1* KO and C10 G58R; *Oma1* KO hearts. Finally, it demonstrates that some morphological changes (specifically the formation of long thin mitochondria) are mediated by OMA1 independently of DELE1. We speculate that this is likely due to L-OPA1 cleavage by OMA1 but cannot rule out that this may be due to cleavage of other substrates by OMA1.

These data appear as a new supplemental Figure (Fig. S5) and are reproduced here:

The transcriptome signature performed in mitochondria from compromised hearts highlights that in the absence of the DELE1 gene, the mt-ISR response is silenced (Fig. 5).

We are appreciative the Reviewer's interpretation matches our own.

Figure 2S panel I: the quality of the OMA1 western blot should be improved.

We attempted this blot again with the same lysates but it was not improved. This is probably due to the age of the lysates. Additional frozen material was not available from all samples to generate new lysates. We believe that, although not ideal, the pattern of OMA1 results demonstrates the correct genotype of Oma1 KO mouse samples in this Supplemental Figure panel. Additionally, the lack of OMA1 can be appreciated in the pattern of OPA1 cleavage (specifically the absence of the c and e bands – clearest in the case of c), which we believe is consistent with the OMA1 levels. We have added additional labels to increase the interpretability of these data for the reader. This revised panel now appears in Fig. S3I and is shown here:

Referee #3:

general summary and opinion about the principle significance of the study, its questions and findings

Lin and colleagues report striking phenotypes that are interesting in the context of mt-ISR regulation. However, the sheer density of data and multiple narratives obstruct its clarity, making the manuscript difficult to read at times. The overall flow is disjointed, and the manuscript requires greater streamlining to be more accessible. The authors show that genetic ablation of DELE1 does not affect normal

physiology, but does have physiological relevance during specific contexts of extreme mitochondrial stress. However, the lack of conceptual clarity makes it challenging to distinguish the novelty of this manuscript from their 2022 paper.

specific major concerns essential to be addressed to support the conclusions

The manuscript presents numerous paradigms, but they are only superficially detailed. The transitions between models and approaches are not clearly articulated. For instance, the authors start with BAT, but the manuscript is clearly muscle-focused, and the switch to muscle is very abrupt. **The authors need to better craft this switch and thread the most important results together with solid rationales for the experimental paradigms chosen.**

We have tried to improve the flow and clarity also in response to Reviewer 2. The largest change is to deemphasize the BAT experiment and to move the description of this experiment to the beginning of Fig. 6, in which we compare the DELE1 mt-ISR across mitochondria-rich tissues. The description of the DELE1 KO mouse in the absence of stress in has now become Fig. S1 (previously the top of Fig. 1).

These changes allow us to start in the main figures with the cross between DELE1 KO and four models of myopathy/cardiomyopathy and allows for better flow from Fig. 1 through Fig. 5. These all focus on survival and analysis of the mt-ISR in the heart.

Starting at the top of Fig. 6, we then compare the heart mt-ISR to other tissues and find the heart and skeletal muscle are most similar.

This transitions into our focus on skeletal muscle at the bottom of Fig. 6 and in Fig. 7. This has also given us space in the main figures for new data generated in response to other suggestions by this Reviewer.

We have also cut some of the results descriptions in the interest of streamlining the narrative and better emphasizing the most important results, as suggested by the Reviewer.

Furthermore, the statistical descriptions lack sufficient detail. **To ensure a clear interpretation, it is necessary to include effect sizes and appropriate tests.**

We have clarified the statistical descriptions in the methods and in the figure legends also in response to Reviewer 1. These are also detailed in our response to Reviewer 1 above.

In the original manuscript, effect sizes were generally represented in the hue of the heat map as log₂ fold changes. But we recognize that it may be difficult to appreciate the magnitude of the effects from the hue alone. In the revised main Figures, we have added subplots indicating the fold change numerically for key comparisons as well as the results of statistical testing for the same comparisons. An example representative of the changes made throughout is reproduced here:

Heart: DELE1-Dep DEGs in 3 of 3

Figure 4

In the new RNAseq data that appears in Fig. 6 in response to another suggestion from this Reviewer, we have emphasized showing individual values of key genes in the main figures, as discussed further below. We hope that this combination of data representation increases the accessibility of our key findings regarding the DELE1 dependence of the transcriptional mitochondrial stress responses.

A major concern is the repeated primary assertion that 'the mt-ISR has not been examined in skeletal muscle'. However, this statement is incorrect as the group of Anu Suomalainen originally profiled stage-wise mt-ISR regulation in disease models for over a decade. In a recent 2019 paper published in Cell

Metabolism, Forsström et al examined the mt-ISR in mitochondrial myopathy. They reported that FGF21 is a local and systemic signal of mitochondrial stress in both preclinical mouse and human models. The authors previously cited this work in their 2022 paper, Shamma et al, but did not do so in this present submission. It is contentious to claim such novelty in light of the pre-existing body of literature. This omission is a very unfortunate oversight. **If this manuscript is revised, a significant rewrite with greater nuance, precision, and appropriate attribution is needed.** The idea of distinct mitochondrial defects triggering the mt-ISR has been around for some time now. See work by the Mootha group (Mick et al. 2020). Taken together, although a lot of work is collated in this submission, it is difficult to see the conceptual novelty of this paper as a whole.

We apologize for the lack of precision regarding our claims of novelty. Our attempt to be concise in the introduction may have left the impression that we were making more general claims of primacy than we intended.

We had not intended to claim that we are the first to examine transcriptional mitochondrial stress responses in skeletal muscle, which we understand was first described by the Suomalainen lab in Tynismaa et al. (2010) [PMID: 20656789], work which we cited in the initial submission. We understand that similar responses were subsequently described by other labs in heart muscle, including those of Aleksandra Trifunovic and Nils-Göran Larsson. It was our intention to make this clear in our initial submission.

As evidence of this intention, in the Introduction of the original submission, we included the sentences:

“The first mitonuclear signal in mammalian cells was identified in the setting of mitochondrial protein unfolding stress and was named the mitochondrial unfolded protein response (mt-UPR) (Zhao et al, 2002). Subsequently, an amino acid starvation-like response was identified in mouse models with defects in mtDNA maintenance (e.g., mutations in Twinkle or Tfam) or mtDNA expression (e.g., mutations in Dars2) causing severe OXPHOS deficiency (Tynismaa et al, 2010; Dogan et al, 2014; Kühl et al, 2017). Responses like these were initially attributed to energy and amino acid sensing pathways outside of the mitochondria, with upstream signaling from AKT, mTOR, and/or GCN2, converging on transcription factors such as ATF4, ATF5, and CHOP (Tynismaa et al, 2010; Khan et al, 2017; Mick et al, 2020).”

In the Results of the initial submission, we included the sentence:

“These genes also overlapped extensively with those previously described as part of mitochondrial stress responses in mitochondrial myopathies, including Atf4, Atf5, Mthfd2, and Lonp1, all of which had > 81% DELE1-dependency in all three models (Fig. 5D) (Anderson et al, 2019; Tynismaa et al, 2010; Dogan et al, 2014; Kühl et al, 2017; Nikkanen et al, 2016).”

And in the Discussion of the initial submission, we included the sentences:

“Notably, the DELE1 mt-ISR transcriptional signature identified here overlapped extensively with that in other models of disrupted mtDNA maintenance or expression. Indeed, a similar response, identified first in Twinkle mutant mice (Tynismaa et al, 2010), has been documented with disruptions at each step of mtDNA maintenance and expression, including mtDNA replication (Twinkle KO), mtDNA maintenance (Tfam KO), mtDNA transcription (Polrmt KO), mtRNA stability and processing (Lrpprc KO), and mtRNA translation (Mterf4 KO and Dar2 KO) (Dogan et al, 2014; Kühl et al, 2017).”

However, we recognize that in citing this work in the initial submission we did not specifically mention skeletal or heart muscle. We are in complete agreement with the Reviewer that these transcriptional stress responses in skeletal muscle were identified over a decade ago and are not first described in our manuscript. We have rewritten the introduction to make this explicit and to be more precise and specific regarding attribution of prior findings.

Our intention was to suggest that the DELE1-dependence of these transcriptional responses has not yet been assessed in skeletal muscle, with the assumption that DELE1 operates primarily through a OMA1-DELE1-HRI-eIF2A-ATF4 cascade. The sentence we believe the Reviewer may be referencing most directly from the initial submission is: “Additionally, the mt-ISR has not been examined in skeletal muscle, the tissue that is primarily affected in mitochondrial myopathies.” This sentence came at the end of a paragraph in the introduction that describes three papers assessing the OMA1-DELE1 dependence of mitochondrial stress responses in the heart. It was meant to contrast what has been recently described as DELE1-dependent in the heart with what is known about DELE1-dependence in skeletal muscle. However, we see how this could be misleading. We have now extensively rewritten this section of the Introduction to provide greater detail on prior discoveries and to be more precise on our claims of novelty. We have also tried to state more clearly our claims of novelty in the final paragraph of the Introduction.

We additionally now cite Forsström et al., which was referenced by the Reviewer. Its omission was not an attempt to obscure this work, as I hope is clear from the fact that we cited three other papers from the same lab concerning stress induced transcriptional responses in muscle. In our JCI (2022) paper, Forsström et al. was cited specifically in reference to FGF21, which we had emphasized in Shamma et al. (2022) and not in the initial draft of our present manuscript. We also now include a discussion of our results from skeletal muscle relative to Forsström et al. in the Discussion:

“Using a model of early-onset MM, we found the DELE1-mediates transcriptional mitochondrial stress responses in skeletal muscle in addition to heart muscle. The transcriptional mitochondrial stress response in skeletal muscle has been best described in the “deletor” mouse model, where it was recently shown to occur in stages, with stage 1 defined by marker genes *Fgf21* and *Mthfd2* (observed starting around 9 months) and stage 2 defined by marker genes *Psat1* and *Pghdh* (starting around 16 months) (Forsström et al, 2019). Expression of stage 2 genes in the “deletor” mice was found to depend on *Fgf21*, suggesting they are activated via an autocrine or paracrine effect of the mitokine FGF21. Notably, we found that in our IMMD model both stage 1 and stage 2 marker genes were elevated in skeletal muscle in the first month of life and both depended on *Dele1* for activation. There are notable differences between the IMMD and “deletor” MM models that may account for these differences in the timing of transcriptional stress responses. The “deletor” mice develop myopathy as adults and mitochondrial dysfunction is highly variable between skeletal muscle fibers, because of the stochastic mtDNA deletion process occurring in each muscle fiber (Tynismaa et al, 2005). By contrast, the IMMD model has early onset of MM and mitochondrial dysfunction likely occurs more uniformly, as evidenced by globally reduced COX staining (but few truly COX negative fibers) in striated muscle (Shamma et al, 2022). The paracrine effects FGF21 may be more critical in the setting of mosaic OXPHOS dysfunction across fibers. Alternatively, local FGF21 levels may reach a critical level for stage 2 gene activation more quickly in the setting of uniform mitochondrial dysfunction. It is not yet clear if there are stages in the

IMMD model that are compressed in time or if this model lacks the staged response observed in the “deletor” mice. Examination of whether *Fgf21* KO blocks activation of stage 2 genes in the IMMD model might help address this question.”

The Reviewer also references Mick et al. (2020) from the Mootha lab. We had previously cited this paper, but we have now expanded the description of this and other prior work examining mitochondrial stress responses in cell lines in the introduction:

“Responses upregulating these AARE genes have also been observed in cultured cells challenged with diverse inhibitors of OXPHOS, disruption of mitochondrial translation, or inhibition of CHCHD4-dependent mitochondrial import (Mick et al, 2020; Forsström et al, 2019; Quirós et al, 2017), and were shown to depend on ATF4 in many of these cell models (Mick et al, 2020; Forsström et al, 2019; Quirós et al, 2017).”

And later in the introduction:

“Similarly, a study of cultured C2C12 myoblasts and myotubes, a common cell culture model of striated muscle, concluded that there may be multiple upstream stress signaling pathways converging on ATF4-targeted genes (Mick et al, 2020).”

We believe our findings are distinct from those in Mick et al. in several respects. First, Mick et al. did not identify a mechanism upstream of ATF4 for mitochondrial stress responses in C2C12 myotubes and concluded (as is evident from the abstract of their paper) that there is not a single mechanism governing mitochondrial stress responses. This was based on their observation that AARE-responsive genes are upregulated via a different mechanism in C2C12 cells in the myoblast state (which was found to be GCN2 dependent) and C2C12 cells in the myotube state (which was found to be GCN2 independent). Second, although Mick et al. examines different OXPHOS inhibitors and found a similar transcriptional response, they did not compare protein misfolding stress and stress from loss of mtDNA maintenance. Third, their results are from cell models growth in cell culture conditions. In contrast, our *in vivo* results demonstrate that in heart muscle DELE1 mediates a stereotyped transcriptional stress response to diverse mitochondrial stressors, including protein misfolding and loss of mtDNA maintenance. We also demonstrate the DELE1-dependence of these transcriptional responses in skeletal muscle from one of these models. We believe these key differences highlight the novelty of our findings relative to the those of Mick et al. (2020).

The survival phenotypes are compelling. To what extent are these a function of developmental maturation? **The authors should discuss and speculate whether conditional Dele1 inactivation in later life could induce pathology.**

This is an excellent question. Based on the difference we observe between the models that have earlier vs. later onset of the DELE1 mt-ISR activation, we speculate that the DELE1 mt-ISR is most important for early growth in the CHCHD10 G58R and CHCHD2/10 DKO models. We have added the following to the Discussion:

“We thus hypothesize that the importance of the DELE1 mt-ISR depends on the developmental stage of the animal, with the DELE1 mt-ISR being most important during periods of rapid hypertrophic growth of

post-mitotic cells. This model would predict that that inhibiting the DELE1 mt-ISR in adult mice under mitochondrial stress would be better tolerated than inhibiting the DELE1 mt-ISR in neonatal mice.”

Much more detail is also required on the genetic background of the mice. **How has the breeding been performed? This needs to be explicitly described in the methods. How does the genetic background impact the survival phenotypes described?**

We now include greater detail on the specific crosses and the genetic background of the animals described. All crosses used now appear in Fig. S17. We have been careful to compare individuals from the same crosses in each experiment to avoid confounding effects from background. This means that in most experiments the control animals were from the same crosses. The only exceptions to this rule were for comparisons involving the CHCHD2/10; Dele1 DKO mice, where appropriate controls could not be derived from the cross (as they would be KO for C2 or C10).

In all cases, the mice were generated and maintained on a C57BL/6J, except for Oma1 KO, which was originally on a mixed background including 129S6/SvEvTac, C57BL/6, and C57BL/6NCr. As noted above, however, all comparisons (except for CHCHD2/10 DKO) were made among mice from the same set of crosses, to control for background effects.

It is possible that the overall survival phenotypes could be quantitatively different on different backgrounds due to genetic modifiers, but we speculate that we would observe the same general relationships between models and among genotypes on other backgrounds. We have added the following to the Discussion to make this caveat explicit:

“Another potential limitation is that the experiments were performed on the C57BL/6J background, and it is recognized that genetic background can modify mitochondrial phenotypes. Nonetheless, we speculate that we would observe the same general relationships between models and among genotypes on other backgrounds.”

The breeding schemes in Fig. S17 is reproduced here:

Supplemental Figure 17

Breeding scheme

The overall phenotyping focuses on the survival effects, which the reviewer acknowledges are very nice - but without drilling down into the in vivo pathomechanisms at play. Striated muscle is not homogeneous, and details on the muscle fiber types affected are lacking from the manuscript. **Does Dele1 protect integrity of specific muscle fiber types, or all?** This question is important and essential to understand the protective mechanistic functions of DELE1 in vivo, given the fiber-specific mitochondrial

features that mice and humans exhibit. **Put simply, the authors should analyse their muscle samples to assess if ablation of *Dele1* accelerates the demise of specific fiber types.**

This is an excellent question. To address it, we have now assessed for the presence of ubiquitinated aggregates and measured the fiber cross-sectional area (CSA) in type 1, 2a, and 2b fibers, separately, using adjacent gastrocnemius sections from three C10 G58R; *Dele1* KO mice. Ubiquitinated aggregates were observed in each fiber type. Qualitatively this included type 2x fibers, although our staining protocol did not allow for efficient quantification of this fiber type (as it is defined by the absence of the other three fiber type markers, using well-validated antibodies that we have adopted). There was a trend toward increased presence of ubiquitinated aggregates in the more oxidative fibers, but this reached significance only for the comparison between type 1 and type 2b fibers. In all three fiber types quantified, those with ubiquitinated aggregates had a smaller CSA than those without. We also stitched 20X images to provide an overall view of the whole gastrocnemius section at sufficient resolution to see fibers with ubiquitinated protein aggregates. Fibers with aggregates tended to be found in near the anterior edge of the gastrocnemius, which had the highest concentration of mixed fiber types.

These data now appear in Fig. S16 and Fig. 7I - K. They are reproduced here:

Fig. S16

C10 G58R; *Dele1* KO gastrocnemius muscle

Fig 7.

The RNA data is derived from microarray analysis, which is still a valid method for some clinical applications and has greater sensitivity than RT-qPCR. However, its dynamic range is inferior to that of whole genome transcriptomics. This method is less sensitive than more widely used contemporary approaches such as RNA-seq/whole genome transcriptomics, and is not aligned with the current state-of-the-art in the field. The authors' rationale for selecting this method is unclear. **Ideally, these findings should be authenticated using RNA-seq/whole genome transcriptomics. To gain more insight and profile the RNA-seq data, Ingenuity, gProfiler or similar pipelines could be used. Additionally, it is important to identify the promoter motifs common to differentially modified transcripts overlapping with the three mouse models. This can be easily assessed by MEME-ChIP analysis of RNA-seq data.**

We agree with the Reviewer that RNAseq has advantages particularly when it comes to detection of splice variants, novel transcripts, and base modifications, and in addition has increased dynamic range. Given we were most interested in evaluating the regulation of well-established transcripts, we elected to use microarray for most experiments, as it was more cost-effective, while still providing similar data quality as RNAseq for those transcripts that are measured. Indeed, this was the recommendation of the experts in our gene expression core, which uses both techniques.

To address the Reviewer's concern, we have now performed additional transcriptomics from the gastrocnemius and heart tissue using RNAseq from the C10 G58R; Dele1 KO cohorts of mice. We additionally performed transcriptomics on the tibialis anterior muscle, which allowed us to compare the overlap of DELE1-dependent DEGs in two skeletal muscles vs. skeletal muscle and heart muscle. We also included additional samples of both sexes for heart and gastrocnemius which allowed us to address the concern of Reviewer 1, regarding whether the DELE1-dependent gene responses are observed in both biological sexes.

As expected, there was good agreement between the results from RNAseq and microarray. We believe these validate the microarray data, as requested by the Reviewer. The correlation is shown for the heart in Fig. S12A and is reproduced here:

Analyzing the new RNAseq data, we confirmed that a core mitochondrial transcriptional response in both heart and skeletal muscle is DELE1-dependent. Interestingly, CHCHD10 was among the DELE1-dependent genes observed in skeletal muscle but not the heart. We also confirmed that a pro-catabolic transcriptional response is observed in the skeletal muscle of C10 G58R; Dele1 KO animals and not other genotypes. This response resembles those that have been reported following starvation and hypoxia challenges (PMID: 23624791). These new data appear in Figures S12 – 15 and Fig. 6 D- J, the latter of which is reproduced here:

Finally, we analyzed the DELE1 dependent DEGS using gProfiler as suggested by the Reviewer. As expected, DELE1-dependent DEGs were most enriched for ATF4 and its associated AARE/CARE-like transcription motif (RNMTGATGCAAY) in each tissue. We additionally compared our data to previous datasets of ATF4 ChIP-Seq data from mouse embryonic fibroblasts, and a recent study that assessed gene expression in the tibialis anterior of conditional ATF4 KO mice (PMID: 37014538). These comparisons confirmed the substantial overlap between the DELE1-dependent DEGs and genes targeted by ATF4 (E in the figure reproduced above) and between DELE1-dependent DEGs and genes dependent on ATF4 for expression in non-stressed conditions (F in the figure reproduced above).

Additionally, providing greater detail on metabolomics methodology, including metabolite extraction and quantification standards, would enhance the reproducibility and interpretation of the data.

We now include additional data on metabolomics methodology in the methods and in what are now Tables S4 and S5. All retention times for features and quantification standards are listed in the additional tabs that now appear in these tables.

minor concerns that should be addressed

Including a detailed limitations section would be prudent, outlining inherent caveats in the findings and delineating avenues for further exploration.

We thank the Reviewer for this suggestion. We understand that a limitations section is not typical for this journal, but we have introduced caveats where appropriate as prompted by the Reviewers concerns and suggestions, as detailed above.

Dear Derek,

Thank you submitting a revised version of your manuscript. It was sent to two of the three reviewers that originally appraised your work; their comments are attached to the bottom of this email. As you will see, Referee #2 is satisfied with the changes you made. Referee #3, lists some recommendations for improving the clarity of your work for a general audience which I urge you to consider. Before we can move forwards towards publication of your manuscript, there are some remaining editorial points which need to be addressed. In this regard, would you please:

- acknowledge funding from grant number 1ZIAN003169 in the manuscript text,
- include up to five keywords,
- rename the Conflict of interests statement the 'Disclosure and competing interests statement',
- include callouts in the manuscript text for Fig. 2H-I, 2K, 6B, Table S9 (which should be renamed Dataset EV9 [see next point]),
- upload Appendix Supplemental Tables as individual dataset files, and rename as Dataset EV1-EV9 with the appropriate callouts; legends should be removed from the README file, and included in the corresponding Excel file as a separate tab,
- use the nomenclature Appendix Figure S1-S17 throughout the Appendix PDF and callouts in the manuscript file,
- EMBO Journals now include a reagent table to help with reproducibility of your work, please provide this,
- provide source data files for micrographs in Fig 3A-F and fill in the SD checklist,
- if images in Figure 3F and Figure EV4E (also in EV4 F and G) are taken from the same micrograph, please state this in the figure legend of Figure EV4E,
- some additional duplications have been found among the micrograph images; for example, there is a partial overlap within Figure EV1 H & J. Please provide source data for all micrograph images,
- define the annotated p values ****/****/* as well as provide the exact p-values for the same in the legend of figure 1b, e; 2a, c, i-k; 5b-g; 6g-i; 7c, e, g, j-k; extended data figure(s) 1b; 5b, e,
- indicate the statistical test used for data analysis in the legends of figures 5b-g; 6b, g-j,
- define 'n' in the legends of figures 1b, e, h, k, n; 5b-g; 6g-i; 7c, e, k, extended data figure(s) 1b,
- define error bars in the legends of figures 1b, e, h, k; 2c, j-k; 5b-g; 6g-i; 7c, e, k, extended data figure(s) 1b; 5b,
- define scale bar for extended data figure(s) 1g-i; EV 3e and figures 3c-f,
- define white, yellow, blank arrows in the legend of figure 7i, and
- correct section order to the following: title page with complete author information, abstract, keywords, introduction, results, discussion, materials & methods, data availability section, acknowledgements, disclosure and competing interests statement, references, main figure legends, tables, expanded figure legends.

We include a synopsis of the paper (see <http://emboj.embopress.org/>). Please provide me with a two-sentence general summary statement and 3-5 bullet points that capture the key findings of the paper.

We also need a summary figure for the synopsis. The size should be 550 wide by [200-400] high (pixels). You can also use something from the figures if that is easier.

I look forward to receiving these changes. EMBO Press is an editorially independent publishing platform for the development of EMBO scientific publications.

Best wishes,

William

William Teale, PhD
Editor
The EMBO Journal
w.teale@embojournal.org

We realize that it is difficult to revise to a specific deadline. In the interest of protecting the conceptual advance provided by the work, we recommend a revision within 3 months (3rd Nov 2024). Please discuss the revision progress ahead of this time with the editor if you require more time to complete the revisions. Use the link below to submit your revision:

Referee #2:

Congratulations to the authors for a thoughtful revision. Even if they did not successfully address all of the concerns raised, their detailed explanation was adequate and clarified all the issues raised in the previous round of revision. I now believe that the paper can be further advanced towards publication.

Referee #3:

general summary and opinion about the principle significance of the study, its questions and findings

This is a re-revision of a manuscript by Lin and colleagues on the role of DELE1 in mitochondrial myopathy. The revised study is certainly clearer and the important elements now shine through. The authors have done a very good job of addressing my initial comments and concerns. The explanations are convincing and the additional experiments are compelling, especially their correlation between transcriptomics and microarray data. The additional bioinformatic and fiber type analyses are appreciated. The remaining issues to be addressed before publication are semantic and do not require further experimentation. The flow and overarching narrative need some refinement to ensure that the significance of the study and its unique elements are clear to a wide readership - I have tried to provide constructive examples in specific comments.

Overall, the study is well executed and has some striking physiological phenotypes. The findings on DELE1 have broad significance, demonstrating myoprotection and systemic benefits without affecting mitostructural defects. The tissue-specific dimension of the DELE1 transcriptional signature is also very interesting. All in all, I am very enthusiastic about this study - it highlights the physiological significance of the mt-ISR, and is a very welcome contribution to the field.

Specific Concerns###

1. Respectfully speaking, the introduction is simply too long. I suggest that the authors get to the "core question" as quickly as possible and this should be presented in the opening paragraphs to improve readability and engagement. As it stands, the entire first page introduction and the beginning of the second page could be strategically condensed with no negative impact to the manuscript, in fact it would enhance its maturity. A suggested example of how authors might achieve such a distillation is provided:

"The mitochondrial network generates cellular energy through oxidative phosphorylation (OXPHOS), which is critical for tissue homeostasis. Mutations in over 250 genes can cause primary mitochondrial disorders by disrupting OXPHOS and its biosynthetic pathways, often resulting in mitochondrial myopathy and abnormal growth. Patient survival may depend on how well striated muscle adapts to these perturbations in mitochondrial metabolism (Hathazi et al. 2020). A form of adaptation to

mitochondrial stress..."

If the authors are very keen on having a longer introduction, this valuable "real estate" could be better used to succinctly emphasize the biology, biochemistry, and known signaling mechanisms of DELE1 and the mtISR- which would be more appropriate compared to a generic introduction to mitochondria, OXPHOS and signaling.

2. The title "DELE1 is likely dispensable in the absence of mitochondrial stress" should be more specific. It would be helpful to clarify exactly what DELE1 is "likely dispensable" for. Consider changing the title to something more generic, such as: "Generation and basic characterization of a Dele1 KO mouse model" or more precise such as, "Basal mitochondrial homeostasis is unaltered in Dele1 KO mice".

I fully agree that the data presented clearly show that the authors' model is aphenotypic at steady state and has no significant gross macroanatomical features. However, a quick search on related models shows that the IMPC has already generated and phenotyped a Dele1 KO mouse and these data are in the public domain (<https://www.mousephenotype.org/data/genes/MGI:1914089>).

While the phenotypes reported by the IMPC are very specific, a brief reference to this information in the manuscript will enhance its timeliness i.e. it cannot be concluded that the mice are completely aphenotypic at steady state. In fact, these data largely support the authors' findings, i.e., no major changes, but some select phenotypic manifestations have been reported. The authors should refer to this characterization in the Results or Limitations section, the latter could be done by stating that the present characterisation was limited and this other data exists (again, not a criticism, but good for completeness).

3. The start of the results section related to Figure 5 (page 19 onwards) is very disjointed due to the abrupt transition from transcriptomics to metabolomics. However, a very simple and short introductory sentence could seamlessly integrate these sections.

For example, starting the section with something like this would be sufficient to guide the reader:

"We next performed metabolomics to understand how the DELE1-mtISR modifies tissue metabolism under mitochondrial stress."

4. While the BAT section is better placed within the manuscript, it still feels very disconnected from the overall study, both narratively and conceptually. Please consider modifying the description to better integrate these findings with the rest of the study. For example, a contrast between chronic and acute stress paradigms might prove effective here:

"Having defined that the DELE1 mt-ISR is activated in several preclinical models of mitochondrial disease that exhibit progressive and chronic pathology, we next investigated its tissue-specific modulation during acute metabolic stress."

5. I strongly believe that a separate limitations section should be included. While I have read and recognise the authors' point that this may not be common practice, it is indeed good scientific practice. Given that many important scientific messages were lost in the original formulation of the manuscript, streamlining these caveats into a dedicated limitations section can only strengthen this complex study. This also ensures responsible dissemination, which is critical for a study that discusses preclinical models, disease mechanisms, and future therapeutic avenues that may impact the patient community. Including this section does not diminish the authors' findings, but rather enhances their long-term credibility and is an invaluable opportunity for authors to mitigate unhelpful hype. Recent examples for reference - PMID: 38839991 and 37621214.

6. The discovery that DELE1 exerts a myoprotective effect but does not attenuate mitostructural defects a raises an important question: should ultrastructural defects in mitochondrial pathology be considered an "inherent feature" rather than a target phenotype to be ameliorated in translational efforts? This is particularly relevant in light of the extensive literature targeting mitochondrial network morphology as a central phenotypic marker to be targeted in various disease contexts. The authors should include a sentence on this, as it is an important finding.

minor concerns that should be addressed

Please update the EM label to be more specific, using a term such as "mitophagosome" or "autophagocytosed" or "sequestered" - rather than "mitophagy". It is not possible to accurately distinguish between non-selective and selective autophagy mechanisms using TEM analyses of complex tissues, and thus an alternative annotation of the structure, rather than the biological process, is more appropriate in the absence of additional experiments.

The authors should consider providing a final summary schematic of the various pathways/stressors and how they converge on DELE1.

any additional non-essential suggestions for improving the study (which will be at the author's/editor's discretion)

None, all are listed above. The authors have a study they can be very proud of. Well done!

Referee #2:

Congratulations to the authors for a thoughtful revision. Even if they did not successfully address all of the concerns raised, their detailed explanation was adequate and clarified all the issues raised in the previous round of revision.

I now believe that the paper can be further advanced towards publication.

We appreciate the Referee's finding our revision provides a sufficient clarification of concerns raised and that the manuscript is now in the Referee's view ready to be advanced toward publication.

Referee #3:

general summary and opinion about the principle significance of the study, its questions and findings

This is a re-revision of a manuscript by Lin and colleagues on the role of DELE1 in mitochondrial myopathy. The revised study is certainly clearer and the important elements now shine through. The authors have done a very good job of addressing my initial comments and concerns. The explanations are convincing and the additional experiments are compelling, especially their correlation between transcriptomics and microarray data. The additional bioinformatic and fiber type analyses are appreciated. The remaining issues to be addressed before publication are semantic and do not require further experimentation. The flow and overarching narrative need some refinement to ensure that the significance of the study and its unique elements are clear to a wide readership - I have tried to provide constructive examples in specific comments.

Overall, the study is well executed and has some striking physiological phenotypes. The findings on DELE1 have broad significance, demonstrating myoprotection and systemic benefits without affecting mitostructural defects. The tissue-specific dimension of the DELE1 transcriptional signature is also very interesting. All in all, I am very enthusiastic about this study - it highlights the physiological significance of the mt-ISR, and is a very welcome contribution to the field.

We appreciate that Referee finds the revision much improved and offering a welcome contribution to the field. We also appreciate the substantial effort the Referee put into the thoughtful suggestions below to further improve the clarity of the manuscript, all of which we have adopted in the revision.

Specific Concerns###

1. Respectfully speaking, the introduction is simply too long. I suggest that the authors get to the "core question" as quickly as possible and this should be presented in the opening paragraphs to improve readability and engagement. As it stands, the entire first page introduction and the beginning of the second page could be strategically condensed with no negative impact to the manuscript, in fact it would enhance its maturity. A suggested example of how authors might achieve such a distillation is provided:

"The mitochondrial network generates cellular energy through oxidative phosphorylation (OXPHOS), which is critical for tissue homeostasis. Mutations in over 250 genes can cause primary mitochondrial

disorders by disrupting OXPHOS and its biosynthetic pathways, often resulting in mitochondrial myopathy and abnormal growth. Patient survival may depend on how well striated muscle adapts to these perturbations in mitochondrial metabolism (Hathazi et al. 2020). A form of adaptation to mitochondrial stress..."

If the authors are very keen on having a longer introduction, this valuable "real estate" could be better used to succinctly emphasize the biology, biochemistry, and known signaling mechanisms of DELE1 and the mtISR- which would be more appropriate compared to a generic introduction to mitochondria, OXPHOS and signaling.

We have shortened the introduction along the lines suggested by the Referee.

2. The title "DELE1 is likely dispensable in the absence of mitochondrial stress" should be more specific. It would be helpful to clarify exactly what DELE1 is "likely dispensable" for. Consider changing the title to something more generic, such as:

"Generation and basic characterization of a Dele1 KO mouse model" or more precise such as, "Basal mitochondrial homeostasis is unaltered in Dele1 KO mice".

We have changed the title of this section to "Basal mitochondrial homeostasis is unaltered in Dele1 KO mice" as suggested by the Referee.

I fully agree that the data presented clearly show that the authors' model is aphenotypic at steady state and has no significant gross macroanatomical features. However, a quick search on related models shows that the IMPC has already generated and phenotyped a Dele1 KO mouse and these data are in the public domain (<https://www.mousephenotype.org/data/genes/MGI:1914089>).

While the phenotypes reported by the IMPC are very specific, a brief reference to this information in the manuscript will enhance its timeliness i.e. it cannot be concluded that the mice are completely aphenotypic at steady state. In fact, these data largely support the authors' findings, i.e., no major changes, but some select phenotypic manifestations have been reported. The authors should refer to this characterization in the Results or Limitations section, the latter could be done by stating that the present characterisation was limited and this other data exists (again, not a criticism, but good for completeness).

We have added this in our limitations section added in response to the Referee's suggestion below.

3. The start of the results section related to Figure 5 (page 19 onwards) is very disjointed due to the abrupt transition from transcriptomics to metabolomics. However, a very simple and short introductory sentence could seamlessly integrate these sections.

For example, starting the section with something like this would be sufficient to guide the reader: "We next performed metabolomics to understand how the DELE1-mtISR modifies tissue metabolism under mitochondrial stress."

We have added the suggested transition.

4. While the BAT section is better placed within the manuscript, it still feels very disconnected from the overall study, both narratively and conceptually. Please consider modifying the description to better integrate these findings with the rest of the study. For example, a contrast between chronic and acute stress paradigms might prove effective here:

"Having defined that the DELE1 mt-ISR is activated in several preclinical models of mitochondrial disease that exhibit progressive and chronic pathology, we next investigated its tissue-specific modulation during acute metabolic stress."

We have added the suggested transition.

5. I strongly believe that a separate limitations section should be included. While I have read and recognise the authors' point that this may not be common practice, it is indeed good scientific practice. Given that many important scientific messages were lost in the original formulation of the manuscript, streamlining these caveats into a dedicated limitations section can only strengthen this complex study. This also ensures responsible dissemination, which is critical for a study that discusses preclinical models, disease mechanisms, and future therapeutic avenues that may impact the patient community. Including this section does not diminish the authors' findings, but rather enhances their long-term credibility and is an invaluable opportunity for authors to mitigate unhelpful hype. Recent examples for reference - PMID: 38839991 and 37621214.

We have added a limitations section in which we now consolidate caveats that had appeared in the Discussion. We agree with the Referee that this does not diminish our findings and did not include one in the initial submission or revision as we understood this to be the Journal's preference.

6. The discovery that DELE1 exerts a myoprotective effect but does not attenuate mitostructural defects a raises an important question: should ultrastructural defects in mitochondrial pathology be considered an "inherent feature" rather than a target phenotype to be ameliorated in translational efforts? This is particularly relevant in light of the extensive literature targeting mitochondrial network morphology as a central phenotypic marker to be targeted in various disease contexts. The authors should include a sentence on this, as it is an important finding.

We agree this is an important finding. We have added a sentence on this point in the Discussion.

minor concerns that should be addressed

Please update the EM label to be more specific, using a term such as "mitophagosome" or "autophagocytosed" or "sequestered" - rather than "mitophagy". It is not possible to accurately distinguish between non-selective and selective autophagy mechanisms using TEM analyses of complex tissues, and thus an alternative annotation of the structure, rather than the biological process, is more appropriate in the absence of additional experiments.

We have changed these labels in the figures to "mitophagosome".

The authors should consider providing a final summary schematic of the various pathways/stressors and how they converge on DELE1.

This is provided now as the Summary Figure to go along with the synopsis, per the Journal's format.

any additional non-essential suggestions for improving the study (which will be at the author's/editor's discretion)

None, all are listed above. The authors have a study they can be very proud of. Well done!

We appreciate this positive assessment of our study and for the Referee's efforts to make it stronger.

Dear Dr. Narendra,

I am pleased to inform you that your manuscript has been accepted for publication in the EMBO Journal.

Congratulations on a really impressive piece of work!

Best wishes,

William

William Teale, PhD
Editor
The EMBO Journal
w.teale@embojournal.org
